# A lethal mitonuclear incompatibility in complex I of natural hybrids

Benjamin M. Moran[1,2 ✉], Cheyenne Y. Payne[1,2], Daniel L. Powell[1,2], Erik N. K. Iverson[3], Alexandra E. Donny[1], Shreya M. Banerjee[1], Quinn K. Langdon[1], Theresa R. Gunn[1], Rebecca A. Rodriguez-Soto[1], Angel Madero[2], John J. Baczenas[1], Korbin M. Kleczko[1], Fang Liu[4], Rowan Matney[4], Kratika Singhal[4], Ryan D. Leib[4], Osvaldo Hernandez-Perez[2], Russell Corbett-Detig[5,6], Judith Frydman[1,7], Casey Gifford[7,8,9], Manfred Schartl[10,11], Justin C. Havird[3] & Molly Schumer[1,2,12 ✉]

The evolution of reproductive barriers is the first step in the formation of new species and can help us understand the diversification of life on Earth. These reproductive barriers often take the form of hybrid incompatibilities, in which alleles derived from two different species no longer interact properly in hybrids[1–3]. Theory predicts that hybrid incompatibilities may be more likely to arise at rapidly evolving genes[4–6] and that incompatibilities involving multiple genes should be common[7,8], but there has been sparse empirical data to evaluate these predictions. Here we describe a mitonuclear incompatibility involving three genes whose protein products are in physical contact within respiratory complex I of naturally hybridizing swordtail fish species. Individuals homozygous for mismatched protein combinations do not complete embryonic development or die as juveniles, whereas those heterozygous for the incompatibility have reduced complex I function and unbalanced representation of parental alleles in the mitochondrial proteome. We find that the effects of different genetic interactions on survival are non-additive, highlighting subtle complexity in the genetic architecture of hybrid incompatibilities. Finally, we document the evolutionary history of the genes involved, showing signals of accelerated evolution and evidence that an incompatibility has been transferred between species via hybridization.

Biologists have long been fascinated by the question of how new species are formed and the mechanisms that maintain isolation between them. One key factor in the formation and maintenance of new species is the emergence of genetic incompatibilities that reduce viability or fertility in hybrids[1]. As originally described by the Dobzhansky–Muller model of hybrid incompatibility[2,3], the unique sets of mutations that accumulate in diverging species may interact poorly when they are brought together in hybrids, given that they have never been tested against one another by selection. Owing to the technical challenges of identifying these interactions[4], only around a dozen genes involved in hybrid incompatibilities have been precisely mapped[5] and exploration of the functional and evolutionary causes of hybrid incompatibilities has been limited to a small number of cases in model organisms[4].

This knowledge gap leaves key predictions about the evolutionary processes that drive the emergence of hybrid incompatibilities untested. For one, theory suggests that incompatibilities should be more common within dense gene networks, both because genes involved in such interactions are expected to be tightly co-evolving and

because the number of potentially incompatible genotypes explodes as the complexity of the genetic interaction increases[7,8]. Consistent with this prediction, mutagenesis experiments have highlighted the sensitivity of multi-protein interactions to changes in any of their components[8]. However, genetic interactions are notoriously difficult to detect empirically[9], and this problem is exacerbated with complex genetic interactions[10]. Such technical challenges may explain the rarity of incompatibilities involving three or more genes in the empirical literature[8] (but see refs. 9,11–14).

Another open question is the degree to which the genes that become involved in hybrid incompatibilities are predictable from their molecular or evolutionary properties. Researchers have proposed that rapid molecular evolution will increase the rate at which incompatibilities accumulate between species[4–6]. Although several known incompatibilities involve genes showing signatures of positive selection, it is unclear how unusual rates of protein evolution are in these genes relative to the genomic background[5,6]. The mitochondrial genome, in particular, has been proposed as a hotspot for the accumulation of

[1]Department of Biology, Stanford University, Stanford, CA, USA. [2]Centro de Investigaciones Científicas de las Huastecas 'Aguazarca', A.C., Calnali, Hidalgo, Mexico. [3]Department of Integrative Biology, University of Texas at Austin, Austin, TX, USA. [4]Stanford University Mass Spectrometry Core, Stanford University, Stanford, CA, USA. [5]Genomics Institute, University of California Santa Cruz, Santa Cruz, CA, USA. [6]Department of Biomolecular Engineering, University of California Santa Cruz, Santa Cruz, CA, USA. [7]Department of Genetics, Stanford University, Stanford, CA, USA. [8]Department of Pediatrics, Stanford University, Stanford, CA, USA. [9]Institute for Stem Cell Biology and Regenerative Medicine, Stanford University, Stanford, CA, USA. [10]The Xiphophorus Genetic Stock Center, Texas State University, San Marcos, TX, USA. [11]Developmental Biochemistry, Biozentrum, University of Würzburg, Würzburg, Germany. [12]Howard Hughes Medical Institute, Stanford, CA, USA. ✉e-mail: benmoran@stanford.edu; schumer@stanford.edu

genetic incompatibilities[15,16], owing to substitution rates up to 25 times higher than the nuclear genome in many animals[17,18] and the potential for sexually antagonistic selection driven by its predominantly maternal inheritance[19,20], among other factors[21]. At the same time, nuclear and mitochondrial proteins must interact with each other in key steps of ATP synthesis, increasing the likelihood of coevolution between these genomes[22,23]. These factors suggest that interactions between mitochondrial- and nuclear-encoded proteins could have an outsized role in the emergence of hybrid incompatibilities[15], consistent with results from numerous species[24–26].

As we begin to identify the individual genes underlying hybrid incompatibilities, the next frontier is evaluating the processes that drive their evolution. Over the past two decades, it has become abundantly clear that hybridization is exceptionally common in species groups where it was once thought to be rare[27,28]. As a result, it is now appreciated how frequently species derive genes from their relatives[29–31]. The effects of historical hybridization on the evolution of hybrid incompatibilities have been poorly investigated[32], since the foundational theory in this area was developed before the ubiquity of hybridization was fully appreciated[7].

Here we use an integrative approach to precisely map the genetic basis and physiological effects of a lethal mitonuclear hybrid incompatibility in swordtail fish and uncover its evolutionary history. The sister species *Xiphophorus birchmanni* and *Xiphophorus malinche* began hybridizing approximately 100 generations ago in multiple river systems[33] after premating barriers were disrupted by habitat disturbance[34], and are a powerful system to study the emergence of hybrid incompatibilities in young species. Despite their recent divergence[35] (around 250,000 generations; 0.5% divergence per basepair), some hybrids between *X. birchmanni* and *X. malinche* experience strong selection against incompatibilities[35,36]. One incompatibility that causes melanoma has been previously mapped in this system and population genetic patterns suggest that dozens may be segregating in natural hybrid populations[35–38]. Moreover, the ability to generate controlled crosses[39,40] and the development of high-quality genomic resources[38,41] makes this system particularly tractable for studying hybrid incompatibilities in natural populations. Leveraging data from controlled laboratory crosses and natural hybrid populations, we pinpoint two nuclear-encoded *X. birchmanni* genes that are lethal when mismatched with the *X. malinche* mitochondria in hybrids, explore the developmental and physiological effects of this incompatibility, and trace its evolutionary history.

## Mapping mitonuclear incompatibilities

To identify loci under selection in *X. birchmanni* × *X. malinche* hybrids, we generated approximately 1× low-coverage whole-genome sequence data for 943 individuals from an $F_2$ laboratory cross and 359 wild-caught hybrid adults, and applied a hidden Markov model to data at more than 600,000 ancestry-informative sites along the genome to infer local ancestry (approximately 1 informative site per kilobase[37,42]; Methods and Supplementary Information 1.1.1–1.1.4). Using these results, we found evidence for a previously unknown incompatibility between the nuclear genome of *X. birchmanni* and the mitochondrial genome of *X. malinche* (Supplementary Information 1.1.5–1.1.10). Our first direct evidence for this incompatibility came from controlled laboratory crosses (Methods and Supplementary Information 1.1.1). Because the cross is largely unsuccessful in the opposite direction, all laboratory-bred hybrids were the offspring of $F_1$ hybrids generated between *X. malinche* females and *X. birchmanni* males and harboured a mitochondrial haplotype derived from the *X. malinche* parent species. Offspring of $F_1$ intercrosses are expected to derive on average 50% of their genome from each parent species. This expectation is satisfied genome wide and locally along most chromosomes in $F_2$ hybrids (on average 50.3% *X. malinche* ancestry; Supplementary Fig. 1). However, we detected six

segregation distorters genome wide[40], with the most extreme signals falling along a 6.5 Mb block of chromosome 13 and a 4.9 Mb block of chromosome 6 (Fig. 1a,d).

Closer examination of genotypes in the chromosome 13 region showed that almost none of the surviving individuals harboured homozygous *X. birchmanni* ancestry in a 3.75 Mb subregion (Fig. 1c and Supplementary Fig. 2; 0.1% observed versus 25% expected). This pattern is unexpected even in the case of a lethal incompatibility involving only nuclear loci (see Supplementary Information 1.1.1), but is consistent with a lethal mitonuclear incompatibility. Using approximate Bayesian computation (ABC) approaches we inferred the strength of selection against *X. birchmanni* ancestry in this region that was consistent with the observed genotypes and ancestry deviations. We estimated posterior distributions of selection and dominance coefficients and inferred that selection on this genotype in $F_2$ is largely recessive and essentially lethal (maximum a posteriori estimate $h = 0.12$ and $s = 0.996$, 95% credible interval $h = 0.010–0.194$ and $s = 0.986–0.999$; Fig. 1b, Extended Data Fig. 1, Methods and Supplementary Information 1.2.1–1.2.2).

The degree of segregation distortion observed in $F_2$ individuals on chromosome 6 is also surprising (Fig. 1d). Only 3% of individuals harbour homozygous *X. birchmanni* ancestry in this region (compared with 0.1% in the chromosome 13 region and 25% on average at other loci across the genome; Fig. 1f), which is again lower than expected for a nuclear–nuclear hybrid incompatibility (Supplementary Information 1.1.1). ABC approaches indicate that selection on homozygous *X. birchmanni* ancestry on chromosome 6 is also severe (maximum a posteriori estimate $h = 0.09$ and $s = 0.91$, 95% credible interval interval $h = 0.01–0.21$ and $s = 0.87–0.94$; Fig. 1e, Extended Data Fig. 1 and Supplementary Information 1.2.2). Thus, our $F_2$ data show that homozygous *X. birchmanni* ancestry in regions of either chromosome 13 or chromosome 6 is almost completely lethal in hybrids with *X. malinche* mitochondria (Fig. 1h).

To formally test for the presence of a mitonuclear incompatibility involving chromosome 13 and chromosome 6, or elsewhere in the genome, we leveraged data from natural hybrid populations. Most naturally occurring *X. birchmanni* × *X. malinche* hybrid populations are fixed for mitochondrial haplotypes from one parental species (Supplementary Information 1.1.2 and 1.1.6). However, a few populations segregate for the mitochondrial genomes of both parental types, and we focused on one such population (the 'Calnali low' population, hereafter referred to as the admixture mapping population). Admixture mapping for associations between nuclear genotype and mitochondrial ancestry (after adjusting for expected covariance due to genome-wide ancestry[36]) revealed two genome-wide significant peaks and one peak that approached genome-wide significance (Fig. 1g and Supplementary Tables 1–3). The strongest peak of association spanned approximately 77 kb and fell within the region of chromosome 13 identified using $F_2$ crosses (Fig. 1g). This peak was also replicated in another hybrid population (Methods, Supplementary Fig. 3 and Supplementary Information 1.1.5) and contains only three genes: the NADH dehydrogenase ubiquinone iron–sulfur protein 5 (*ndufs5*), E3 ubiquitin–protein ligase, and microtubule–actin cross-linking factor 1. Of these three genes, only *ndufs5* forms a protein complex with mitochondrially encoded proteins, which along with other evidence implicates it as one of the interacting partners in the mitonuclear incompatibility (Fig. 1c, Extended Data Fig. 2 and Supplementary Fig. 4; see Supplementary Information 1.1.6–1.1.9).

We also identified a peak on chromosome 6 that approached genome-wide significance (Fig. 1g, Supplementary Fig. 5, Supplementary Table 2 and Supplementary Information 1.1.10) and fell precisely within the segregation distortion region previously mapped in $F_2$ hybrids (Fig. 1d and Supplementary Information 1.1.1). This peak contained 20 genes, including the mitochondrial complex I gene *ndufa13* (Extended Data Fig. 2, Supplementary Fig. 5, Supplementary Information 1.1.10 and Methods). Depletion of non-mitochondrial

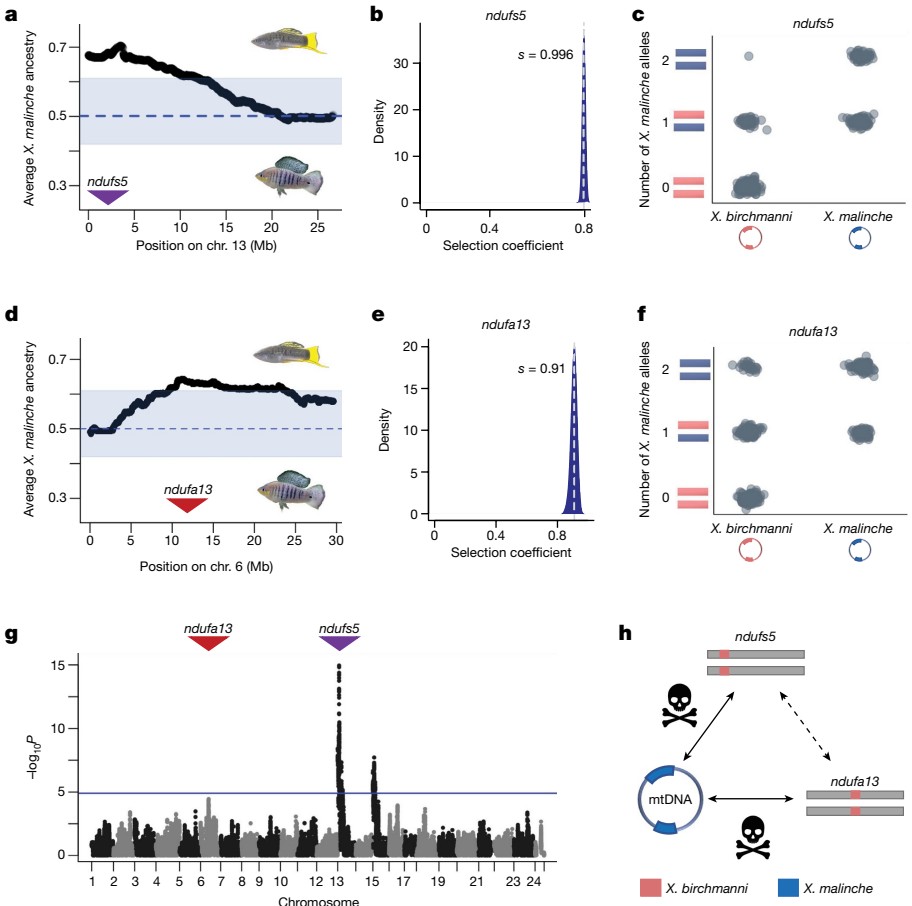

**Fig. 1 | Admixture mapping pinpoints mitonuclear incompatibility in *Xiphophorus*. a**, Average ancestry of F$_2$ hybrids on chromosome (chr.) 13 reveals a large region of segregation distortion towards *X. malinche* ancestry. The region shaded blue shows the 99% quantiles of *X. malinche* ancestry at all ancestry informative sites genome wide. The dashed line represents expected *X. malinche* ancestry for this cross. The purple arrow points to the position of *ndufs5*. **b**, Results of ABC simulations estimating the strength of selection on *X. malinche* mitochondria combined with *X. birchmanni* ancestry at *ndufs5*. Shown is the posterior distribution from accepted simulations; the vertical line indicates the maximum a posteriori estimate for selection coefficient (*s*). **c**, Observed genotype frequencies of different genotype combinations of *ndufs5* and mitochondrial haplotypes in admixture mapping population. **d**, Average ancestry of F$_2$ individuals on chromosome 6 reveals a large region of segregation distortion towards *X. malinche* ancestry. The region shaded blue shows the 99% ancestry quantiles and expected ancestry as in **a**, and the red arrow points to position of *ndufa13*. **e**, Results of ABC simulations estimating the strength of selection on *X. malinche* mitochondria combined with *X. birchmanni ndufa13*, as in **b**. **f**, Observed genotype frequencies of different genotype combinations of *ndufa13* and mitochondrial haplotypes in the admixture mapping population. **g**, Admixture mapping of associations between nuclear ancestry and mitochondrial haplotype in natural hybrids using a partial correlation approach, controlling for genome-wide ancestry. The blue line indicates the 10% false-positive rate genome-wide significance threshold determined by simulations. The peak visible on chromosome 15 is driven by interactions with the *X. birchmanni* mitochondria and an unknown nuclear gene (Supplementary Information 1.1.10 and 1.2.1). **h**, Schematic of identified interactions with the *X. malinche* mitochondrial genome from mapping data. The dashed line indicates a subtle interaction between *ndufs5* and *ndufa13* (see text, Fig. 2 and Supplementary Information 1.2.5).

parent ancestry at *ndufa13* was unidirectional (Fig. 1f), consistent with selection acting only against the combination of the *X. malinche* mitochondria with homozygous *X. birchmanni* ancestry at *ndufa13* (see Supplementary Information 1.2.3–1.2.4). Genomic analyses in natural hybrid populations confirmed this asymmetry (Extended Data Fig. 2).

Together, these results indicate that at least two *X. birchmanni* nuclear genes cause incompatibility when they are mismatched in ancestry with the *X. malinche* mitochondria (Fig. 1h and Supplementary Information 1.2.5). These genes, *ndufs5* and *ndufa13*, belong to a group of proteins and assembly factors that form respiratory complex I (ref. 43) (see Supplementary Table 1 for locations of the 51 annotated complex I genes in the *Xiphophorus* genome). Complex I is the first component of the mitochondrial electron transport chain that ultimately enables the cell to generate ATP. Both nuclear proteins interface with several mitochondrially derived proteins at the core of the complex I

structure, hinting at the possibility that physical interactions could underlie this multi-gene mitonuclear incompatibility.

## Interactions with *X. birchmanni* mitochondrial DNA

Admixture mapping analysis also identified a strong peak of mitonuclear association on chromosome 15, which we briefly discuss here and in Supplementary Information 1.1.10 and 1.2.1. This peak was associated with *X. birchmanni* mitochondrial ancestry (Extended Data Fig. 3), indicating that it has a distinct genetic architecture from the incompatibility involving the *X. malinche* mitochondria and *X. birchmanni ndufs5* and *ndufa13*. Specifically, analysis of genotypes at the admixture mapping peak indicates that *X. birchmanni* mitochondrial ancestry is incompatible with homozygous *X. malinche* ancestry on chromosome 15 (Fig. 1c and Extended Data Fig. 3). This region did not contain any members of complex I, but dozens of genes in this

interval interact with known mitonuclear genes (see Supplementary Table 3 and Supplementary Information 1.1.10). The fact that we detect incompatible interactions with both the *X. malinche* mitochondria (at *ndufs5* and *ndufa13*) and the *X. birchmanni* mitochondria (*ndufs5* and chromosome 15) in our admixture mapping results supports the idea that mitonuclear interactions can act as 'hotspots' for the evolution of hybrid incompatibilities[15].

## Lethal effects in early development

The combination of *X. birchmanni ndufs5* or *ndufa13* with the *X. malinche* mitochondria appears to be lethal by the time individuals reach adulthood. To investigate the developmental timing of the incompatibility, we genotyped pregnant females from the admixture mapping population and recorded the developmental stages of their embryos[44] (swordtails are livebearing fish; Methods). We found a significant interaction between developmental stage and *ndufs5* genotype, whereas *ndufa13* genotype did not affect developmental stage (Fig. 2a,b, Supplementary Figs. 6–9 and Supplementary Information 1.3.1). Genotyping results revealed that embryos with homozygous *X. birchmanni* ancestry at *ndufs5* and *X. malinche* mitochondria are present at early developmental stages, but that these embryos did not develop beyond a phenotype typical of the first seven days of gestation (the full length of gestation is 21–28 days in *Xiphophorus*; Fig. 2a,b,d,e). Individuals with mismatched ancestry at *ndufs5* whose siblings were fully developed still had a detectable heartbeat but had consumed less yolk than their siblings and remained morphologically underdeveloped (Fig. 2d, Extended Data Fig. 4 and Supplementary Figs. 10–14). Unlike other species, in *Xiphophorus* this developmental lag could itself cause mortality, since embryos that do not complete embryonic development inside the mother do not survive more than a few days after birth (Supplementary Information 1.3.1 and Supplementary Table 4). Given that complex I inhibition lethally arrests development in zebrafish embryos[45,46], we also tested the effects of complex I inhibition on *X. birchmanni* and *X. malinche* fry, and found a similar level of sensitivity (Supplementary Information 1.3.2).

In contrast to individuals with mismatched ancestry at *ndufs5*, those with *ndufa13* mismatch survived embryonic development but suffered mortality in the early post-natal period (Fig. 2c). We tracked 74 F$_2$ fry from 24 h post birth to adulthood (Supplementary Information 1.3.3). We found that most fry with incompatible genotypes at *ndufa13* had already suffered mortality by the time tracking began, with only 7 individuals found 24 h post birth that were homozygous *X. birchmanni* at *ndufa13* (versus 19 expected; binomial *P* = 0.0005). No natural mortality was observed between 1 day and 3 months post birth (Supplementary Information 1.3.3).

## Physiology and complex fitness effects

Our analysis of developing embryos indicates that individuals with the *ndufs5* incompatibility exhibited abnormal embryonic development, whereas those with the *ndufa13* incompatibility did not. This suggests that these genes may drive lethality through partially distinct mechanisms. Thus, we chose to further investigate the effects of *ndufs5* and *ndufa13*, and their possible interactions. We sampled 235 F$_2$ embryos at a range of developmental stages and measured their overall rates of respiration (Supplementary Information 1.3.4–1.3.5). We also used imaging of these embryos to track cardiovascular phenotypes as these have been associated with *ndufa13* defects in mammals[47]. We found that incompatible genotypes at *ndufs5* and *ndufa13* affected a range of phenotypes, including heart rate, length relative to compatible siblings, and length-corrected head size (Extended Data Figs. 4 and 5, Supplementary Figs. 11–17 and Supplementary Tables 5–7). *ndufa13* mismatch has a large effect on cardiovascular phenotypes, including heart rate and the size of the sinu-atrium (an embryo-specific heart

chamber; Fig. 2g,h, Supplementary Figs. 14 and 16 and Supplementary Tables 6 and 8), whereas *ndufs5* affects only heart rate (Supplementary Fig. 14 and Supplementary Table 6). We find initial evidence that cardiac defects persist into adulthood in surviving individuals with *ndufa13* mismatch (Extended Data Fig. 5 and Supplementary Information 1.3.3). By contrast, *ndufs5* mismatch has a major effect on rates of respiration and yolk consumption during development (Fig. 2f, Supplementary Figs. 18–20 and Supplementary Tables 9–11).

Naively, the separable impacts of incompatible genotypes at *ndufs5* and *ndufa13* could indicate that even though these proteins are in physical contact in complex I (see below), they represent two distinct hybrid incompatibilities. We investigated this question by taking advantage of rare survivors of the *ndufa13* incompatibility in an expanded dataset of 1,010 F$_2$ hybrids. Using this dataset, we were able to identify dozens of survivors of the *ndufa13* incompatibility (3.4% of individuals) and found that genotypes at *ndufa13* and *ndufs5* were not independent ($\chi^2$ association test *P* = 0.032; Supplementary Information 1.2.5). Upon further investigation, we found that the majority of survivors of the *ndufa13* incompatibility had homozygous *X. malinche* ancestry at *ndufs5*, suggesting that harbouring even one *X. birchmanni* allele at *ndufs5* may sensitize fry to the *ndufa13* incompatibility. Indeed, we found that individuals that had heterozygous ancestry at *ndufs5* were significantly under-represented among surviving *ndufa13* incompatible individuals (Permutation test *P* = 0.015; Fig. 2i and Supplementary Information 1.2.5). These findings highlight a subtle but significant non-additive effect of *ndufs5* and *ndufa13* on survival.

## Mitochondrial biology in heterozygotes

Because few individuals homozygous for incompatible genotypes at *ndufs5* or *ndufa13* survive past birth, our previous experiments focused on embryos. However, the small size of *Xiphophorus* embryos prevents us from using assays that directly target complex I. To further explore the effects of the hybrid incompatibility on complex I function in vivo, we turned to adult F$_1$ hybrids (Fig. 3a). Since F$_1$ hybrids that derive their mitochondria from *X. malinche* and are heterozygous for ancestry at *ndufs5* and *ndufa13* are fully viable, we tested whether there was evidence for compensatory regulation that might be protective in F$_1$ hybrids. We found no evidence for significant differences in expression of *ndufs5* or *ndufa13* (Supplementary Information 1.3.6 and Supplementary Figs. 21 and 22) or in mitochondrial copy number (Fig. 3b and Supplementary Information 1.3.7) between F$_1$ hybrids and parental species.

With no indication of a compensatory regulatory response, we reasoned that we might be able to detect reduced mitochondrial complex I function in hybrids heterozygous for ancestry at *ndufs5* and *ndufa13*. We quantified respiratory phenotypes in isolated mitochondria using an Oroboros O2K respirometer in adult hybrids and parental species (Methods, Supplementary Fig. 23 and Supplementary Information 1.3.8). We found that complex I efficiency was lower in hybrids (Fig. 3c and Supplementary Fig. 24, orthogonal contrast *t* = −2.53, *P* = 0.023, *n* = 7 per genotype), and that the time required for hybrids to reach maximum complex I-driven respiration was around 2.5 times longer (orthogonal contrast *t* = 4.303, *P* < 0.001; Fig. 3d and Supplementary Fig. 25). Conversely, overall levels of mitochondrial respiration were unaffected by genotype (Fig. 3e, orthogonal contrast *t* = 0.078, *P* = 0.94, *n* = 7 per genotype; Supplementary Information 1.3.8) as were other measures of mitochondrial integrity and function (Supplementary Figs. 26 and 27 and Supplementary Information 1.3.8 and 1.3.9). Together, these data point to reduced function of complex I without broader phenotypic consequences in individuals that are heterozygous for incompatible alleles[48].

Given the physiological evidence for some reduction in complex I function in hybrids heterozygous at *ndufs5* and *ndufa13*, we predicted that there might be an altered frequency of protein complexes incorporating both *X. malinche* mitochondrial proteins and *X. birchmanni*

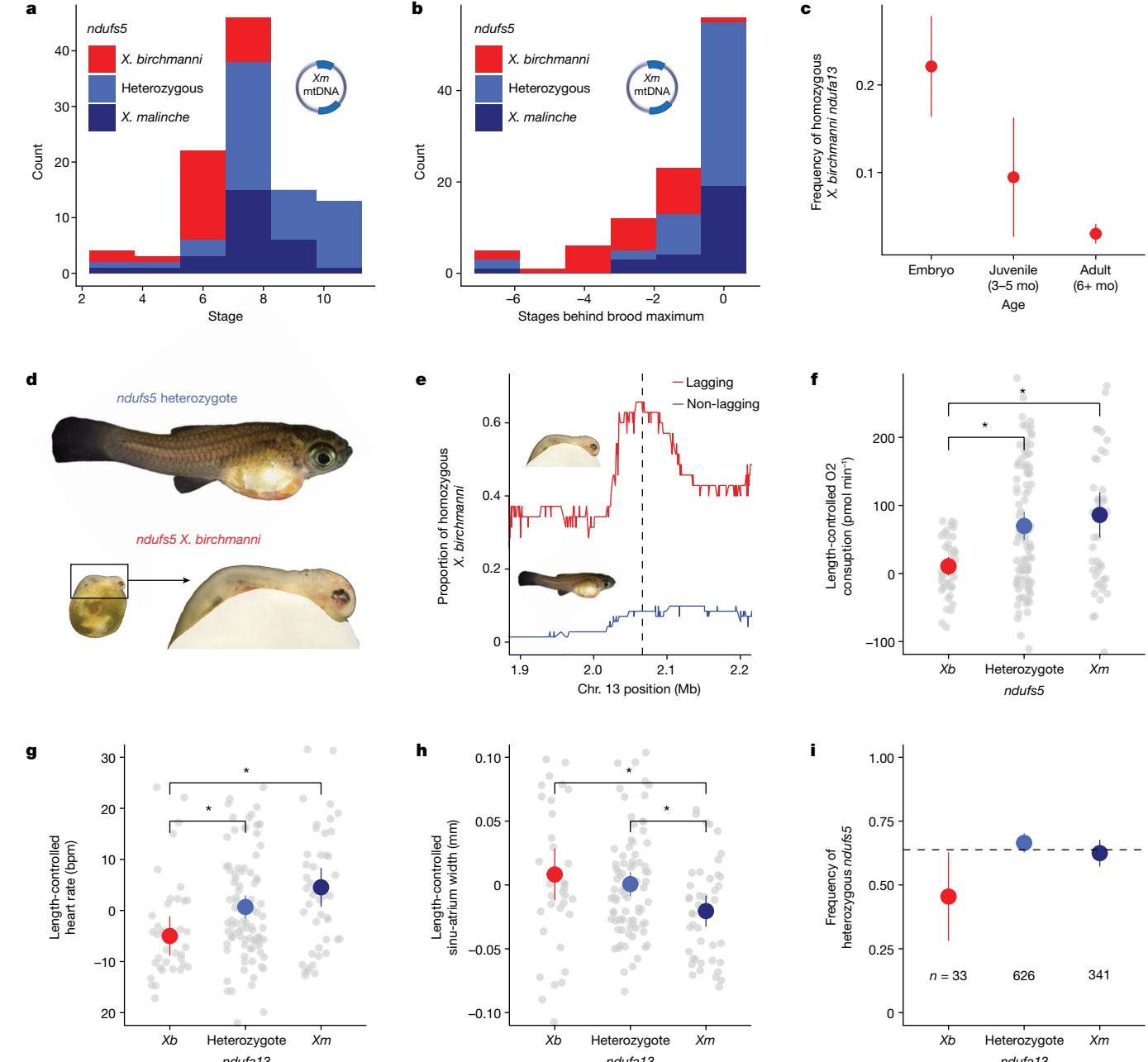

**Fig. 2 | Effect of incompatibility on *Xiphophorus* hybrid embryos.**
**a**, Developmental stage and *ndufs5* genotypes of hybrid embryos with *X. malinche* (*Xm*) mitochondria. **b**, Developmental lag of embryos with *X. malinche* mitochondria with varying *ndufs5* genotypes compared with their most developed broodmate. **c**, Frequency of homozygous *X. birchmanni ndufa13* ancestry over $F_2$ hybrid development. Dots and lines represent observed frequency ± 2 × s.e.m. (*n* = 208 embryos, 74 juveniles, 932 adults). **d**, $F_2$ siblings showing different phenotypes as a function of *ndufs5* genotype: a heterozygote (top) and an *ndufs5*-incompatible sibling with matched scale (bottom left) and magnified (bottom right). **e**, Frequency of homozygous *X. birchmanni* ancestry along chromosome 13 in embryos with *X. malinche* mitochondrial DNA (mtDNA) that lagged their siblings by at least 1 developmental stage (red) versus those without developmental lag (blue) (Supplementary Information 1.1.7). Dashed line indicates *ndufs5* location. Corresponding analyses of chromosomes 6 and 15 are shown in Supplementary Figs. 7–9. **f**–**h**, Respiratory and morphometric

measurements in $F_2$ embryos. To control for length, residuals of each variable regressed against length are plotted (Supplementary Information 1.3.4 and 1.3.5). Grey dots denote individual measurements, coloured points and bars show mean ± 2 × s.e.m. for each genotype, and brackets with asterisks denote significant differences from Tukey's honest significant difference test. **f**, Relationship between *ndufs5* genotype and respiration rate in hybrids (*n* = 40 *X. birchmanni*, 102 heterozygotes, 47 *X. malinche*). *Xb*, *X. birchmanni*. **g**, Relationship between *ndufa13* genotype and heart rate in hybrids (*n* = 39 *X. birchmanni*, 95 heterozygotes, 46 *X. malinche*). **h**, Relationship between *ndufa13* genotype and width of sinu-atrium, a peristaltic canal between the yolk and embryonic atrium, in hybrids (*n* = 37 *X. birchmanni*, 82 heterozygotes, 42 *X. malinche*). **i**, Frequency of heterozygotes at *ndufs5* among juveniles and adults of varying *ndufa13* genotypes. Dots and lines represent observed frequency ± 2 × s.e.m. The dashed line represents expected frequency under additive selection (Supplementary Information 1.2.5).

proteins at *ndufs5* and *ndufa13* in $F_1$ hybrids. To test this prediction, we took a mass spectrometry-based quantitative proteomics approach. We used stable isotope-labelled peptides to distinguish between the

*X. birchmanni* and *X. malinche ndufs5* and *ndufa13* peptides in mitochondrial proteomes extracted from $F_1$ hybrids (*n* = 5; see Methods and Supplementary Information 1.4.1–1.4.4). Although endogenous

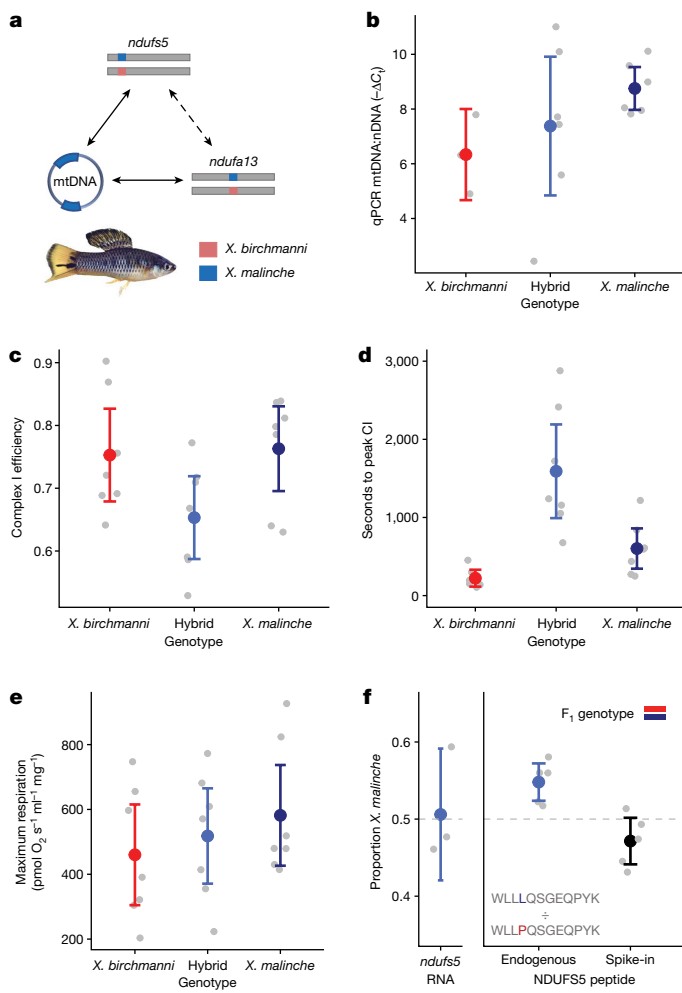

**Fig. 3 | Physiology and proteomics of viable heterozygotes. a**, Schematic of ancestry at loci of interest in *X. birchmanni × malinche* F₁ hybrids (the incompatibility is not lethal in F₁; Supplementary Information 1.2.2). **b**, Real-time quantitative PCR analysis of differences in mitochondrial copy number in liver tissue by genotype. The ratio is derived from the difference in $C_t$ between the single-copy nuclear gene *Nup43* and the mitochondrial gene *nd1* (*n* = 3 *X. birchmanni*, 6 F₁, 6 *X. malinche*). **c**–**e**, Results of Oroboros O2K respirometer assay for adult *X. birchmanni*, *X. malinche* and hybrid individuals (*n* = 7 fish per genotype), with *X. malinche* mitochondria and heterozygous ancestry at *ndufs5* and *ndufa13*. **c**, Complex I efficiency in F₁ hybrids and parental species. **d**, Time to reach the maximum rate of complex I-driven respiration after ADP addition in F₁ hybrids and parental species (see Supplementary Fig. 25 for example time-to-peak curves). For corresponding analysis of time to reach peak complex I- and complex II-driven respiration after addition of succinate, see Supplementary Fig. 26. **e**, Maximum respiration rates during full O2K protocol in F₁ hybrids and parental species. **f**, Allelic balance of *ndufs5* in the F₁ hybrid transcriptome and proteome. Left, allele-specific expression of *ndufs5* in adult F₁ hybrids (*n* = 3 fish). Right, results of quantitative mass spectrometry analysis of *ndufs5* peptides in mitochondrial proteomes derived from adult F₁ hybrids (*n* = 5 fish). Dots show the proportion of area under the spectral curve contributed by the *X. malinche* allele in each individual. The data on the left are for endogenous peptides present in F₁ hybrids, and the data on the right are results for heavy isotope-labelled peptide controls. The peptides quantified for each species are shown on the graph. **a**–**f**, Coloured dots and bars show the mean ± 2 × s.e.m., and grey dots show individual data.

*ndufa13* peptides were not observed frequently enough to quantify accurately, we found consistent deviations from the expected 50:50 ratio of *X. birchmanni* to *X. malinche* peptides for *ndufs5* in F₁ hybrids, with a significant overrepresentation of matched ancestry at *ndufs5* in the mitochondrial proteome (*t* = 3.96, *P* = 0.016; Fig. 3f, Supplementary

Fig. 28 and Supplementary Information 1.4.5). Since we did not observe allele-specific expression of *ndufs5* (Fig. 3f and Supplementary Information 1.3.6), this result is consistent with disproportionate degradation of *X. birchmanni*-derived *ndufs5* peptides in the mitochondrial proteome or differences in translation of *ndufs5* transcripts derived from the two species.

## Mitonuclear substitutions in complex I

To begin to explore the possible mitochondrial partners of *ndufs5* and *ndufa13* among the 37 non-recombining genes in the swordtail mitochondrial genome, we turned to protein modelling, relying on high-quality cryo-electron microscopy (cryo-EM)-based structures[49–51]. Although these structures are only available for distant relatives of swordtails, the presence of the same set of supernumerary complex I subunits and high sequence similarity suggest that using these structures is appropriate (Supplementary Tables 12 and 13, Supplementary Figs. 29–31 and Supplementary Information 1.4.6).

Barring a hybrid incompatibility generated by regulatory divergence (see Supplementary Information 1.3.6), our expectation is that hybrid incompatibilities will be driven by amino acid changes in interacting proteins[52]. We used the program RaptorX[53] to generate predicted structures of *X. birchmanni* and *X. malinche* Ndufs5, Ndufa13 and nearby complex I proteins encoded by mitochondrial and nuclear genes, which we aligned to a mouse cryo-EM complex I structure[49] (Fig. 4a, Supplementary Figs. 29–31 and Methods). Using these structures, we visualized amino acid substitutions between *X. birchmanni* and *X. malinche* at the interfaces of Ndufs5, Ndufa13 and mitochondrial-encoded proteins (Extended Data Fig. 6 and Supplementary Figs. 32 and 33). Whereas there are dozens of substitutions in the four mitochondrial-encoded proteins that are in close physical proximity to Ndufs5 or Ndufa13 (Supplementary Fig. 29; Nd2, Nd3, Nd4l and Nd6), there are only five cases where amino acid substitutions in either nuclear-encoded protein are predicted to be close enough to contact substitutions in any mitochondrial-encoded protein, all of which involve Nd2 or Nd6 (Fig. 4a and Extended Data Table 1; see Supplementary Fig. 33 for pairwise visualizations of interacting proteins). These paired substitutions in regions of close proximity between mitochondrial- and nuclear-encoded proteins suggest that *nd2* and *nd6* are the genes most likely to be involved in the mitochondrial component of the hybrid incompatibility (Fig. 4a,b Extended Data Fig. 6 and Supplementary Figs. 33–35), and will be promising candidates for functional validation when such approaches become possible in swordtails.

## Rapid evolution of complex I proteins

Theory predicts that hybrid incompatibilities are more likely to arise in rapidly evolving genes[4–7]. Consistent with this hypothesis, *ndufs5* is among the most rapidly evolving genes genome-wide between *X. birchmanni* and *X. malinche* (Fig. 4c,d). Aligning the *ndufs5* coding sequences of *X. birchmanni*, *X. malinche* and 12 other swordtail species revealed that all 4 amino acid substitutions that differentiate *X. birchmanni* and *X. malinche* at *ndufs5* were derived on the *X. birchmanni* branch (Fig. 4c). Phylogenetic tests indicate that there has been accelerated evolution of *ndufs5* on this branch (inferred ratio of non-synonymous substitutions per non-synonymous site to synonymous substitutions per synonymous site ($d_N/d_S$) > 99, *N* = 4, *S* = 0, codeml branch test *P* = 0.005; Fig. 4c). Similar patterns of rapid evolution are observed at *ndufa13*, which also showed evidence for accelerated evolution in *X. birchmanni* (Fig. 4e; $d_N/d_S$ = 1.2, *N* = 3, *S* = 1, codeml branch test *P* = 0.002). Although explicit tests for adaptive evolution at *ndufs5* and *ndufa13* could not exclude a scenario of relaxed selection (Extended Data Table 2 and Supplementary Information 1.5.1 and 1.5.2), our comparisons across phylogenetic scales highlight strong conservation in some regions of

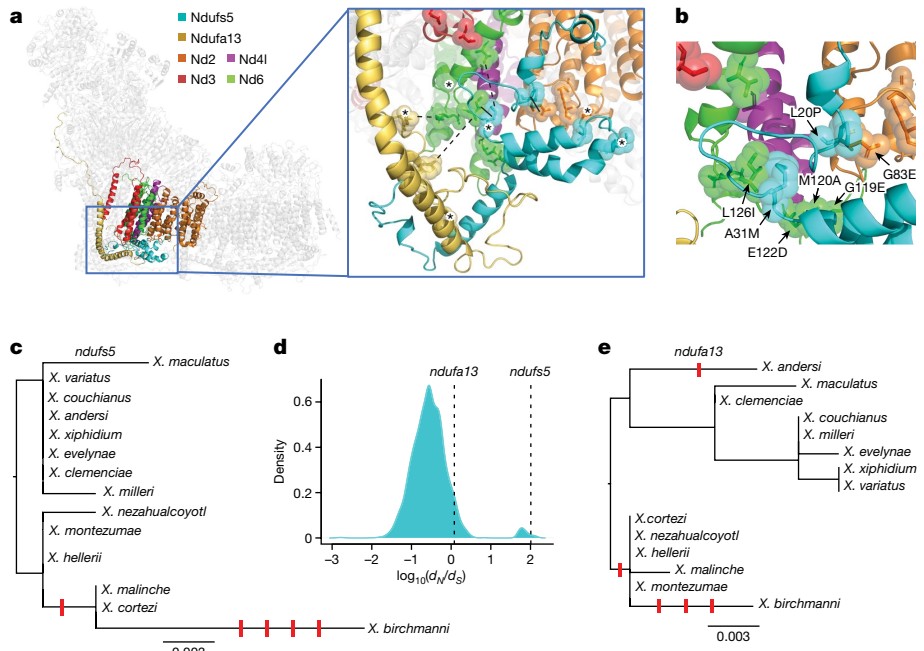

**Fig. 4 | Predicted structures of *Xiphophorus* respiratory complex I and evolutionary rates of incompatible alleles. a**, Left, *Xiphophorus* respiratory complex I structures generated by RaptorX using alignment to a template mouse cryo-EM structure. Coloured protein structures include Ndufs5 and Ndufa13 and the four mitochondrial-encoded nd gene products in contact with Ndufs5 or Ndufa13. Right, expanded view showing the surface of predicted contacts between these proteins. Solid black lines highlight two areas predicted to be in close contact between interspecific substitutions (alpha carbon distance ≤ 10 Å for all models), and dashed lines show three additional areas in which there was weaker evidence for a predicted contact based on computational analyses (side chain distance ≤ 12 Å in at least one model). Asterisks denote residues with substitutions in *X. birchmanni* computationally predicted to affect protein function (Extended Data Table 3). **b**, Detailed view of the interface between Ndufs5, Nd2 and Nd6. Spheres highlight substitutions between *X. birchmanni* and *X. malinche*. For substitutions predicted to be in close proximity, residues are labelled with letters denoting the *X. malinche* allele, the residue number, and the *X. birchmanni* allele, respectively (Extended Data Table 3 and Supplementary Information 1.4.6). **c**, Gene tree for *ndufs5* generated with RAxML, highlighting an excess of substitutions along the *X. birchmanni* branch. The scale bar represents the number of nucleotide substitutions per site. Derived non-synonymous substitutions are indicated by red ticks along the phylogeny; spacing between ticks is arbitrary. **d**, Distribution of $\log_{10}(d_N/d_S)$ between *X. birchmanni* and *X. malinche* across all nuclear genes with values for *ndufs5* and *ndufa13* highlighted. **e**, Gene tree for *ndufa13* (as in **c**), highlighting an excess of substitutions along the *X. birchmanni* branch.

the proteins and rapid turnover in others, complicating our interpretation of this test (Supplementary Fig. 36).

Rapid evolution of *ndufs5* and *ndufa13* could be driven by coevolution with mitochondrial substitutions, a mechanism that has been proposed to explain the outsized role of the mitochondria in hybrid incompatibilities[15,54]. Indeed, there is an excess of derived substitutions in the *X. birchmanni* mitochondrial protein Nd6, one of the proteins that physically contacts Ndufs5 and Ndufa13 (Extended Data Fig. 7 and Extended Data Table 2; codeml branch test *P* = 0.005). Moreover, several of the substitutions observed in both mitochondrial and nuclear genes are predicted to have functional consequences (Extended Data Table 3 and Supplementary Information 1.5.1), including ones predicted to be in contact between Ndufs5, Ndufa13, Nd2 and Nd6 (Fig. 4a,b and Extended Data Fig. 6).

## Introgression of incompatibility genes

The presence of a mitonuclear incompatibility in *Xiphophorus* is especially intriguing, given previous reports that mitochondrial genomes may have introgressed between species[29]. While *X. malinche* and *X. birchmanni* are sister species based on the nuclear genome, they are mitochondrially divergent, with *X. malinche* and *Xiphophorus cortezi* grouped as sister species based on the mitochondrial phylogeny[29] (Fig. 5a,b). As we show, all *X. cortezi* mitochondria sequenced to date are nested within *X. malinche* mitochondrial diversity (Fig. 5b, Supplementary Fig. 37 and Supplementary Information 1.5.3 and 1.5.4). Simulations indicate that gene flow, rather than incomplete lineage

sorting, drove replacement of the *X. cortezi* mitochondria with the *X. malinche* sequence (*P* < 0.002 by simulation; Fig. 5c and Supplementary Information 1.5.4).

The introgression of the mitochondrial genome from *X. malinche* into *X. cortezi* raises the possibility that other complex I genes may have co-introgressed[55]. Indeed, the nucleotide sequence for *ndufs5* is identical between *X. malinche* and *X. cortezi*, and the sequence of *ndufa13* differs by a single synonymous mutation (although conservation of both genes is high throughout *Xiphophorus*; Supplementary Figs. 38 and 39). The identical amino acid sequences of the proteins suggest that hybrids between *X. cortezi* and *X. birchmanni* are likely to harbour the same mitonuclear incompatibility we observe between *X. malinche* and *X. birchmanni*, as a result of ancient introgression between *X. malinche* and *X. cortezi* (Fig. 5d and Supplementary Information 1.5.3–1.5.5).

This inference is supported by analysis of three contemporary *X. birchmanni* × *X. cortezi* hybrid populations[40] (Supplementary Fig. 40). We find that all known *X. birchmanni* × *X. cortezi* hybrid populations are fixed for the mitochondrial genome from *X. cortezi* (that originated in *X. malinche*) and show a striking depletion of *X. birchmanni* ancestry at *ndufs5* and *ndufa13* (Fig. 5e and Supplementary Fig. 41). This replicated depletion is not expected by chance (Fig. 5e and Supplementary Information 1.5.6, *P* = 0.0001) and instead indicates that selection has acted on these regions. These results suggest that the mitonuclear incompatibility observed in *X. birchmanni* × *X. malinche* is also active in hybridizing *X. birchmanni* × *X. cortezi* populations. This exciting finding shows that genes underlying hybrid incompatibilities can introgress together, transferring incompatibilities between related species.

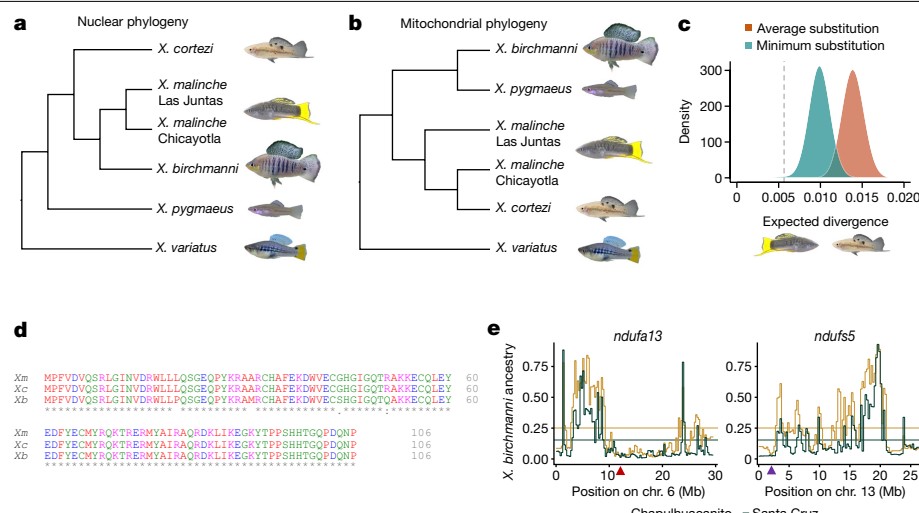

**Fig. 5 | Phylogenetic analysis and ancestry mapping suggest that genes underlying the mitonuclear incompatibility have introgressed from *X. malinche* into *X. cortezi*. a**, Nuclear phylogeny of *Xiphophorus* species, showing that *X. birchmanni* and *X. malinche* are sister species[29]. **b**, Phylogeny constructed from whole mitochondrial DNA sequences showing that *X. cortezi* mitochondria are nested within *X. malinche* mitochondrial diversity. **c**, Results of simulations modelling expected mitochondrial divergence between *X. malinche* and *X. cortezi* in a scenario with no gene flow. The first set of simulations used the average mitochondrial substitution rate observed between *Xiphophorus* species (red), and the second used the minimum mitochondrial substitution rate observed (teal). The dotted line shows observed divergence between mitochondrial haplotypes in *X. malinche* and *X. cortezi*, indicating that past mitochondrial introgression is more likely than incomplete lineage sorting. **d**, Clustal alignment of Ndufs5 sequences shows that *X. malinche* and

*X. cortezi* (*Xc*) have identical amino acid sequences at Ndufs5, hinting at possible introgression of the nuclear *ndufs5* gene, whereas *X. birchmanni* is separated from them by four substitutions. Similar patterns are observed for *ndufa13*. Colours indicate properties of the amino acid, and asterisks indicate locations where the amino acid sequences are identical. **e**, Non-mitochondrial parent ancestry is strongly depleted in natural *X. cortezi* × *X. birchmanni* hybrid populations fixed for the *X. cortezi* mitochondrial haplotype at *ndufs5* and *ndufa13*. Upward arrowheads along *x*-axes show the locations of *ndufa13* (red) and *ndufs5* (purple) on chromosomes 6 and 13, respectively, which fall in minor parent ancestry deserts in both independently formed populations, as expected for strong hybrid incompatibilities. Step curves show average *X. birchmanni* ancestry in 250-kb windows, and horizontal lines show the genome-wide average for each population.

## Discussion

Here we investigate the genetic and evolutionary forces that drive the emergence of hybrid incompatibilities. Theory predicts that hybrid incompatibilities involving multiple genes should be common[7,8], but with few exceptions[9,11–13], they remain almost uncharacterized at the genic level[8]. We have identified incompatible interactions in mitochondrial complex I that cause hybrid lethality in laboratory and wild populations. Our findings in naturally hybridizing species echo predictions from theory and studies in laboratory models[9,11–13], suggesting that protein complexes may be a critical site of hybrid breakdown.

Researchers have proposed mitonuclear interactions as hotspots for the emergence of hybrid incompatibilities, given that mitochondrial genomes often experience higher substitution rates between species[17,18,56], yet must intimately interact with nuclear proteins to perform essential cellular functions[22,23]. Our findings support this prediction, identifying incompatible interactions with both the *X. malinche* and *X. birchmanni* mitochondria. We also show that there has been exceptionally rapid evolution in both mitochondrial and interacting nuclear genes in *X. birchmanni* (Fig. 4). Whether driven by adaptation or some other mechanism, our findings support the hypothesis that the coevolution of mitochondrial and nuclear genes could drive the overrepresentation of mitonuclear interactions in hybrid incompatibilities[22,23,54]. More broadly, our results are consistent with predictions that rapidly evolving proteins are more likely to become involved in hybrid incompatibilities than their slowly evolving counterparts[4–6].

Characterizing the incompatibility across multiple scales of organization enabled us to explore the mechanisms through which it acts[57–59]. Our results suggest that in the case of the *X. malinche* mitochondria

hybrid lethality is mediated through arrested development in utero of individuals with mismatched ancestry at *ndufs5*, whereas individuals with *ndufa13* mismatch have vascular defects and typically die shortly after birth. Intriguingly, individuals with *ndufa13* mismatch that do survive are much less likely to harbour any *X. birchmanni* alleles at *ndufs5* (Fig. 2i). Together, our results indicate that a subtle three-way interaction overlays two strong pairwise mitonuclear incompatibilities at *ndufs5* and *ndufa13*. Evolutionary biologists have been fascinated by the idea that hybrid incompatibilities may commonly involve three or more genes following theoretical work by Orr[7] nearly 30 years ago, but this question has been challenging to address empirically. Our results highlight how the nuances of actual fitness landscapes may defy simplifying assumptions.

Finally, this mitonuclear incompatibility provides a new case in which the same genes are involved in incompatibilities across multiple species[30,38,60]. However, tracing the evolutionary history of the genes that underlie it adds further complexity to this phenomenon: we found that introgression has resulted in the transfer of genes underlying the incompatibility from *X. malinche* to *X. cortezi*, and evidence from *X. birchmanni* × *X. cortezi* hybrid populations indicates that the incompatibility is probably under selection in these populations as well. The possibility that hybridization could transfer incompatibilities between species has not been previously recognized, perhaps due to an underappreciation of the frequency of hybridization. The impact of past hybridization on the structure of present-day reproductive barriers between species is an exciting area for future inquiry.

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

## Methods

### Biological materials

Wild parental and hybrid individuals used in this study were collected from natural populations in Hidalgo, Mexico (permit no. PPF/DGOPA-002/19). Artificial $F_1$ and $F_2$ hybrids were generated using mesocosm tanks as described previously[39]. Caudal fin clips were used as the source for all DNA isolation and for flow cytometry, and liver tissue for RNA-seq, respirometry, and proteomic assays were collected following Stanford Administrative Panel on Laboratory Animal Care (APLAC) protocol no. 33071.

### Genotyping and local ancestry calling

Genomic DNA was extracted from fin clips and individually barcoded tagmentation-based libraries were generated (Supplementary Information 1.1.3). Hybrids were genotyped with low-coverage whole-genome sequencing followed by local ancestry inference across the 24 *Xiphophorus* chromosomes and the mitochondrial genome using the *ancestryinfer* pipeline[38,39,42,61] (Supplementary Information 1.1.3 and 1.1.4). We converted posterior probabilities for each ancestry state to hard calls for downstream analysis, using a posterior probability threshold of 0.9, and analysed ancestry variation across the genome.

### QTL and admixture mapping

The regions interacting with the mitochondrial genome were first identified based on analysis of segregation distortion in 943 $F_2$ hybrids generated from $F_1$ crosses between *X. malinche* females and *X. birchmanni* males (Supplementary Information 1.1.1 and Langdon et al.[40]). Since all hybrids in this artificial cross harboured the *X. malinche* mitochondria, we scanned for regions of exceptionally high *X. malinche* ancestry along the genome (>60% *X. malinche* ancestry), identifying one such region on chromosome 13 and one on chromosome 6 (Fig. 1; see also ref. 40). Evidence for interactions between these regions and the mitochondrial genome were confirmed using admixture mapping in two hybrid populations that segregated for the mitochondrial haplotype of both species (Supplementary Information 1.1.2): the Calnali Low hybrid population ($n = 359$) and the Chahuaco falls hybrid population ($n = 244$). In brief, we used a partial correlation analysis to identify regions of the genome strongly associated with mitochondrial ancestry, after regressing out genome-wide ancestry to account for covariance in ancestry due to population structure (see ref. 36 and Supplementary Information 1.1.5 and 1.1.9). Significance thresholds were determined using simulations (Supplementary Information 1.1.5; see also Supplementary Information 1.1.6).

### Estimating selection on the incompatibility

We used an ABC approach to estimate the strength of selection against the incompatible interaction between the *X. malinche* mitochondrial haplotype and *X. birchmanni* ancestry at the two nuclear genes involved in the hybrid incompatibility: *ndufs5* and *ndufa13* (Supplementary Information 1.2.2). For these simulations, we asked what selection coefficients (0–1) and dominance coefficients (0–1) could generate the observed deviations from the expectation of 50:50 *X. birchmanni*–*X. malinche* ancestry in $F_2$ hybrids at *ndufs5* and *ndufa13* after two generations of selection. We performed 500,000 simulations for each interaction and accepted or rejected simulations based on comparisons to the real data using a 5% tolerance threshold (Supplementary Information 1.2.2). We also evaluated evidence for incompatible interactions with the *X. birchmanni* mitochondrial haplotype (Supplementary Information 1.2.1–1.2.4).

### Embryo staging and genotyping

To pinpoint when in development the incompatibility between the *X. malinche* mitochondria and *X. birchmanni* nuclear genotypes causes lethality, we collected a dataset on the developmental stages of embryos with different genotype combinations. Whole ovaries were removed from pregnant females and embryos were individually dissected. Each embryo was assigned a developmental stage ranging from 1–11 based on established protocols for poeciliid embryos[44]. Unfertilized eggs were excluded from analysis. Following staging, individual embryos ($n = 296$) were genotyped as described above and in Supplementary Information 1.3.1. We tested for significant differences in developmental stage between siblings with compatible and incompatible genotype combinations using a two-sided two-sample *t*-test (Supplementary Information 1.3.1) and examined differences in ancestry between large groups of siblings that varied in their developmental stages (Supplementary Information 1.1.7). We also collected data on embryonic stage and variability between siblings in embryonic stage from both pure parental species (Supplementary Information 1.3.1). We used a different approach to pinpoint the timing of *ndufa13* lethality given that it appeared to act postnatally (Supplementary Information 1.3.3).

### Embryo respirometry and morphometrics

To study the mechanisms of *ndufs5*- and *ndufa13*-driven lethality, we performed oxygen consumption measurements on $F_2$ embryos in a Loligo plate respirometer (Supplementary Information 1.3.4). Embryos were dissected from mothers and transferred to wells of a 24-well plate, where their oxygen consumption was measured over 60 min. The measurement was then repeated in media dosed with 5 µM rotenone to test sensitivity to complex I inhibition, after which the embryos were video recorded and photographed for morphometrics in ImageJ. We used linear models to test the effect of *ndufs5* genotype, *ndufa13* genotype, and individual standard length on a number of variables, controlling for batch effects (Supplementary Information 1.3.4 and 1.3.5).

### Mitochondrial respirometry

To further evaluate mitochondrial function in individuals heterozygous for the mitonuclear incompatibility (Supplementary Information 1.3.6 and 1.3.7), we conducted respirometry assays on *X. birchmanni*, *X. malinche*, and hybrid individuals that had the *X. malinche* mitochondria and were heterozygous for the nuclear components of the hybrid incompatibility ($n = 7$ of each genotype). Mitochondria were isolated from whole liver tissue and mitochondrial respiration was quantified using the Oroboros O2K respirometry system[62] (Supplementary Fig. 23). A step-by-step description of this protocol and methods used to calculate respiratory flux control factors is outlined in Supplementary Information 1.3.8. We complemented the results of these respirometry experiments with measures of mitochondrial membrane potential using a flow cytometry-based approach (Supplementary Information 1.3.9).

### Parallel reaction monitoring proteomics

For parallel reaction monitoring (PRM) with mass spectrometry, we used a similar approach to that used for respirometry to isolate whole mitochondria from five $F_1$ hybrids (which harboured *X. malinche* mitochondria). This approach is described in detail in Supplementary Information 1.4.1. In brief, we designed heavy isotope-labelled peptides to distinguish between the *X. birchmanni* and *X. malinche* copies of Ndufs5 and Ndufa13, facilitating quantification of the peptides of interest in the mitochondrial proteome (Supplementary Information 1.4.2). Mitochondrial isolates were prepared for mass spectrometry and combined with heavy isotope-labelled peptides in known quantities (see Supplementary Information 1.4.3), then submitted to Orbitrap mass spectrometry with separation with ultra performance liquid chromatography and PRM for ion selection. The protocol for mass spectrometry and PRM is described in detail in Supplementary Information 1.4.4.

To analyse the results, the focal peptide's spectral peak was identified based on the peak of the heavy isotope-labelled spike-in peptide. We focused analysis on the Ndufs5 peptide WLL[L/P]QSGEQPYK, since other endogenous peptides were below the expected sensitivity limits

of our PRM protocol (Supplementary Information 1.4.5). We normalized the intensity of the Ndufs5 peptide based on the known spike-in quantity, and quantified the proportion of Ndufs5 in each $F_1$ individual derived from *X. malinche* versus *X. birchmanni* (Supplementary Information 1.4.5). We tested whether these ratios significantly deviated from the 50:50 expectation for $F_1$ hybrids using a two-sided one-sample *t*-test.

### Complex I protein modelling

Mapping results allowed us to identify *ndufs5* and *ndufa13* as *X. birchmanni* genes that interact negatively with *X. malinche* mitochondrial genes. We used a protein modelling-based approach with RaptorX (http://raptorx.uchicago.edu) to identify the mitochondrial genes most likely to interact with *ndufs5* and *ndufa13* (see Supplementary Information 1.4.6). Using the mouse cryo-EM structure (Protein Data Bank (PDB) ID 6G2J) of complex I, we identified proteins in contact with Ndufs5 and Ndufa13, which included several mitochondrial (Nd2, Nd3, Nd4l and Nd6) and nuclear (Ndufa1, Ndufa8, Ndufb5 and Ndufc2) proteins. We then used RaptorX to predict structures for both the *X. birchmanni* and *X. malinche* versions of the proteins. In addition, we evaluated the robustness of these predictions to choice of cryo-EM template; see Supplementary Information 1.4.6.

### Analysis of evolutionary rates

Comparison of predicted protein sequences encoded by *ndufs5, ndufa13* and mitochondrial genes of interest (*nd2* and *nd6*) revealed a large number of substitutions between *X. birchmanni* and *X. malinche*. We calculated $d_N/d_S$ between *X. birchmanni* and *X. malinche* for all annotated protein coding genes throughout the genome and found that both *ndufs5* and *ndufa13* have rapid protein evolution (Fig. 4d and Supplementary Information 1.5.1). Examining these mutations in a phylogenetic context revealed that many substitutions in *ndufs5, nudfa13* and *nd6* were derived in *X. birchmanni*. We tested for significant differences in evolutionary rates on the *X. birchmanni* lineage and for predicted functional impacts of these substitutions; these analyses are described in Supplementary Information 1.5.1.

### Tests for ancient introgression

Previous work had indicated that the mitochondrial phylogeny in *Xiphophorus* is discordant with the whole-genome species tree[29]. Specifically, although *X. birchmanni* and *X. malinche* are sister species based on the nuclear genome, *X. malinche* and *X. cortezi* are sister species based on the mitochondrial genome. We used a combination of PacBio amplicon sequencing of 10 individuals (2 or more per species, Supplementary Information 1.5.3) and newly available whole-genome resequencing data to confirm this result and polarize the direction of the discordance by constructing maximum likelihood mitochondrial phylogenies with the program RAxML[63]. We performed similar phylogenetic analyses of the nuclear genes that interact with the *X. malinche* mitochondria (*ndufs5* and *ndufa13*; Supplementary Information 1.5.3). Combined with phylogenetic results, simulation results suggest that gene flow from *X. malinche* into *X. cortezi* is the most likely cause of the discordance we observe between the mitochondrial and nuclear phylogenies (Supplementary Information 1.5.3 and 1.5.4). Since *X. malinche* and *X. cortezi* are not currently sympatric, this suggests ancient gene flow between them (Supplementary Information 1.5.5).

### *X. birchmanni* × *X. cortezi* hybridization

To investigate the possibility that hybrids between *X. birchmanni* and *X. cortezi* share the same mitonuclear incompatibility as observed in hybrids between *X. birchmanni* and *X. malinche* (Supplementary Information 1.5.6), we took advantage of genomic data from recently discovered hybrid populations between *X. birchmanni* and *X. cortezi*[64]. Using data from three different *X. birchmanni* × *X. cortezi* populations and a permutation-based approach, we tested whether ancestry at *ndufs5* and *ndufa13* showed lower mismatch with mitochondrial ancestry than expected given the genome-wide ancestry distribution. This analysis is described in detail in Supplementary Information 1.5.6.

### Animal care and use

All methods were performed in compliance with Stanford APLAC protocol no. 33071.

### Reporting summary

Further information on research design is available in the Nature Portfolio Reporting Summary linked to this article.

## Data availability

Raw sequencing reads used in this project are available under NCBI SRA Bioprojects PRJNA744894, PRJNA746324, PRJNA610049, PRJNA361133 and PRJNA745218. Mass spectrometry data are available on PRIDE with identifier PXD046217, and other datasets necessary to recreate the results of the publication are available on Dryad (https://doi.org/10.5061/dryad.j3tx95xmx). Templates for complex I protein structural modelling were accessed from the Protein Data Bank (PDB) with accession numbers 6G2J, 6G72, 5LDW, 5LNK and 5XTC.

## Code availability

All custom scripts used to generate results are available on Github at https://github.com/Schumerlab/mitonuc_DMI and https://github.com/Schumerlab/Lab_shared_scripts.

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

**Acknowledgements** The authors thank P. Andolfatto, S. Aguillon, Y. Brandvain, J. Coughlan, H. Fraser, Y. Haba, N. Phadnis, M. Przeworski, K. Thompson and members of the Schumer laboratory for helpful discussion and/or feedback on earlier versions of this manuscript, and A. Pollock for help performing rotenone trials. We thank the Federal Government of Mexico for permission to collect fish. Stanford University and the Stanford Research Computing Center provided computational support for this project. We thank the Vincent Coates Foundation Mass Spectrometry Laboratory, Stanford University Mass Spectrometry (RRID:SCR_017801) for technical and experimental support. This work was supported by a Knight–Hennessy Scholars fellowship and NSF GRFP 2019273798 to B.M.M., a CEHG fellowship and NSF PRFB (2010950) to Q.K.L., NIH P30 CA124435 in utilizing the Stanford Cancer Institute Proteomics/Mass Spectrometry Shared Resource, NIH grant 1R35GM142836 to J.C.H., and a Hanna H. Gray fellowship, Sloan Fellowship, and NIH grant 1R35GM133774 to M. Schumer.

**Author contributions** B.M.M., D.L.P. and M. Schumer designed the project. B.M.M., C.Y.P., E.N.K.I., S.M.B., A.E.D., R.A.R.-S., A.M., J.J.B., K.M.K., F.L., R.M., K.S., O.H.-P., J.C.H., A.M. and M. Schumer collected data. B.M.M., C.Y.P., Q.K.L., F.L., J.C.H., R.A.R.-S., A.M. and M. Schumer performed analyses. D.L.P., T.R.G., R.D.L., C.G., R.C.-D., J.F. and M. Schartl provided expertise and technical support.

**Competing interests** The authors declare no competing interests.

**Additional information**
**Correspondence and requests for materials** should be addressed to Benjamin M. Moran or Molly Schumer.

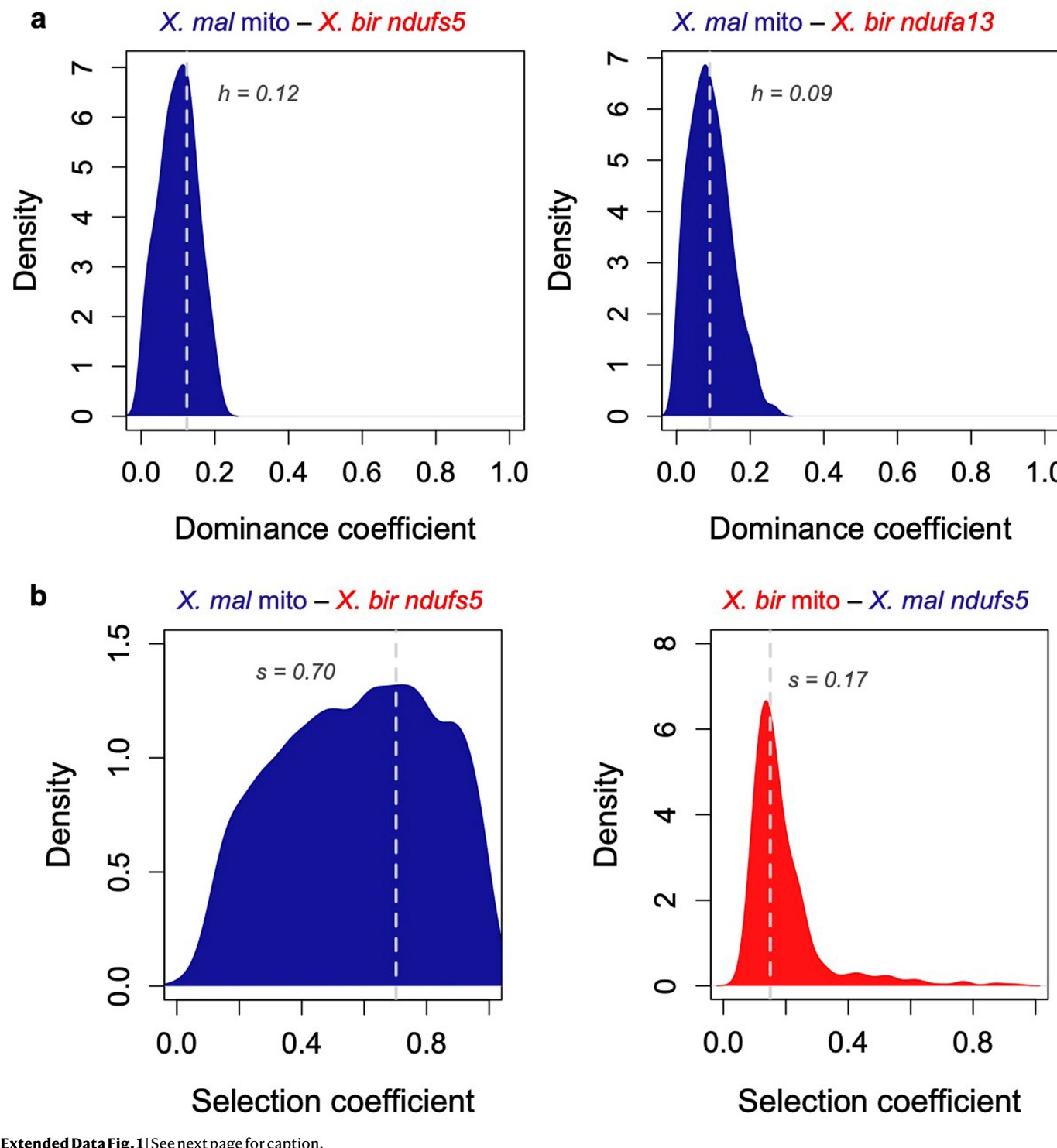

**Extended Data Fig. 1** | See next page for caption.

**Extended Data Fig. 1 | ABC inference of additional selection parameters.**
(**a**) Posterior distributions of dominance coefficients from approximate Bayesian computation (ABC) simulations fitting observed data in $F_2$ hybrids. Left distribution shows the results of simulations modeling the *X. malinche* mitochondria – *X. birchmanni ndufs5* component of the hybrid incompatibility, and right distribution shows the results of simulations modeling the *X. malinche* mitochondria – *X. birchmanni ndufa13* component of the hybrid incompatibility. Each posterior distribution shows the accepted dominance coefficients in 500 simulations and the gray dashed line indicates the maximum a posteriori estimate for the dominance coefficients of interactions involving *ndufs5* and *ndufa13* (**b**) Posterior distributions of selection coefficients from ABC simulations fitting observed data in the Calnali Low hybrid population. Left distribution shows the results of simulations modeling the *X. malinche* mitochondria – *X. birchmanni ndufs5* incompatibility and right distribution shows the results of simulations modeling the reverse *X. birchmanni* mitochondria – *X. malinche ndufs5* incompatibility. In the main text we focus on inferred selection coefficients from $F_2$ hybrids for the incompatibility involving the *X. malinche* mitochondria (see Fig. 1) but present results from fitting the Calnali Low population data here and in Supplementary Information 1.2.2. Distribution shows the accepted selection coefficients in 500 simulations and the gray dashed line indicates the maximum a posteriori estimate. Note that we recover a much broader distribution and a lower maximum a posteriori estimate of the selection coefficient for the *X. malinche* mitochondria – *X. birchmanni ndufs5* incompatibility here compared to $F_2$ simulations (Fig. 1). We interpret this result to be driven by the fact that simulations fitting the Calnali Low population data modeled >20 generations of admixture. Thus, a range of selection coefficients are consistent with the observation that no individuals have *X. malinche* mtDNA and homozygous *X. birchmanni* ancestry at *ndufs5* in this population.

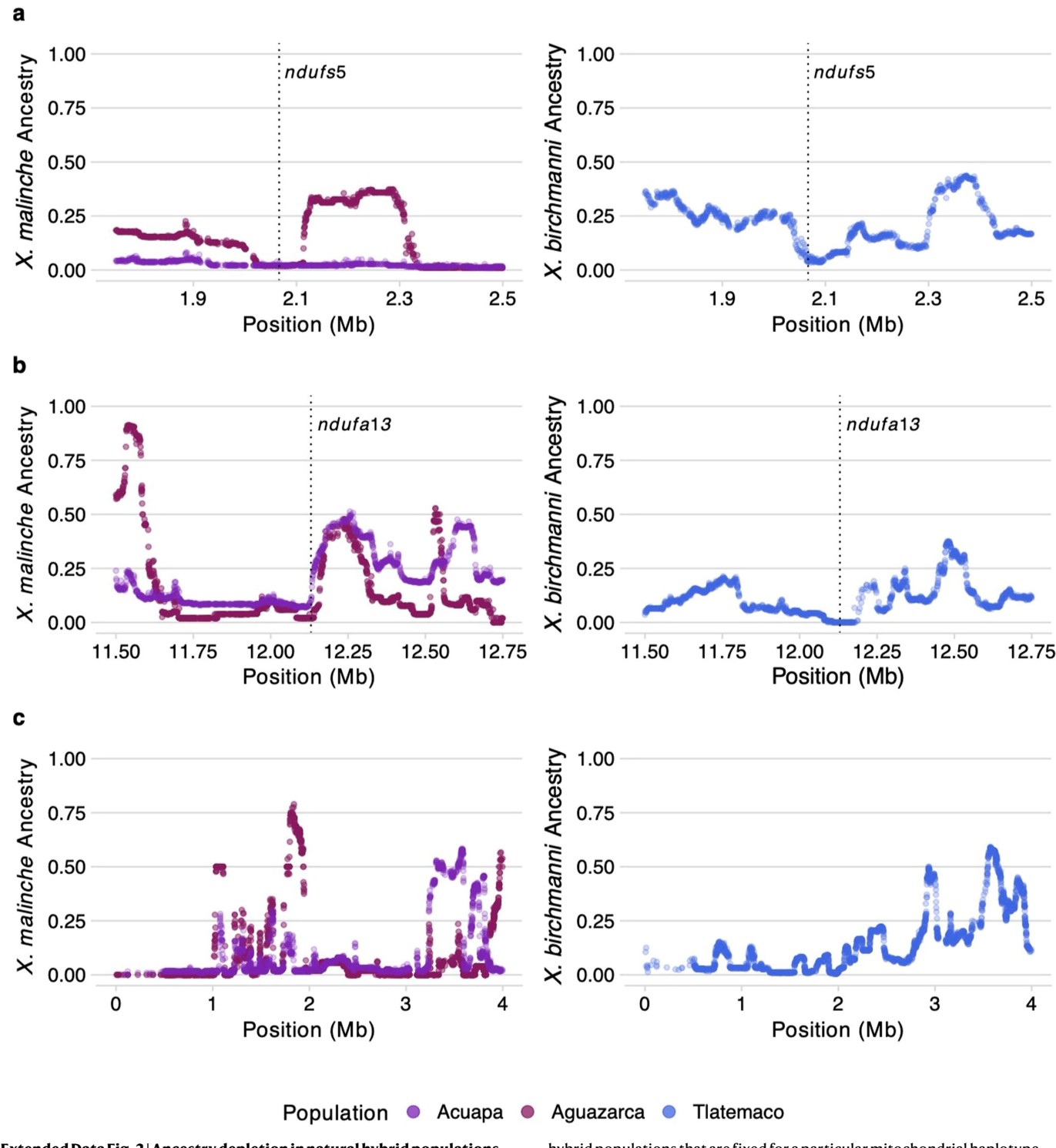

**Extended Data Fig. 2 | Ancestry depletion in natural hybrid populations.** Average non-mitochondrial parent ancestry on (**a**) chromosome 13, (**b**) chromosome 6, and (**c**) chromosome 15 in natural hybrid populations fixed for *X. birchmanni* (Aguazarca, Acuapa, left column) or *X. malinche* (Tlatemaco, right column) mitochondrial haplotypes. Vertical dashed lines represent the position of *ndufs5* and *ndufa13* within the admixture mapping regions. (**A**) Non-mitochondrial parent ancestry is depleted around *ndufs5* in all natural

hybrid populations that are fixed for a particular mitochondrial haplotype. This pattern is strongly suggestive of a history of selection on this region in natural hybrid populations. In (**B**), *ndufa13* is fixed for *X. malinche* ancestry only in the population with *X. malinche* mitochondria (Tlatemaco), mirroring the architecture of the genetic interaction between this gene and the mitochondria. Specifically, interactions with *ndufa13* are only expected to be under selection in combination with the *X. malinche* mitochondrial haplotype (Fig. 1c).

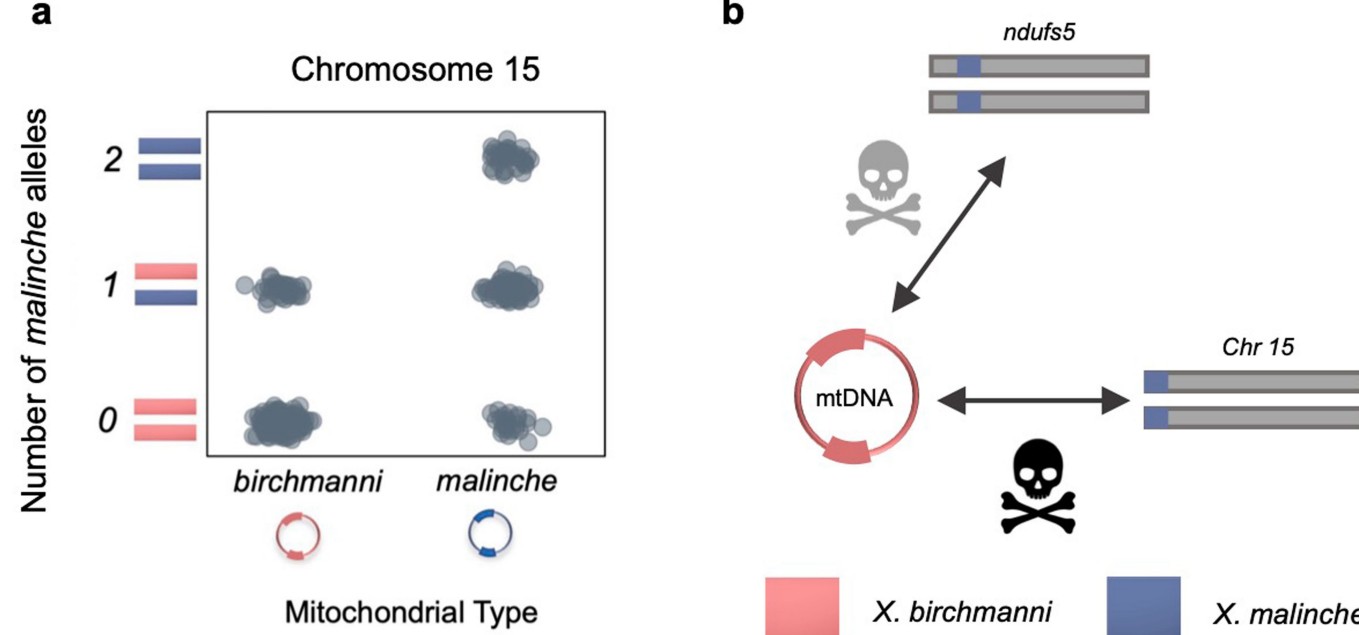

**a**

**Chromosome 15**

**b**

**Extended Data Fig. 3 | Chromosome 15 incompatibility. (a)** Observed genotype frequencies at the peak associated marker (3.37 Mb) on chromosome 15 in the admixture mapping population. **(b)** Schematic of identified interactions with the *X. birchmanni* mitochondrial genome from our mapping data and strength of selection underlying each interaction in hybrids (gray skull – moderate, black skull – near lethal). We discuss interactions with the *X. birchmanni* mitochondria in more detail in Supplementary Information 1.2.1–1.2.2.

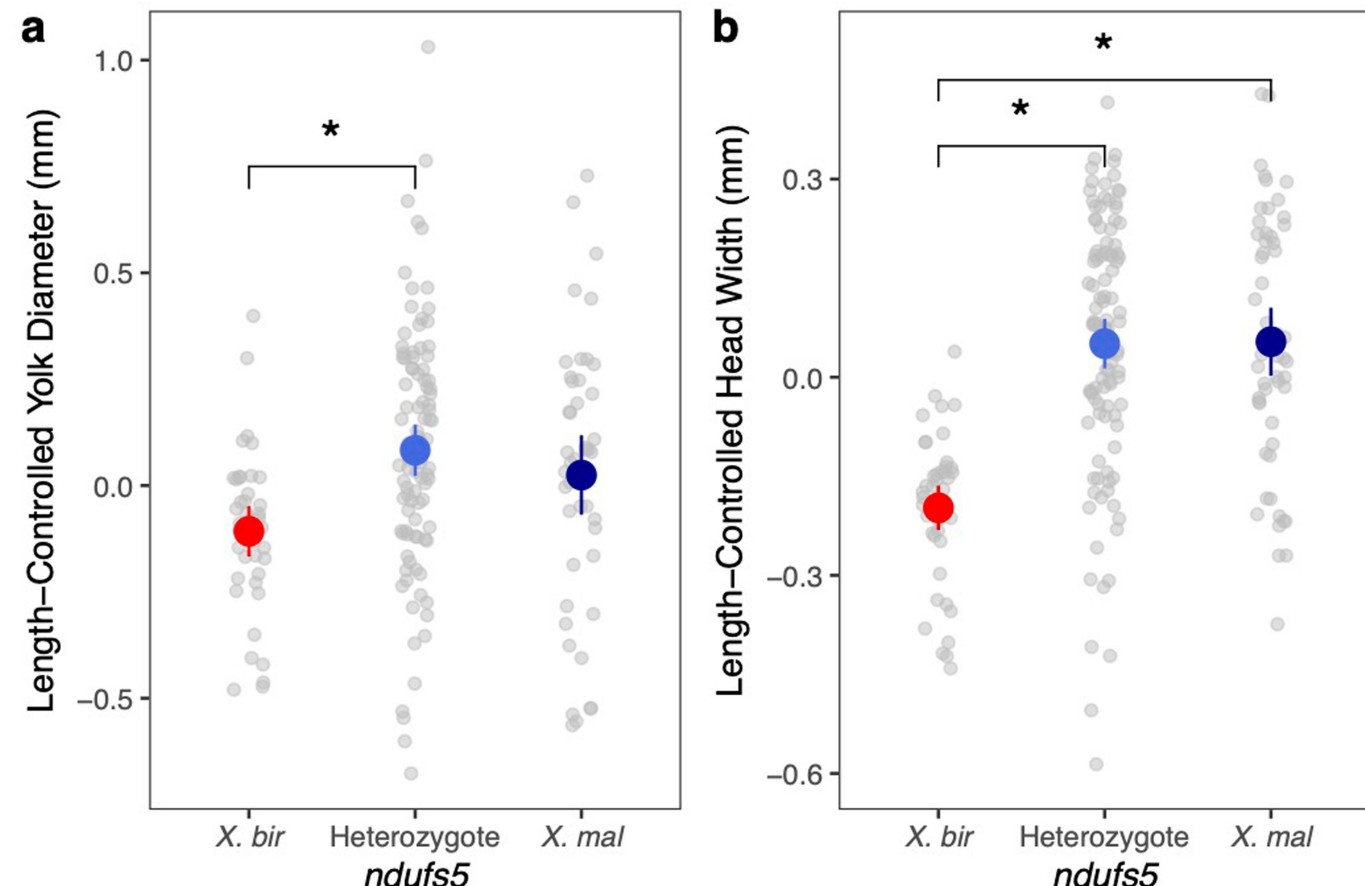

**Extended Data Fig. 4 | Additional F$_2$ embryo morphometrics by *ndufs5* genotype.** Relationship between (**a**) yolk diameter and *ndufs5* genotype (n = 41 *birchmanni*, 96 heterozygotes, and 46 *malinche*) and (**b**) head width and *ndufs5* genotype (n = 44 *birchmanni*, 108 heterozygotes, and 55 *malinche*) in F$_2$ hybrid embryos. To control for the strong effect of length, the residuals of each variable after accounting for body length are plotted. Grey points represent individual measurements, colored points with vertical lines represent group mean ± 2 SE, and brackets with asterisks denote significant differences from Tukey's HSD test.

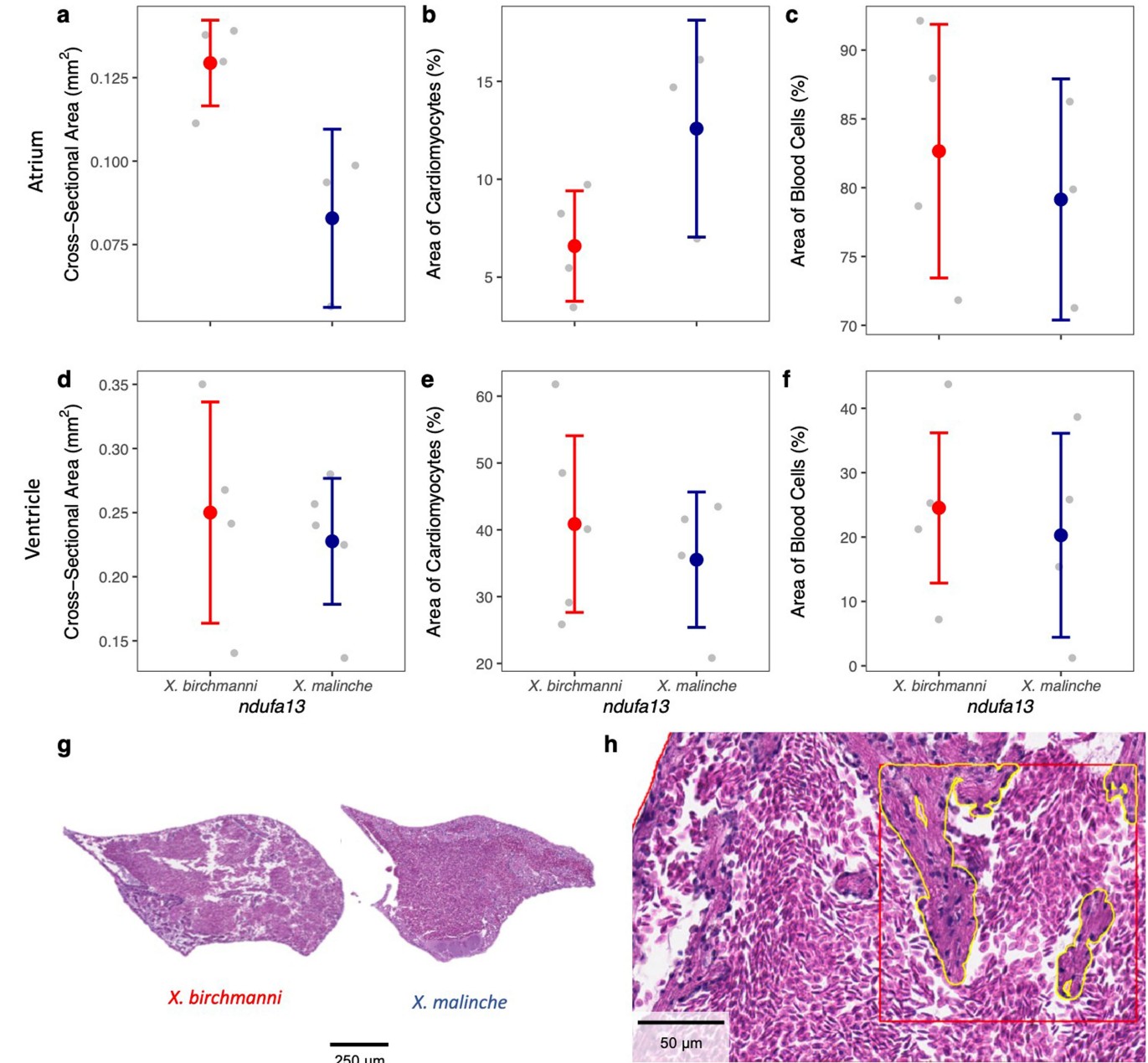

**Extended Data Fig. 5 | Juvenile F$_2$ heart morphology by *ndufa13* genotype.**
In all panels, colored points and bars show the mean ± 2 SE, and gray points show individual data. Individuals that were homozygous *X. birchmanni* at *ndufa13* (the incompatible genotype) and homozygous *X. malinche* at *ndufa13* (the compatible genotype) were raised in common laboratory conditions and sampling occurred at approximately 5 months of age (n = 5 juveniles per genotype). Measurements were taken from the sagittal section of largest cross-sectional area for the atrium (**a**-**c**) and ventricle (**d**-**f**), and area of occupancy for each cell class was calculated from the average of three quadrats (Supplementary Information 1.3.3). (**g**) Representative images of atria from incompatible (*X. birchmanni)* and compatible (*X. malinche*) individuals. Images are from the slide with maximum cross-sectional atrial area from each individual. (**h**) Example of histology analysis process in a representative atrium. Red square indicates randomly placed quadrat in which occupancy was calculated, and yellow borders represent areas which were manually annotated as cardiomyocytes. Note that the epicardium is visible at top left. All fish were raised and processed as one experimental group, with no independent attempts to test reproducibility at the experimental level.

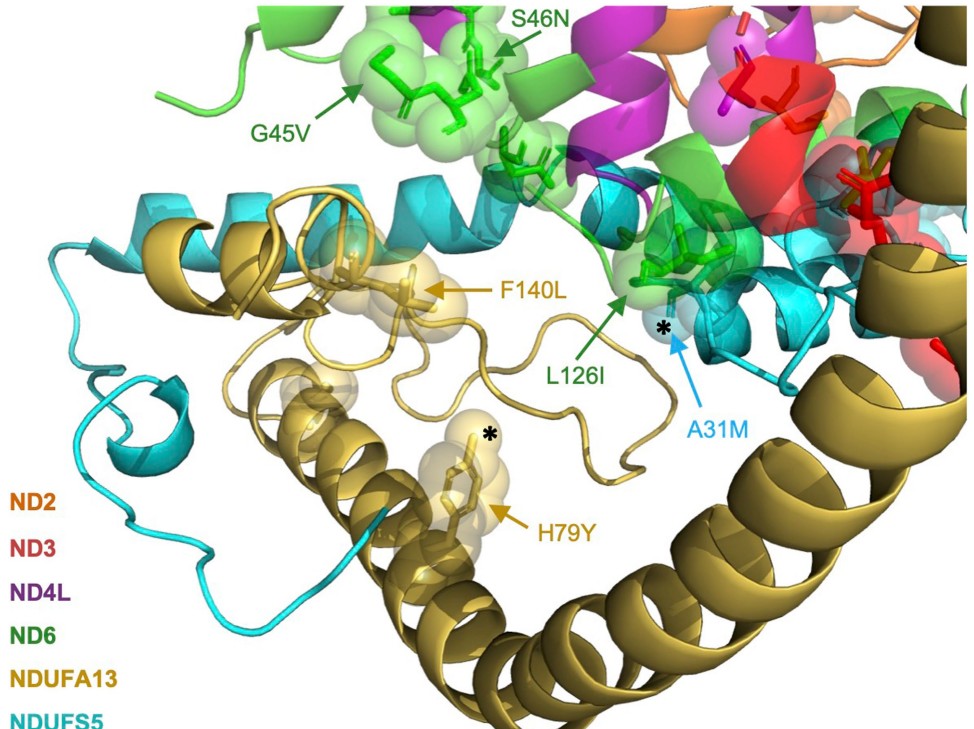

**Extended Data Fig. 6 | Interface between *ndufa13, ndufs5,* and *nd6* in RaptorX model.** Arrows highlight substitutions in *ndufa13, ndufs5* and mitochondrial proteins in proximity in the model, with alphanumeric codes denoting the *X. malinche* amino acid, the residue number, and the *X. birchmanni* amino acid, from left to right. Colors distinguish proteins, as denoted by colored protein names at left. Asterisks denote residues with substitutions in *X. birchmanni* computationally predicted to affect protein function (Extended Data Table 3).

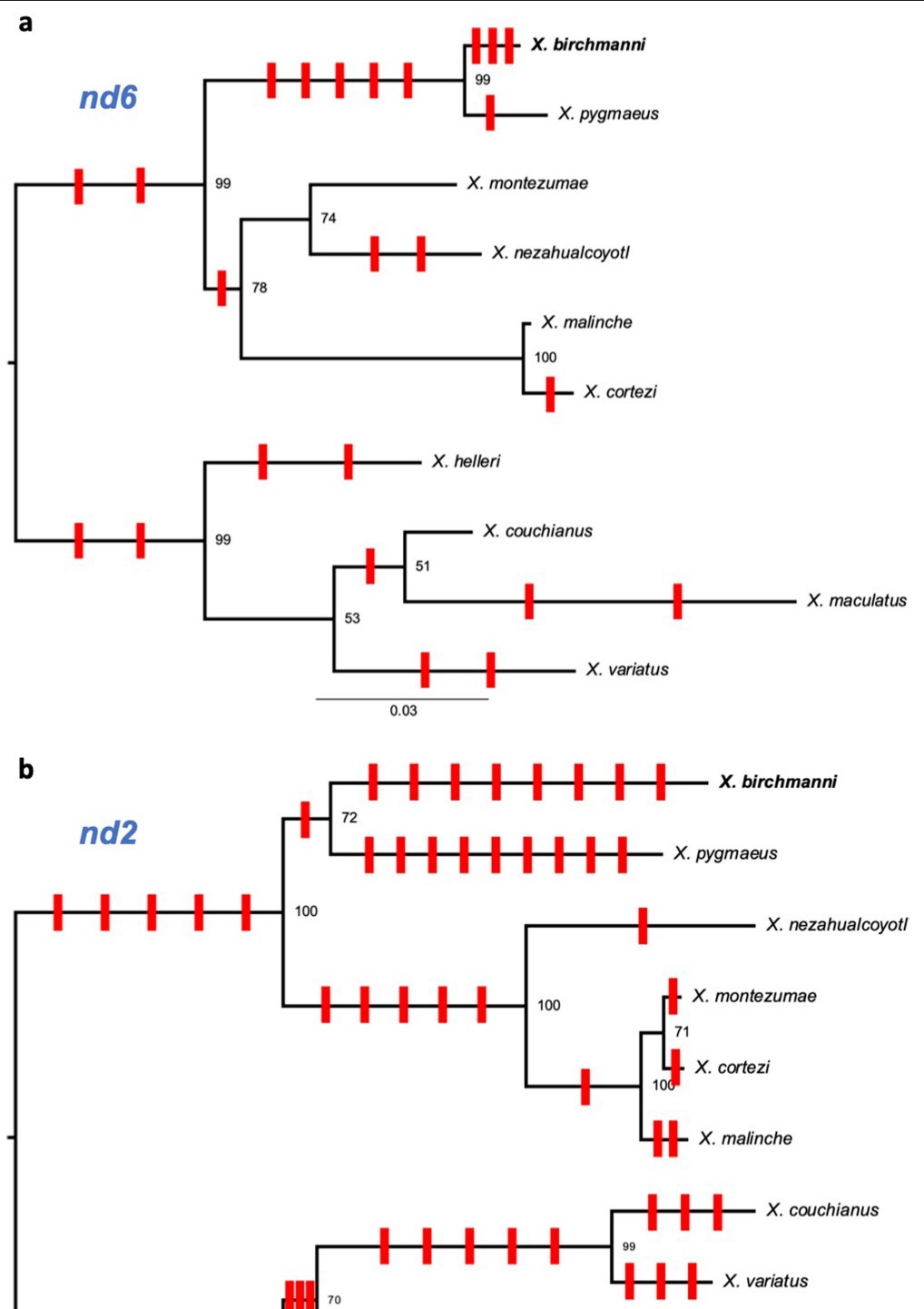

**Extended Data Fig. 7 | Complex I mtDNA gene trees.** Gene trees were generated with RAxML for (**A**) *nd6* and (**B**) *nd2*, highlighting an excess of substitutions along the *X. birchmanni* branch in *nd6*. Scale bar represents number of nucleotide substitutions per site, and derived non-synonymous substitutions are indicated by red ticks along the phylogeny. Note that spacing between ticks is arbitrary and substitutions were placed on branches to maximize parsimony. In some cases, the distribution of substitutions cannot be explained by a single event, in such cases we illustrate the minimum number of events leading to observed distribution of the substitution.

**Extended Data Table 1 | Sensitivity of inter-residue distances to modelling approach**

### NDUFS5 M31

| Model | E119 | A120 | D122 | I126 |
|---|---|---|---|---|
| RaptorX | 23.79 | 20.44 | 13.76 | 3.48 |
| MODELLER - 5LNK | 8.93 ± 1.23 | 9.53 ± 2.66 | 12.42 ± 1.95 | 14.61 ± 1.19 |
| 5LDW | 10.22 ± 1.30 | 9.95 ± 1.00 | 11.75 ± 1.43 | 15.21 ± 1.55 |
| 5XTC | 8.38 ± 1.96 | 10.64 ± 1.60 | 11.60 ± 1.13 | 14.80 ± 0.48 |
| 6G2J | 10.68 ± 3.24 | 10.32 ± 1.48 | 8.71 ± 3.37 | 11.23 ± 1.94 |

### NDUFA13 Y79

| Model | E119 | A120 | D122 | I126 |
|---|---|---|---|---|
| RaptorX | 27.68 | 23.86 | 16.81 | 13.08 |
| MODELLER - 5LNK | 7.03 ±1.97 | 6.42 ± 1.79 | 8.48 ± 2.01 | 16.25 ± 1.64 |
| 5LDW | 10.88 ± 1.33 | 7.61 ± 1.72 | 5.29 ± 0.75 | 12.69 ± 1.28 |
| 5XTC | 9.30 ± 1.29 | 7.81 ± 0.82 | 5.86 ± 0.36 | 15.08 ± 0.58 |
| 6G2J | 6.92 ± 1.77 | 7.58 ± 0.95 | 6.28 ± 2.64 | 13.44 ± 0.45 |

### NDUFA13 L140

| Model | E119 | A120 | D122 | I126 |
|---|---|---|---|---|
| RaptorX | 28.51 | 25.70 | 19.26 | 16.95 |
| MODELLER - 5LNK | 12.58 ± 2.42 | 11.94 ± 2.74 | 11.14 ± 2.28 | 13.08 ± 0.55 |
| 5LDW | 13.73 ± 1.84 | 13.73 ± 2.07 | 10.94 ± 1.95 | 13.63 ± 0.63 |
| 5XTC | 14.62 ± 1.48 | 11.06 ± 1.38 | 10.29 ± 1.10 | 12.26 ± 0.24 |
| 6G2J | 13.70 ± 2.09 | 11.36 ± 1.00 | 14.6 ± 1.43 | 12.97 ± 1.08 |

Effects of structural modelling program (RaptorX or MODELLER) and template on calculated distances between interspecific substitutions in *X. birchmanni* nuclear genes and four nearby substitutions in *nd6*. Mean±SD distance is listed across five replicate MODELLER runs for four different templates, each identified by their PDB ID. RaptorX used a single template for each protein and was run without replication, as described in Supplementary Information 1.4.6. Distance was calculated between alpha carbons except in the case of *ndufa13* tyrosine at position 79, where the side chain's OH group was chosen as a landmark due to its large size and stable orientation within an alpha helix.

**Extended Data Table 2 | CodeML results for Complex I genes**

| Gene | dN/dS Estimate | | | P-value | |
| | Variable Rate | | Constant Rate | Variable vs. Constant Rate | X. birchmanni Free vs. = 1 |
| | X. birchmanni | Others | All branches | | |
|---|---|---|---|---|---|
| ndufs5 | 999 | 0.08 | 0.26 | 0.005 | 0.186 |
| ndufa13 | 999 | 0.04 | 0.17 | 0.002 | 0.157 |
| nd1 | 0.10 | 0.07 | 0.07 | 0.39 | 4.064e-8 |
| nd2 | 0.11 | 0.10 | 0.10 | 0.91 | 3.375e-7 |
| nd3 | 0.24 | 0.11 | 0.12 | 0.22 | 0.021 |
| nd4 | 0.18 | 0.08 | 0.09 | 0.031 | 3.227e-8 |
| nd4l | 0.13 | 0.12 | 0.12 | 0.95 | 0.020 |
| nd5 | 0.12 | 0.08 | 0.09 | 0.35 | 1.022e-8 |
| nd6 | 10.5 | 0.09 | 0.10 | 0.005 | 0.404 |

The variable vs. constant rate P-value listed is from a two-sided likelihood ratio test comparing a model with a constant rate to an alternative model of a distinct substitution rate on the X. birchmanni branch. The P-value in rightmost column is from a two-sided likelihood ratio test of two models: one where dN/dS on the X. birchmanni branch is allowed to vary versus a null model the X. birchmanni branch dN/dS is fixed at 1.

**Extended Data Table 3 | SIFT analyses of Complex I genes**

| Gene | Substitutions predicted: | |
| | Tolerated | Not Tolerated |
| --- | --- | --- |
| *ndufs5* | L20P | A31M |
| | G44S | R51Q |
| *ndufa13* | H79Y | N90S |
| | | F140L |
| *nd2* | Y27F | N148D |
| | G83E | |
| | T89M | |
| | I98T | |
| | I191L | |
| | V243I | |
| | T309A | |
| | M311T | |
| | N317K | |
| | R319H | |
| | T331M | |
| | F335L | |
| | A338I | |
| *nd3* | I21V | L64P |
| | D80N | |
| | A83T | |
| | H86Y | |
| | A90I | |
| | V104M | |
| *nd4l* | T57A | M37T |
| | N94S | |
| *nd6* | V7I | S52P |
| | G45V | |
| | S46N | |
| | G119E | |
| | M120A | |
| | E122D | |
| | L126I | |
| | L135V | |
| | G147A | |

SIFT tests for tolerated substitutions were run for all amino acid substitutions that differ between *X. birchmanni* and *X. malinche* at *ndufs5*, *ndufa13* and the mt-DNA encoded proteins *ndufs5* contacts. All substitutions listed in both proteins are inferred to be derived in *X. birchmanni*. Substitutions are listed with the single letter amino acid code for the *X. malinche* allele first, followed by the *X. birchmanni* allele, separated by the residue number.

# Reporting Summary

## Statistics

For all statistical analyses, confirm that the following items are present in the figure legend, table legend, main text, or Methods section.

| n/a | Confirmed | |
|---|---|---|
| ☐ | ☒ | The exact sample size (*n*) for each experimental group/condition, given as a discrete number and unit of measurement |
| ☐ | ☒ | A statement on whether measurements were taken from distinct samples or whether the same sample was measured repeatedly |
| ☐ | ☒ | The statistical test(s) used AND whether they are one- or two-sided<br>*Only common tests should be described solely by name; describe more complex techniques in the Methods section.* |
| ☐ | ☒ | A description of all covariates tested |
| ☐ | ☒ | A description of any assumptions or corrections, such as tests of normality and adjustment for multiple comparisons |
| ☐ | ☒ | A full description of the statistical parameters including central tendency (e.g. means) or other basic estimates (e.g. regression coefficient) AND variation (e.g. standard deviation) or associated estimates of uncertainty (e.g. confidence intervals) |
| ☐ | ☒ | For null hypothesis testing, the test statistic (e.g. *F*, *t*, *r*) with confidence intervals, effect sizes, degrees of freedom and *P* value noted<br>*Give P values as exact values whenever suitable.* |
| ☐ | ☒ | For Bayesian analysis, information on the choice of priors and Markov chain Monte Carlo settings |
| ☐ | ☒ | For hierarchical and complex designs, identification of the appropriate level for tests and full reporting of outcomes |
| ☒ | ☐ | Estimates of effect sizes (e.g. Cohen's *d*, Pearson's *r*), indicating how they were calculated |

*Our web collection on statistics for biologists contains articles on many of the points above.*

## Software and code

Policy information about availability of computer code

| Data collection | Simulation data were collected using SLiM v3.3.1, SELAM v0.8 (https://github.com/russcd/SELAM, commit d89bcd244d7e007e457933ae0a6bc2ba0b94db0f) and custom R code (R v4.3.0) to be made available at https://github.com/Schumerlab/mitonuc_DMI. Mass spectrometry data were collected using Skyline v19.1.0.193. |
|---|---|
| Data analysis | Raw genotype data were converted to local ancestry probabilities using ancestryinfer (https://github.com/Schumerlab/ancestryinfer) and custom scripts available at https://github.com/Schumerlab/Lab_shared_scripts. Bioinformatic analyses were carried out using BLASTn v2.11.0, samtools v1.8, bcftools v1.8, bwa v0.7.17, Picard Tool v1.118, GATK v3.4-46-gbc02625, PAML v4.8a, SIFT web server (https://sift.bii.a-star.edu.sg/www/SIFT_aligned_seqs_submit.html), Clustal Omega v1.2.4, RAxML v8.2.12, pbAA v0.1.2, and the R packages DESeq2 (v1.42.0) and ppcor (v1.0). Protein structural modelling was performed with the RaptorX web server (http://raptorx.uchicago.edu). Raw embryo respirometry data was processed in Loligo Microresp v1.6, and morphometrics images were processed in ImageJ v1.52q. Heart histology slides were measured in QuPath v0.4.3 |

For manuscripts utilizing custom algorithms or software that are central to the research but not yet described in published literature, software must be made available to editors and reviewers. We strongly encourage code deposition in a community repository (e.g. GitHub). See the Nature Portfolio guidelines for submitting code & software for further information.

## Data

Policy information about availability of data

All manuscripts must include a data availability statement. This statement should provide the following information, where applicable:

- Accession codes, unique identifiers, or web links for publicly available datasets
- A description of any restrictions on data availability
- For clinical datasets or third party data, please ensure that the statement adheres to our policy

Raw sequencing reads used in this project are available under SRA Bioprojects PRJNA744894, PRJNA746324, PRJNA610049, PRJNA361133, and PRJNA745218. Mass spectrometry data are available on PRIDE with identifier PXD046217, and other datasets necessary to recreate the results of the publication are available on Dryad (https://doi.org/10.5061/dryad.j3tx95xmx). Templates for Complex I protein structural modeling were accessed from the Protein Data Bank (PDB) with accession numbers 6G2J, 6G72, 5LDW, 5LNK, and 5XTC.

## Research involving human participants, their data, or biological material

Policy information about studies with human participants or human data. See also policy information about sex, gender (identity/presentation), and sexual orientation and race, ethnicity and racism.

| | |
|---|---|
| Reporting on sex and gender | N/A |
| Reporting on race, ethnicity, or other socially relevant groupings | N/A |
| Population characteristics | N/A |
| Recruitment | N/A |
| Ethics oversight | N/A |

Note that full information on the approval of the study protocol must also be provided in the manuscript.

# Field-specific reporting

Please select the one below that is the best fit for your research. If you are not sure, read the appropriate sections before making your selection.

☐ Life sciences      ☐ Behavioural & social sciences      ☒ Ecological, evolutionary & environmental sciences

For a reference copy of the document with all sections, see nature.com/documents/nr-reporting-summary-flat.pdf

# Ecological, evolutionary & environmental sciences study design

All studies must disclose on these points even when the disclosure is negative.

| | |
|---|---|
| Study description | The study tested for evidence of mitonuclear incompatibilities in hybrid fishes by using a combination of lab-born hybrids and natural hybrid populations. Most quantitative analyses were performed in a linear model framework, with genotypes at key loci as independent variables (along with confounding factors such as genome-wide ancestry, brood, age, etc.) and various phenotypes of interest (survival, respiration rates, morphometrics) as dependent variables. |
| Research sample | Our sample included a total of 1253 lab-born F2 hybrids between Xiphophorus birchmanni and Xiphophorus malinche, as well as 952 natural late-generation X. birchmanni x malinche hybrids from five populations in three rivers. This combination of lab-raised vs. wild hybrids and early- vs. late-generation hybrids from multiple demographic sources allowed the best possible combination of statistical power, precision, natural replication and experimental tractability in inferring the effects of particular genotype combinations. Wild source populations included Aguazarca (N = 126), Calnali Low (N = 359), and Chahuaco Falls (N = 244) all on the Rio Calnali, as well as Acuapa (N = 117) on the Rio Huazalingo, and Tlatemaco (N = 126) on a tributary to the Rio Claro. All pure X. birchmanni and X. malinche individuals used in the study for experimentation and generation of F2 hybrids were collected from Coacuilco, Hidalgo, Mexico, and Chicayotla, Hidalgo, Mexico. Two additional X. malinche for phylogenetics were collected at Tetipanchalco and Tecpaco, Hidalgo, Mexico. Of the F2 hybrids, 943 were drawn from an existing dataset of adults (6+ months old) raised at the CICHAZ field station in Hidalgo, Mexico for genetic mapping (Powell et al. 2021 Current Biology, SRA BioProject PRJNA692059), 75 were juveniles (3-5 months old) born and raised at Stanford University used to track lethality timing and heart histological phenotypes (one of which was euthanized and not included in statistical analyses), and 235 were embryos dissected out of 10 F1 hybrid mothers at Stanford for embryo respirometry assays. For differential expression and allele-specific expression analyses, we used existing RNA-seq data from a prior project (Payne et al. 2022 Molecular Ecology, SRA BioProject PRJNA746324) containing 3 each of male F1 hybrids, X. birchmanni, and X. malinche. We used a total of 16 X. malinche, 16 X. birchmanni, 14 F1 hybrids, and 9 lab-born progeny of Calnali Low hybrids for qPCR, Oroboros respirometry, mitochondrial membrane potential, and mass spectrometry analyses. For embryo stage analyses in wild hybrids, we dissected embryos out of 38 females from the previously mentioned collections, and successfully genotyped a total of 269 embryos included in the manuscript. We also successfully genotyped 8 and 11 broods dissected from 29 X. |

birchmanni and 36 X. malinche wild-caught females, respectively. For analysis of embryo survival outside the womb, we used a total of 20 X. birchmanni fry from two females and 25 F1 fry from four females. In testing Xiphophorus fry sensitivity to Complex I inhibition, we used a total of 39 X. birchmanni fry from three broods, and 11 X. malinche fry from one brood. To construct de novo whole-mtDNA phylogenies, we used 5 X. birchmanni, 3 X. malinche, and 2 X. cortezi from Huichihuayan, San Luis Potosi, Mexico, and the Xiphophorus Stock Center at Texas State University. For mapping of the incompatibility between X. birchmanni and X. cortezi, we used 284 natural X. birchmanni x cortezi hybrids from Santa Cruz, Huextetitla, and Chapulhuacanito, San Luis Potosi, Mexico.

| | |
|---|---|
| Sampling strategy | Sample sizes for the Powell et al. F2 cross were determined using a power analysis for a previous QTL mapping study, where sample size was chosen for an estimated 90% chance of detecting a QTL explaining 5% of phenotypic variance. Sample sizes for all individuals collected from natural populations were determined by sampling success during field seasons and the maximum permitted across all sampling sites by collection permits. Sample sizes for all analyses based on lab-born hybrids were either the total number available with the desired genotypes at the time of experimentation, or the maximum feasible while still preserving the viability of lab colonies (which can only periodically be refreshed with wild collections). |
| Data collection | Data collection procedures are listed in the main text and Supplement (being too extensive to list here). SMB, AED, and JJB performed wet lab work for genotyping, TRG and MS performed bioinformatics for genotyping, and BMM and QKL contributed to both. TRG performed Complex I pharmacological inhibition trials and all other measurements of fry survival. BMM, DLP, and SMB collected embryo stage data from wild-caught hybrids. BMM performed respirometry and morphometrics in live embryos, while ENKI and JCH performed Oroboros respirometry assays on isolated mitochondria. SMB performed qPCR for mitochondrial copy number, CYP performed differential expression and allele-specific expression analyses, and AM performed assays of mitochondrial membrane polarization. FL, RM, KS, and RDL performed mass spectrometry assays on mitochondrial isolates prepared by BMM and AED. RRS analyzed histology images of juvenile F2 hearts. BMM constructed all phylogenies and associated testing of evolutionary rates, performed structural modeling, and carried out coevolution tests. MS and BMM performed all simulation and statistical analyses. |
| Timing and spatial scale | All natural hybrid samples and parental species were collected over three years of field trips to the CICHAZ field station in Calnali, Hidalgo, Mexico, from ten different sites within an area of Hidalgo and San Luis Potosi ~40 km in diameter. Sampling was roughly quarterly with the exception of the year 2020, when sampling was limited by the COVID-19 pandemic. |
| Data exclusions | 18 individuals were excluded from whole embryo respirometry and morphometrics due to damage (punctured yolks, torn tissue, severe internal bleeding, etc) at the time of photography as identified by BMM; these damages could affect body measurements, and it was unclear when the damage occurred, such that a confounding effect on respirometry measurements could not be ruled out. Likewise, four heart compartment slides were excluded from histology analysis due to visible tearing of the tissue on the slide, which was deemed sufficient to affect cross-sectional area by BMM. These criteria were not pre-established, but the observer was unaware of the experimental treatment group when making the decision to exclude. |
| Reproducibility | Replicability was largely confirmed by comparing across biologically independent datasets and statistical approaches: the depletion of mismatched genotypes was repeated in segregation distortion in lab-born F2 hybrids, partial correlation analysis in two natural hybrid populations with both mitochondrial ancestries segregating, and permutation-based testing of depletion in three natural hybrid populations fixed for mitochondrial ancestry. Replicability of in silico protein modelling was tested by repeating structure prediction with five different random seeds and four different mammalian cryo-EM templates. Whole embryo respirometry was replicated across ten broods in two major sets separated by ~6 months. |
| Randomization | Randomization is not relevant to our study because all analyses either lacked experimental treatments or applied a single treatment to all individuals, with the genotypes of individual being the independent variable of interest. Where applicable, the covariate of genome-wide ancestry fraction was controlled for using partial correlation analysis, and any variation in the administration of respirometry methods were accounted for for by using date of testing as a blocking variable in downstream statistics. |
| Blinding | Blinding was not employed in this study. Many analyses featured only a single group, such that blinding was not relevant, and in others, the placement of individuals in experimental groups of interest (e.g. genotypes) was impossible until after all data collection and analysis had already occurred. In the case of Oroboros respirometry, treatment groups of individuals were visible to experimenters based on phenotype. |

Did the study involve field work?   ☒ Yes   ☐ No

# Field work, collection and transport

| | |
|---|---|
| Field conditions | Collections were performed in September, November, January, February, March, May, and June in the Sierra y Huasteca region of Mexico, with sites ranging from ~10-30 C depending on the season, and rainfall varying from absent during the peak dry season (winter and spring) to frequent and substantial in the rainy season (summer and fall). |
| Location | Sampling was carried out in 0-2 meter of water at all sites, including:<br>Coacuilco:  21.097544° latitude, -98.588917° longitude, 315 m elevation<br>Chicayotla: 20.924232°, -98.576144°, 1020 m<br>Xontla Falls Up:  20.926958°, -98.588595°, 1195m<br>Tetipanchalco:  20.879285°, -98.799525°, 680 m<br>Aguazarca:  20.898505°, -98.602150°, 980 m<br>Calnali Low: 20.899356°, -98.575438°, 920 m<br>Chahuaco Falls: 20.906892°, -98.537260°, 610 m<br>Tlatemaco:  21.022704°, - 98.790106°, 375 m<br>Acuapa: 20.955922°, -98.571274°, 510 m<br>Santa Cruz:  21.157675°, -98.520497°, 145 m |

| | Huextetitla: 21.162172°, -98.557554°, 165 m<br>Chapulhuacanito: 21.210835°, -98.670220°, 145 m |
| --- | --- |
| Access & import/export | Samples were collected under Collection Permit No. PPF/DGOPA-002/19 issued from the Mexican National Commission of Aquaculture and Fisheries (CONAPESCA, issued 8/28/2020) and imported under Long-Term Importation Permit #2023 - 7129 from the California Department of Fish and Game (issued 2/9/2023). |
| Disturbance | No disturbance was caused by the study except the removal of Xiphophorus from the streams. Sample sizes were small relative to the population sizes of these species and the timespan over which the samples were collected, but we nonetheless monitored for signs of population perturbation caused by sampling (changes in age distribution, sex ratio, catch per trap, etc.) |

# Reporting for specific materials, systems and methods

We require information from authors about some types of materials, experimental systems and methods used in many studies. Here, indicate whether each material, system or method listed is relevant to your study. If you are not sure if a list item applies to your research, read the appropriate section before selecting a response.

## Materials & experimental systems

| n/a | Involved in the study |
| --- | --- |
| ☒ | Antibodies |
| ☒ | Eukaryotic cell lines |
| ☒ | Palaeontology and archaeology |
| ☐ | ☒ Animals and other organisms |
| ☒ | Clinical data |
| ☒ | Dual use research of concern |
| ☒ | Plants |

## Methods

| n/a | Involved in the study |
| --- | --- |
| ☒ | ChIP-seq |
| ☒ | Flow cytometry |
| ☒ | MRI-based neuroimaging |

# Animals and other research organisms

Policy information about studies involving animals; ARRIVE guidelines recommended for reporting animal research, and Sex and Gender in Research

| Laboratory animals | Laboratory animals included adult male and female Xiphophorus birchmanni and X. malinche, as well as F1 and F2 hybrids between them. Xiphophorus fry were fed newly hatched (<24 hours old) brine shrimp (Artemia franciscana, Great Salt Lake strain) |
| --- | --- |
| Wild animals | Wild-caught animals used in this study included male and female X. birchmanni, X. malinche, X. pygmaeus, X. cortezi, X. birchmanni x malinche hybrids, and X. birchmanni x cortezi hybrids. X. pygmaeus and X. cortezi individuals were all adult (sexually mature, exact ages are impossible to know in wild fish but likely greater than 6 months post-birth), while all other sampling included embryos (less than ~30 days post-fertilization and pre-birth), juveniles (sexually immature, likely <6 months post-birth), and adults (sexually mature, likely >6 months post-birth). Individuals were caught with baited minnow traps and transported in minnow buckets for local travel and methylene-blue treated water containers for long-distance travel. If lethal sampling was not necessary, captive animals were either non-lethally fin clipped for DNA before release within a week of capture at the point of capture, or returned to the laboratory to establish breeding colonies. Dissection was necessary to remove embryos or liver for RNA, protein, and respirometry analyses; in these cases, captive animals were euthanized using lethal overdoses of tricaine (MS-222). |
| Reporting on sex | Sex was not considered in analyses involving embryos, fry, or juveniles, as sexual differentiation had not yet occurred and the sex determining loci for X. birchmanni and X. malinche have yet to be identified. Sex was balanced across groups and tested for significant effects in the Oroboros mitochondrial respiration assays, but had no significant effect on any parameters. Analyses based on genotypes of wild-caught individuals include males and females in the proportions captured, and so selection and dominance coefficients referenced in the manuscript are sex-averaged. RNA-seq and membrane polarization assays were performed exclusively on males, and mass spectrometry exclusively on females, to exclude potential sex effects. |
| Field-collected samples | This study did not involve samples collected from the field |
| Ethics oversight | Ethical approval was provided by the Stanford Administrative Panel on Laboratory Animal Care (APLAC), protocol #33071 |

Note that full information on the approval of the study protocol must also be provided in the manuscript.

