## [Peer Review File · Nature]

Manuscript Title: A Lethal Mitonuclear Incompatibility in Complex I of Natural Hybrids

Reviewer Comments & Author Rebuttals

Reviewer Reports on the Initial Version:

Referees' comments:

Referee #1 (Remarks to the Author):

This is a fascinating study on the evolution of mitochondrial-nuclear incompatibilities. The study leverage naturally occurring hybrids of two fish species to disentangle the cause of observed hybrid incompatibilities. The hierarchical approach that the manuscript takes from genomes to physiology is a strong approach for addressing the mechanistic basis of mitochondrial-nuclear incompatibilities.

One thing for me is that the manuscript does not sufficiently acknowledge the work that has already been done on this system. Much of the data was generated and analyzed in previous papers by the same group and this study builds on the strong foundation they have already built. For example, additional sentences about the existing incompatibilities and motivation for the study and crosses (lines 100-101) would be appropriate. I find the lack of acknowledgement and more direct statement of this to be a bit misleading, especially for researchers of non-model organisms who may be interested in studying hybrid incompatibility.

The introduction outlines several limitations to the study of hybrid incompatibilities, including the limited exploration in a small number of model species, I would have liked the discussion to return to some of the broader ideas outlined in the introduction and generalization of the process.

I found the statement (line 100) "some hybrids between *X. birchmanni* and *X. malinche* are viable and fertile, while others experience strong selection against incompatibilities" to be confusing. The selection is against the resulting hybrids, which are incompatible.

The authors had previously found a sword length QTL on chromosome 13 in this same cross, was the same region found or a different region on chromosome 13? Was the sword length QTL erroneous and reflecting the hybrid incompatibility?

The statement (line 138) "To directly evaluate evidence for a mito-nuclear incompatibility and pinpoint the region involved on chromosome 13," is unclear to me as the Chr 13 had already been shown as an incompatibility. This was a natural population to confirm the laboratory cross findings.

The genomic and physiological data points to a strong role of *ndufs5* than *ndufa13* in the mitochondrial incompatibilities. However, the gene *ndufaf4* is not studied despite several lines of evidence that it may be involved in the mito-nuclear incompatibilities. While there are no non-synonymous substitutions, were there any synonymous substitutions that may affect protein structure and therefore protein interactions?

The branch tests in codeml was not sufficiently described to rule out neutrality. The apparent rapid evolution may be due to either relaxation of selection or rapid evolution. A comparison to $w=1$ should also be included in the analyses. The results are compelling regardless of the evolutionary history, but it is important to analyze those data correctly.

Is there linkage disequilibrium between the putatively rapidly adaptively evolving genes and/or to the mitochondria? There are multiple tests to detect the coevolution of loci, this would be appropriate to implement given the hypothesis of the coevolution of these loci.

The findings that *X. cortezi* may have inherited hybrid incompatibility loci via hybridization is very exciting. Compelling and tantalizing final point in the discussion!

Referee #2 (Remarks to the Author):

The manuscript addresses evolution and genetic compatibility between multiple genes – an exciting topic of discussion in mitochondrial research where (in)compatibilities between the nuclear and mito genomes would be expected in the oxphos complexes that combine the gene products of both – but have not been substantiated/identified. This manuscript reports one such incompatibility, involving nuclear and mito gene products that ‘meet’ in complex I. Having read the abstract, I was excited to read the paper and discover a real example of a multi-gene incompatibility characterized on all levels – population, genetics, evolution, protein.

First, I note that I am not a geneticist. I followed the manuscript through the first sections without substantial difficulties and found the presentation clear and logical – but I have no expert review comments to offer.

I then encountered three major issues with the report. First, I am not satisfied with the functional analysis (Figures 2D and 2E) – direct evidence for complex I dysfunction appears lacking [1]. Second, I was unable to properly evaluate the structure models and the structure analysis and was unable to form a clear picture of the number mutations present and their locations in the structure [2]. Third, and undermining my opinion of the whole manuscript considerably, I find the analysis of the protein-protein interactions unsatisfactory and was unable to convince myself that this final piece of the puzzle (physical/structural explanation of the incompatible residues/mutations) really has dropped into place [3].

1. Figures 2D and 2E. I do not find this evidence for the decreased function of complex I satisfactory. This may be a problem with the data, or due to the presentation/my understanding of it. First, I am confused by the measure of ‘complex I efficiency’ – stated on SI-pg 17 as “how much ADP increased respiration relative to the prior respiratory state in each run (analogous to the respiratory control ratio)”. How can the ratio be less than one, if ADP increased the rate? Considering Fig S14, and measuring once the rate has become constant after the addition of MPG (condition A) I estimate (very roughly) around 12 and around 35 after addition of ADP (condition B) – this would give around

3. The values plotted in D are 0.6-0.8, so I am further confused about what is shown. I also don't think that the term 'complex I efficiency' is appropriate here. Efficiency could be kinetic – how fast can it go – in which case, the rotenone-sensitive rate after addition of CCCP would be a better measure. Or it could be energetic – but how is this ratio a measure of the efficiency of energy conversion by complex I? This measured ratio is likely a mixture of a number of effects, including the integrity of the membrane (more leaky = smaller value, in my formulation) and the rate of catalysis of ATP synthase (faster rate = bigger value). In 2E the maximum respiration is (presumably) the total rate after addition of CCCP. Why not the rotenone-sensitive component of this to focus the value on complex I? For example, perhaps there is compensation through increase of complex II-linked catalysis in the case of dysfunctional complex I?

2. Structure predictions

- i) What are the %identities and %similarities in the amino acid sequences of the subunits in question between the swordtail and the template mammalian species? Please present sequence alignments. I am unable to tell if it is very straightforward to identify the locations of the variant amino acids (high %identity) or whether this is a challenge for structure prediction (greater uncertainty). My instinct is the former, and that the presentation is more complicated than necessary – but without this background information I can't tell. That is, would an approach of sequence homology and mapping conserved residues directly be simpler and sufficient? In terms of overall similarity - are the swordtail species predicted to contain the same complement of supernumerary subunits as the mammalian species? Are the subunits in question similar lengths, or are they truncated or extended relative to the mammalian templates?
- ii) For the output of the predictions, please present overlays of the modeled swordtail structures with the mammalian templates eg. in an overview like 3A and in an analogous view to S17 - for cross-referencing to the sequence alignments.
- iii) I note on SI-page 20 you refer to two different conformations of a helix in ND6 – this is correct and both are likely relevant to complex I, in different states, as has been discussed for example in ref. 43. Were both taken into account?

Identification and importance of substitutions

- i) Please present sequence alignments for *X. birchmanni* and *X. malinche* for all the subunits in question. The secondary structures of the sequences could be annotated for easy cross referencing to the structure. Positions considered important could be highlighted. Are all the substitutions listed in Table S4?
- ii) The positions at which variants are found should all be mapped onto the structure and displayed – not just the ones the authors have selected as relevant. This would give a better idea of the complexity of the question, and a better overview of the level of variation. It would also allow the reader to validate the interpretations.
- iii) My understanding is that there should be partner mutations in the mito and nuclear genomes: AA works (in one species), BB works (in a second species), but AB and BA do not work (A and B are incompatible). Where is this analysis? I am left hanging with general comments about mutations at interfaces, and mutations predicted to affect protein function – but with a lack of any specific information. We are informed “one *ndufs5* substitution directly contacts a substitution in *nd2*” on page 13 – Fig S16?? and Fig 3B (but what should I look at in 3B?) We are referred to an 'unstructured

loop in ND6' – but 99.4% of ND6 was modeled in mouse complex I – and is this loop on the matrix side or the intermembrane space side (together with the proteins highlighted in 3A)? I find the analyses of the mutations in the protein structure too generic and lacking specific information to substantiate the proposal of interacting substitutions. Have the authors actually identified physically interacting positions or not? Perhaps one? I would like to see a comprehensive analysis of the structural interdependence of these mutations in order to properly substantiate the statement that “the lethal form of the hybrid incompatibility is driven by dysfunctional protein-protein interactions in Complex I of the mitochondrial electron transport chain”.

Referee #3 (Remarks to the Author):

Xiphophorus is an outstanding system to understand the genetic basis of hybrid defects that maintain species apart. In this piece Moran et al. try to identify alleles that might lead to hybrid mortality in early stages of divergence using genome patterns of segregation in naturally occurring hybrid zones. The manuscript has several sections that aim to identify the multiple genetic components of what the authors suggest is a complex hybrid incompatibility. However, I have several major concerns that leave me unconvinced.

First, the authors use association mapping in hybrids to identify the nuclear partners of the incompatibility between *X. birchmanni* × *X. malinche*. The approach is the same as the used in a previous paper (Powell et al. 2020) and has the potential to reveal hybrid incompatibilities in species that hybridize in nature. A difference in this paper is that the authors do not regress on a particular deleterious phenotype (such as the size of melanoma) but instead do so on the mitochondrial genotype of the hybrids. The details on this approach could be further elaborated in the methods as they were not clear to me. A question that came to my mind was whether the ancestry of xmark and myrip (identified from Powell et al. 2020) could affect the results of the mapping here presented because of hybrid lethality.

The authors look at admixture deviation in a controlled F2 cross in the lab. They find an X-linked haplotype that is rare in one of the directions of the cross and conclude that this paucity of admixture is caused by a mitonuclear interaction. In the results from this first part of the manuscript the authors identify an X-linked haplotype of 380 kb that contains over 30 genes. Since this haplotype is so rare in F2s, the authors conclude that this dearth of genetic ancestry must be caused by a mitonuclear hybrid incompatibility. Given the relatively low power of the mapping approach (as stated by the authors in line 175), I am not certain they can rule out that a nuclear-nuclear interaction might be involved in the trait, as readily as they do in the manuscript. This result and further assumption are fundamental for the paper because the admixture mapping results are contingent on the existence of a mitochondrial partner in the interaction. This makes sense but seems rather arbitrary as a conclusion. For example, the dearth of ancestry could be caused by other maternal effects (such as RNA deposition), especially given that the downstream phenotype is a delay in developmental time. Since the authors do not elaborate about the other genes in this genome segment I can't assess whether this hypothesis is possible or likely. I am not implying that the authors' conclusion is not possible, but it seems like a more systematic exploration of the

alternatives is necessary.

The authors then explore their conclusion of the putative mitonuclear interaction in more depth. They conclude that of the ~30 genes in the focal 380 kb segment, two have mitochondrial functions, but only one interacts directly with mitochondrially encoded proteins, implicating this gene (*ndufs5*) as the cause of the incompatibility. This rationale to narrow down the allele is somewhat odd because the difference between 'having a mitochondrial function' and 'a direct mitochondrial interaction' is never really elaborated in the manuscript, and several re-reads, I don't understand why the incompatibility must occur through proteins that physically interact with each other and not through alleles that are epistatic to each other.

In the third part of the manuscript, the authors state that there is evidence for a third partner to this epistatic interaction. The evidence for the three-way interaction is rather weak. The rationale is that if they are able to detect other alleles that have undergone selection in hybrids at the Mitochondrial protein respiratory Complex I, then those alleles must be part of the hybrid incompatibility. I might be missing something (the methods are brief in this aspect) but my understanding is that the whole premise of this is a partial correlation analysis in which they find that *ndufa13* has no mixed ancestry in hybrid zones. This makes me revisit a point that I made earlier. The authors state that there is no nuclear-nuclear interactions involved in the trait, and yet, the potential third partner is autosomal. This result needs to be better reconciled with the results from the genome association mapping. The paper makes no effort to rule out potential causes for this long range admixture LD which makes me think that the results are less conclusive the authors make it sound.

In short, the authors pose an audacious hypothesis that will require herculean genetic work to verify. In spite of my enthusiasm for the topic, the paper has too many assumptions which dampens my enthusiasm for the current piece.

Referee #4 (Remarks to the Author):

The molecular mechanisms underlying hybrid incompatibility are fundamental questions in evolutionary biology. When discovered, they can explain why and how closely related species maintain reproductive barriers that control speciation. Nuclear-mitochondrial mismatch is known to be a major mechanism of hybrid incompatibility, particularly in yeast and plants. This is evidenced by mitonuclear coevolution during speciation and is a result of the strict mitochondrial-nuclear coordination required for successful biogenesis of the electron transport chain (ETC). However, the specific mismatches between the 13 mt-encoded proteins and the >120 nuclear-encoded ETC subunits/assembly factors that drive hybrid incompatibility remain undefined. In this work, Moran and colleagues, through a heroic effort using genetic mapping of naturally hybridizing wild admixture and lab-bred swordtail populations, narrow down the players responsible to hybrid incompatibility to just 4 genes: *nd2*, *nd6*, *ndufs5* and potentially *ndufa13*. They propose that incompatibility of mitochondrial *nd2/6* with nuclear *ndufs5* leads to embryonic lethality of hybrids as a result of impaired Complex I enzymatic function. Heterozygous hybrids attempt to mitigate the

incompatibility by preferentially incorporating the compatible *ndufs5* homologue into CI. They further pinpoint a handful of residues that might be responsible for the incompatibility between *x.malinche* and *x.birchmanni* *nd2/6* and *ndufs5*. Lastly, they show that *ndufs6/a13* undergo rapid evolution and co-introgression with compatible mitochondrial genotypes. The authors argue that these candidates fulfill requirements of classic models as molecular determinants of hybrid incompatibility. I will judge the paper according to the following criteria, wherein I will also incorporate my critique.

1) Is it novel and significant?

Since the concept of mito-nuclear incompatibility is well known, conceptually this study offers a moderate advancement, in that it solidifies a known theory by advancing it to a genic level. However, it is the first study to implicate specific genes in hybrid incompatibility in wild naturally hybridizing populations, which makes it a pioneering study. Overall novelty is 8/10.

The study's significance largely depends on how widely applicable these genes are to general hybrid incompatibility. Can the authors address the question of how general the genetic interactions between *nd2/6* and *ndufs5/a13* are in other hybridizing species that experience incompatibility? In simple words, how generally applicable are their findings to other hybridizing species? If so, I would say their findings are very significant and offers as broad conceptual advance.

2) Are the data of high quality?

Yes, reading through the methods, this study was very well executed. Data quality 9/10.

3) Did the authors make the right interpretation? Is it likely to be true?

This is perhaps what I found to be the most problematic with the study. The segregation patterns demonstrated in Figure 2A, B in viable versus non-viable hybrids convinces me that the authors have indeed identified the correct locus. However, given that there are 32 genes in this interval, I am not entirely convinced that they have indeed identified the right gene.

I raise this objection because the small reduction seen in Complex I efficiency (Fig 2D), which does not affect maximal respiration (Fig 2E) is unlikely to have such profound impacts on embryonic development. Similarly, Fig 2G demonstrates a 10% preference bias in incorporation of the compatible allele. If the incompatible *x.birchmanni* proteins really destabilize CI leading to lethal effects, these should never be incorporated/stable enough to be detected. Generally, a 10% drop in CI activity will not lead to lethality due to the inherent buffering capacity of mitochondria and of high glycolytic rates during larval development. Zebrafish larvae can withstand rotenone treatment to quite high levels (>100nM, which significantly affects CI activity) and still remain viable (PMID: 28732770). Model organisms like mice, and humans, can withstand *NDUFS4* mutations and be born live, only to succumb to Leigh syndrome in childhood.

Therefore, the authors conclusion that a small reduction in CI efficiency can lead to such drastic negative selection needs further examination.

Some suggestions to enhance the credibility of the authors' conclusions:

- i. Demonstrate that mild inhibition of CI pharmacologically can lead to lethality of *Xiphophorus* larvae.
- ii. Demonstrate that *X.birchmanni ndufs5* is sufficient to destabilize CI in cells with *X.malinche* mt-genome. Some suggestions:

-Blue Native PAGE of liver/muscle mitochondria from heterozygous hybrids compared to parental mitochondria should reveal significant CI disassembly, particularly when probed with proteins from the ND1/2 modules.

-Recapitulate bioenergetic and biochemical defects in fibroblasts derived from *X.malinche* with stable overexpression of *X.birchmanni* *ndufs5*.

-Recapitulate the proposed incompatible residues in a cell culture system i.e. mutate the substituted NDUFS5 residues in mammalian cell culture system and demonstrate that it leads to mito-nuclear incompatibility. This would also help to prove that this system works in even higher organisms.

-In experimental genetics, the classic way to prove that something is responsible for a phenotype is to rescue it. I am not sure if transgenic techniques work for swordfish. Is there was a way to introduce *X.malinche* *ndufs5* mRNA into hybrids to rescue the *X.birchmanni* x *X.malinche* *ndufs5* homozygotes? That would be very convincing. Any other methods to demonstrate that the other 31 genes are not responsible will also be welcomed.

Author Rebuttals to Initial Comments:

Response to reviewers

Referee #1 (Remarks to the Author):

This is a fascinating study on the evolution of mitochondrial-nuclear incompatibilities. The study leverage naturally occurring hybrids of two fish species to disentangle the cause of observed hybrid incompatibilities. The hierarchical approach that the manuscript takes from genomes to physiology is a strong approach for addressing the mechanistic basis of mitochondrial-nuclear incompatibilities.

We thank reviewer 1 for their recognition of the strengths of our manuscript and for their suggestions for improvement, which we have now incorporated.

One thing for me is that the manuscript does not sufficiently acknowledge the work that has already been done on this system. Much of the data was generated and analyzed in previous papers by the same group and this study builds on the strong foundation they have already built. For example, additional sentences about the existing incompatibilities and motivation for the study and crosses (lines 100-101) would be appropriate. I find the lack of acknowledgement and more direct statement of this to be a bit misleading, especially for researchers of non-model organisms who may be interested in studying hybrid incompatibility.

We apologize that this information was absent from the original submission. We agree that it is important context and have included this in the revised manuscript (lines 99-106).

The introduction outlines several limitations to the study of hybrid incompatibilities, including the limited exploration in a small number of model species, I would have liked the discussion to return to some of the broader ideas outlined in the introduction and generalization of the process.

We have revised the discussion to return to the broader ideas outlined in the introduction and put them in the context of our findings.

I found the statement (line 100) “some hybrids between *X. birchmanni* and *X. malinche* are viable and fertile, while others experience strong selection against incompatibilities” to be confusing. The selection is against the resulting hybrids, which are incompatible.

We have reworded this sentence and hope that it is now clearer (lines 99-101).

The authors had previously found a sword length QTL on chromosome 13 in this same cross, was the same region found or a different region on chromosome 13? Was the sword length QTL erroneous and reflecting the hybrid incompatibility?

Reviewer 1 raises an interesting point. Indeed, the major QTL that we previously identified underlying sword length (which controls 10% of the heritable variation in length) also falls on chromosome 13, approximately 14 Mb away from the mitonuclear DMI locus. In early generation hybrids there is weak linkage between these loci, and they are not in linkage disequilibrium in later generation hybrids. In response to reviewer 1’s comment, we re-analyzed our sword QTL dataset (N=537 F₂ males), conditioning on individuals with different recombination patterns along chromosome 13. Specifically, we looked at individuals that had a recombination event resulting in a different genotype at the sword QTL peak and *ndufs5* and asked if the effect of the sword QTL was weakened in this subset of individuals. We found no

evidence of a change in the effect size of the sword QTL peak across these different subsets of individuals. These results are shown in the plot below but we chose not to include them in the manuscript.

Figure legend: Two classes of adult F₂ males are found in our data, individuals homozygous for *X. malinche* ancestry at *ndufs5* (left) and individuals heterozygous for *X. malinche* ancestry at *ndufs5*. We separated our phenotypic data for adult males into these two ancestry classes and analyzed their genotypes at the sword QTL peak (~16 Mb on chromosome 13; *ndufs5* is at 2.06 Mb on the same chromosome). Individuals in the BB and MB categories in the left plot, and BB and MM categories in the right plot have an ancestry transition over the interval between *ndufs5* and the sword QTL. We find that the effect size of the sword QTL does not differ as a function of ancestry at *ndufs5*, indicating that selection on *ndufs5* in F₂ hybrids is not driving signal in our sword mapping results (presented in Powell et al. *Current Biology* 2021).

The statement (line 138) “To directly evaluate evidence for a mito-nuclear incompatibility and pinpoint the region involved on chromosome 13,” is unclear to me as the Chr 13 had already been shown as an incompatibility. This was a natural population to confirm the laboratory cross findings.

We have revised the wording here and hope that it is now clear (line 152-153). Specifically, while our artificial cross analysis showed evidence for a lethal incompatibility (and genotype patterns suggested it was mito-nuclear), because all individuals had *X. malinche* mitochondria, we could not directly test for mitochondrial interactions until examining hybrid populations with variable mitochondrial ancestry.

The genomic and physiological data points to a strong role of *ndufs5* than *ndufa13* in the mitochondrial incompatibilities. However, the gene *ndufaf4* is not studied despite several lines of evidence that it may be involved in the mito-nuclear incompatibilities. While there are no non-synonymous substitutions, were there any synonymous substitutions that may affect protein structure and therefore protein interactions?

We were able to revisit this result with some additional data in the revised manuscript. With an increase in sample size of ~50%, the regions linked to *ndufa13* and *ndufaf4* are significant or nearly-significant at a genome-wide 10% FPR threshold. Intriguingly, the region linked to *ndufaf4* interacts with the *X. birchmanni* mitochondria (Fig. 1), not the *X. malinche* mitochondria (the mitochondrial interaction we primarily focus on in the paper).

Despite the stronger mapping results in the revised manuscript, we still consider the evidence for the involvement of *ndufaf4* to be less clear. This is because the region associated with *X. birchmanni* mitochondrial ancestry around *ndufaf4* is large, stretching over several megabases and containing ~80 genes. Moreover, *ndufaf4* falls outside of the associated interval (Fig. S7). This suggests that *ndufaf4* may not be driving the signal of mitochondrial incompatibility. Coupled with the fact that there is no evidence of functional divergence between species in *ndufaf4*, and that this region involves interactions with the *X. birchmanni* mitochondria which is not the focus of the manuscript, we chose not to highlight it in the main text. However, we now discuss these results more thoroughly in the supplement, highlighting the comment from reviewer 1 about the potential phenotypic impacts of synonymous substitutions on translational efficiency or mRNA stability (see Supplementary Information 1.1.10).

The branch tests in codeml was not sufficiently described to rule out neutrality. The apparent rapid evolution may be due to either relaxation of selection or rapid evolution. A comparison to $w=1$ should also be included in the analyses. The results are compelling regardless of the evolutionary history, but it is important to analyze those data correctly.

We have now implemented this test as Reviewer 1 suggested. We find that we cannot reject a model of omega equal to 1 along the *X. birchmanni* branch and now report this in Table S8, Supplementary Information 1.5.1 and lines 341-344. While this could reflect relaxation of constraint in the *X. birchmanni* lineage as the reviewer points out, we believe it could also reflect variation in constraint along the protein. Specifically, we find much stronger conservation of amino acids at the nuclear interface of Complex I compared to the mitonuclear interface over longer evolutionary timescales (Fig. S31). Thus, we are cautious about the implications of tests for relaxed selection.

Is there linkage disequilibrium between the putatively rapidly adaptively evolving genes and/or to the mitochondria?

In hybrid populations, we observe strong ancestry linkage disequilibrium between *ndufs5*, *ndufa13*, and the mitochondrial haplotype (Calnali low $R_{ndufs5-mito} = 0.43$, $p < 10^{-19}$, $R_{ndufa13-mito} = 0.2$, $p = 0.005$; Chahuaco falls $R_{ndufs5-mito} = 0.32$, $p < 10^{-6}$, $R_{ndufa13-mito} = 0.29$, $p < 10^{-5}$). We do not observe ancestry linkage disequilibrium between *ndufs5* and *ndufa13* above what is expected given that both loci are in LD with the mitochondrial haplotype. Similarly, we do not observe strong LD between *ndufs5* and *ndufa13* in a population genetic dataset collected for *X. birchmanni* ($R = 0.08$; data from Schumer et al. *Science* 2018).

There are multiple tests to detect the coevolution of loci, this would be appropriate to implement given the hypothesis of the coevolution of these loci.

We appreciate the suggestion from reviewer 1 to explore explicit coevolutionary tests for *ndufs5*, *ndufa13*, *nd2* and *nd6*. To do so, we applied the GREMLIN analysis method to test for coevolution between sites in a multiple sequence alignment, and now include this information in

Supplementary Information 1.5.2. In brief, we found strong signals of coevolution between *ndufs5* and *ndufa13*, some significant signals of coevolution between *ndufs5* and *nd6*, and no evidence of coevolution between *ndufa13* and *nd6* or between *ndufs5* and *nd2* (Fig. S45). However, the residues with the strongest signatures of coevolution between *ndufs5* and *nd6* were not those that are in physical contact in the protein structures, nor did they overlap with substitutions that distinguish *X. birchmanni* and *X. malinche*. This may be in part because our analysis was underpowered due to the number of sequences available (see documentation at <http://gremlin.bakerlab.org>).

We also tested whether the RaptorX-Contact program (which is distinct from RaptorX template-based structure prediction) could predict the contact locations between *ndufs5* and each of *nd2*, *nd6*, and *ndufa13* from amino acid sequence alignments. RaptorX-Contact uses both coevolution information and training sets of protein structures to predict the locations of amino acids in physical contact. As in GREMLIN, the results of RaptorX-Contact pointed to stronger coevolution between *ndufs5* and *nd6* than *ndufs5* and *nd2* (Fig. S46). Moreover, this analysis highlighted potential coevolution between amino acids that differ between species at the interface of *ndufs5* and *nd6* as well as *ndufs5* and *ndufa13*. However, because RaptorX-Contact uses a combination of coevolution signals and a deep neural network trained on known protein structures we cannot directly attribute these results to coevolution between *ndufs5*, *nd6*, and *ndufa13*. We discuss the results of both programs in detail in Supplementary Information 1.5.2.

The findings that *X. cortezi* may have inherited hybrid incompatibility loci via hybridization is very exciting. Compelling and tantalizing final point in the discussion!

We agree that this is a very intriguing finding and are excited to explore it more!

Referee #2 (Remarks to the Author):

The manuscript addresses evolution and genetic compatibility between multiple genes – an exciting topic of discussion in mitochondrial research where (in)compatibilities between the nuclear and mito genomes would be expected in the oxphos complexes that combine the gene products of both – but have not been substantiated/identified. This manuscript reports one such incompatibility, involving nuclear and mito gene products that ‘meet’ in complex I. Having read the abstract, I was excited to read the paper and discover a real example of a multi-gene incompatibility characterized on all levels – population, genetics, evolution, protein. First, I note that I am not a geneticist. I followed the manuscript through the first sections without substantial difficulties and found the presentation clear and logical – but I have no expert review comments to offer.

We are glad that the reviewer found the topic and findings of great interest, the genetic results clear, and hope that our revised manuscript addresses their concerns (detailed below).

I then encountered three major issues with the report. First, I am not satisfied with the functional analysis (Figures 2D and 2E) – direct evidence for complex I dysfunction appears lacking [1]. We appreciate this point from reviewer 2, which was also shared by reviewer 4, and have added new data on the functional effects of the mitonuclear hybrid incompatibility, as well as new analysis and discussion. Regarding the concerns about functional data, we note that swordtails are far from a laboratory model (indeed to our knowledge our lab is only one of two successfully keeping *X. malinche* alive in captivity), so the experiments we present are unusually ambitious for this system and reflect nearly a decade of work.

Our measures of Complex I function are in individuals heterozygous for the hybrid incompatibility, as individuals that are homozygous do not survive past the middle stages of embryonic development. These heterozygous individuals have normal probabilities of survival through embryonic development and to adulthood, and thus we would expect impacts on Complex I in these individuals to be slight. This is indeed what we observe: individuals heterozygous for the mitonuclear incompatibility show some signs of altered Complex I function compared to either parental species, but overall similar patterns of mitochondrial function (Fig. 3).

Both reviewer 2 and reviewer 4 raised an important question that we explore in more detail in the revised manuscript. If heterozygotes have only modest Complex I dysfunction, why is the incompatibility lethal in homozygotes? While we alluded to this in the original version of the manuscript, we now more clearly discuss the possibility that the interaction between the developmental biology of swordtails and Complex I function may be key to its lethality (see Supplementary Information 1.3.1). Since swordtails must complete embryonic development in the maternal environment, we propose that delayed development itself could explain lethality (Table S4; Supplementary Information 1.3.1).

We have added additional data on the lethality of premature birth in swordtails (Supplementary Information 1.3.1), along with sequence data from a total of 296 embryos (69 added in revision) that more clearly shows the link between mitonuclear mismatch at *ndufs5* and developmental delay (Fig. 2). Interestingly, with this larger sample size we are able to more clearly conclude

that there is not a strong impact of *ndufa13* genotype on the developmental delay phenotype (Fig. S11, S13), although mismatch between the *X. malinche* mitochondria and *X. birchmanni* ancestry at *ndufa13* is largely lethal before adulthood.

Second, I was unable to properly evaluate the structure models and the structure analysis and was unable to form a clear picture of the number mutations present and their locations in the structure [2]. Third, and undermining my opinion of the whole manuscript considerably, I find the analysis of the protein-protein interactions unsatisfactory and was unable to convince myself that this final piece of the puzzle (physical/structural explanation of the incompatible residues/mutations) really has dropped into place [3].

We appreciate reviewer 2's feedback on the structural modeling in the manuscript and hope they will find the revised manuscript much improved in each of these respects. As we come from a genetics background, we found the reviewer's expert feedback on these issues invaluable and have revised the analyses, data presented, and manuscript in response to each issue, as we detail below.

1. Figures 2D and 2E. I do not find this evidence for the decreased function of complex I satisfactory. This may be a problem with the data, or due to the presentation/my understanding of it. First, I am confused by the measure of 'complex I efficiency' – stated on SI-pg 17 as “how much ADP increased respiration relative to the prior respiratory state in each run (analogous to the respiratory control ratio)”. How can the ratio be less than one, if ADP increased the rate? Considering Fig S14, and measuring once the rate has become constant after the addition of MPG (condition A) I estimate (very roughly) around 12 and around 35 after addition of ADP (condition B) – this would give around 3. The values plotted in D are 0.6-0.8, so I am further confused about what is shown. I also don't think that the term 'complex I efficiency' is appropriate here. Efficiency could be kinetic – how fast can it go – in which case, the rotenone-sensitive rate after addition of CCCP would be a better measure. Or it could be energetic – but how is this ratio a measure of the efficiency of energy conversion by complex I? This measured ratio is likely a mixture of a number of effects, including the integrity of the membrane (more leaky = smaller value, in my formulation) and the rate of catalysis of ATP synthase (faster rate = bigger value). In 2E the maximum respiration is (presumably) the total rate after addition of CCCP. Why not the rotenone-sensitive component of this to focus the value on complex I? For example, perhaps there is compensation through increase of complex II-linked catalysis in the case of dysfunctional complex I?

We value reviewer 2's points regarding the complexities of our mitochondrial physiology protocol, and hope that our responses here address their concerns. The “Complex I efficiency” value we report is not a ratio, but a flux control factor (FCF). The FCF is analogous to the respiratory control ratio, but scaled to a value between 0 and 1. For example, a respiratory control ratio of 2 (respiration increases 2-fold with ADP) would equal 0.5 for our CI Efficiency FCF. In the hypothetical case proposed by reviewer 2, the FCF would be $1 - (12/35) = 0.66$.

The “Complex I efficiency” FCF represents the proportion of total electron transport which is attributable to Complex I-driven respiration, as opposed to proton leak, in the absence of the Complex II substrate succinate. Though we agree that the name “Complex I efficiency” alone raises some of the confusions the reviewer touches on, we also point out that both this approach

and the use of FCFs are standard in the mitochondrial physiology literature (e.g. Pesta & Gnaiger 2012 *Methods Mol Biol*, Burtscher et al. 2015 *Mitochondrion*, Hoppel et al. 2021 *Cells*).

Reviewer 2 is right to suggest that this ratio does not reflect kinetic efficiency of Complex I, but rather an efficiency of energy conversion, specifically that of the entire electron transport system when routed through Complex I. As reviewer 2 points out, a decrease in Complex I efficiency could reflect a decrease in Complex I activity, a decrease in activity of a rate-limiting reaction downstream of Complex I in the electron transport system (such as ATP synthase), or an increase in proton leak due to changes in the permeability of the inner mitochondrial membrane.

- In regards to the reviewer's first concern about ATP synthase, the "CI + CII Efficiency" FCF measures the response of respiration to inhibition of this enzyme. While our power for this analysis is modest, we see no significant difference in this measure between genotypes, suggesting that ATP synthase activity is not the driver of Complex I efficiency differences (Fig. S18).
- To address the potential for differences in membrane integrity, we have now performed an explicit analysis of respiration rate in the LEAK state prior to ADP addition, and find no difference between genotypes (Fig. S18).
- In addition, we have evaluated membrane integrity in live cells using a flow cytometry-based assay (Fig. S19). Using a mitochondrial membrane potential dye and uncoupling agent (TMRE and FCCP), we found that hybrids heterozygous for the incompatibility were indistinguishable from parental individuals in both their baseline membrane potential and the change in membrane potential caused by FCCP permeabilization.

Together, these results point to reduced Complex I activity, not Complex V or increased proton leak, as the likely driver of reduced Complex I efficiency in hybrids in our respirometry assays.

We also appreciate Reviewer 2's question regarding the choice to emphasize Complex I efficiency, rather than the rotenone-sensitive component of maximum ETS capacity (a.k.a. the "CI flux control" FCF). Though these ratios both quantify changes in respiration after the manipulation of Complex I activity, the details of our multi-stage respirometry assay mean that they differ in two critical respects. First, the measurement of Complex I efficiency takes place while the mitochondria are in a fully coupled, ATP-generating state, while CI flux control is calculated in an uncoupled state (after the addition of CCCP). Second, the measurement of Complex I efficiency does not include Complex II-driven respiration, while CI flux control is measured with Complex II operating at maximum capacity (after the addition of succinate). Because we are primarily focused on changes in the activity of Complex I in the context of ATP-coupled respiration, we consider Complex I efficiency to be a more relevant measure on both accounts.

We also do not see the lack of change in CI flux control ratio as evidence against our hypothesis of decreased Complex I activity. For example, if the combined flux of electrons through Complexes I and II is not the rate-limiting step for the electron transport system in an uncoupled state, then the CI flux control ratio could be the same between hybrids and parentals even if Complex I activity is not. Specifically, even if Complex I function differs in hybrids, if the combined output of CI & CII in both hybrids and parentals exceeds the capacity of the downstream rate limiting step, the lower output of CI in hybrids will be hidden in this metric.

We believe that the results of our current assays are sufficient to demonstrate a change in Complex-I driven respiration in viable hybrids heterozygous for the incompatibility.

In revisiting the data based on reviewer 2's feedback, we were inspired to include a new analysis, which we had not presented previously. Specifically, mitochondria from hybrid individuals heterozygous for the incompatibility reach maximum Complex I-driven respiration at a dramatically slower rate than either parental species (now shown in Fig. 2). The same is not true for Complex II-driven respiration, where only nuclear genes are involved (Fig. 2). We believe this result is consistent with additional physiological effects on hybrids that are heterozygous for the incompatibility, and may speak to the kinetic efficiency of Complex I, as mentioned by reviewer 2.

2. Structure predictions

i) What are the %identities and %similarities in the amino acid sequences of the subunits in question between the swordtail and the template mammalian species? Please present sequence alignments. I am unable to tell if it is very straightforward to identify the locations of the variant amino acids (high %identity) or whether this is a challenge for structure prediction (greater uncertainty). My instinct is the former, and that the presentation is more complicated than necessary – but without this background information I can't tell. That is, would an approach of sequence homology and mapping conserved residues directly be simpler and sufficient?

We apologize that this information was not available in the original manuscript and have now included a supplementary table containing pairwise amino acid divergence between the *Mus* protein and *Xiphophorus* for all proteins in Complex I that were included in structural analysis (Table S5). Briefly, the nuclear proteins in Complex I are strongly conserved, with upwards of ~60% shared amino acid sequence identity (75% for *ndufs5* and 72% for *ndufa13*), while the mitochondrial proteins are somewhat less so (upwards of ~40% sequence identity). Notably, sequence similarity (quantified using BLOSUM 45 parameters) is >74% for all mitochondrial and mitonuclear genes modeled (Table S5).

Alignments of the proteins (now provided in Supplementary File 1 and Fig. S43-S44) indicate that the overall lengths of these proteins are largely conserved, but a few have amino acid insertions or deletions (including *nd6*, Table S5, S6). This motivated us to model the protein structures as described in the text rather than using the approach suggested by the reviewer. However, we are happy to explore that approach if the reviewer deems it appropriate given the information we have now provided on sequence homology.

In terms of overall similarity - are the swordtail species predicted to contain the same complement of supernumerary subunits as the mammalian species? Are the subunits in question similar lengths, or are they truncated or extended relative to the mammalian templates?

Swordtail species are predicted to contain the same complement of supernumerary subunits as mammalian species and the subunits in question are almost identical in length. Differences in length compared to the *Mus* sequences are all less than or equal to 10 amino acids, and most proteins with changes in length in swordtails compared to *Mus* also differ in length between

mammalian species (see Table S6, Supplementary Data File 1). Importantly, the predicted structures for swordtails are generally robust to the choice of model sequence (see Supplementary Information 1.4.6), and we highlight in the text the rare cases where the choice of model sequence impacts our conclusions.

We have now included this information in our description of the structural modeling (Supplementary Information 1.4.6), along with the supplementary tables and files containing alignments.

ii) For the output of the predictions, please present overlays of the modeled swordtail structures with the mammalian templates eg. in an overview like 3A and in an analogous view to S17 - for cross-referencing to the sequence alignments.

We now include figures that overlay the swordtail structures and mammalian templates (Fig. S23) in addition to the sequence alignments (provided in Supplementary Data File 1 and Fig. S43-S44). While RaptorX predictions yield nearly identical protein structures for mammalian and swordtail proteins (Fig. S23), as noted above, there are many amino differences between species, as expected given their deep evolutionary divergence. We have highlighted amino acid differences between the mammalian template and swordtail sequences in a separate set of figures (Fig. S24).

In generating these figures, we noted that residues at the interface between mitochondrial and nuclear proteins are far less conserved between swordtails and mammals than residues that do not contact mitochondrial proteins (Fig. S31). While not directly related to the reviewer's comments here, we think this observation provides an interesting picture of the interplay between protein-protein interactions, coevolution, and sequence conservation.

iii) I note on SI-page 20 you refer to two different conformations of a helix in ND6 – this is correct and both are likely relevant to complex I, in different states, as has been discussed for example in ref. 43. Were both taken into account?

We believe that reviewer 2 is referring to the presence or absence of a π -bulge in transmembrane helix 3 of ND6, which is associated with disorder and deactivation of Complex I. We thank the reviewer for their suggestion to delve more deeply into this phenomenon in *Xiphophorus*. Sequence alignment shows that the region between the π -bulge and the middle of transmembrane helix 4 is identical across *Xiphophorus* and is the most conserved region of the protein between *Xiphophorus* and the mammalian templates (65.6% of sites identical across all species at amino acids 54-85, Fig. S24), supporting the crucial role of this region in Complex I activity. Because of this lack of interspecific substitutions (and its distance from *ndufs5* and *ndufa13*) we expect that the presence or absence of a π -bulge is not the cause of Complex I dysfunction in *Xiphophorus* hybrids.

The turn that we are referring to in Supplementary Information 1.4.6 is located between transmembrane helices 4 and 5, rather than 3 and 4. We apologize that this was unclear in the original description, and have revised the current version. Specifically, a beta hairpin is present in sheep (PDB 5LNK) and cow (5LDW) structures, but not in the human (5XTC) or mouse (6G2J) structures. Inspection of additional cow and sheep structures (Kampjut and Sasanov

2020; Letts *et al.* 2019; Zhu *et al.* 2016; Blaza *et al.* 2018) showed that the beta hairpin was present across all known native catalytic states and conformations in these species. In contrast, no structures for human or mouse Complex I include this beta hairpin (Guo *et al.* 2017; Bridges *et al.* 2020; Agip *et al.* 2017; Yin *et al.* 2021; Chung *et al.* 2021). This led us to hypothesize that the presence of this turn likely differs from species to species.

We found that the predicted structure of this region in *Xiphophorus* depended entirely on whether we used a model structure with (sheep, cow) or without (human, mouse) the turn. Thus, we believe that we lack information about the *Xiphophorus* structure in this specific region and discuss it primarily as a source of uncertainty in our template-based structure predictions.

Identification and importance of substitutions

i) Please present sequence alignments for *X. birchmanni* and *X. malinche* for all the subunits in question. The secondary structures of the sequences could be annotated for easy cross referencing to the structure. Positions considered important could be highlighted. Are all the substitutions listed in Table S4?

We now have included sequence alignments for *X. birchmanni* and *X. malinche* that include annotations of secondary structures (Fig. S43-S44). All substitutions between species in any of the subunits modeled with RaptorX are now listed in Table S9 (previously Table S4). We had not originally included information for *nd3* and *nd4l*.

ii) The positions at which variants are found should all be mapped onto the structure and displayed – not just the ones the authors have selected as relevant. This would give a better idea of the complexity of the question, and a better overview of the level of variation. It would also allow the reader to validate the interpretations.

We apologize that this analysis was unclear and incomplete in the original version of the manuscript. We originally included the positions of all variants at the protein interface mapped onto the structures shown in Fig. 4 and Fig. S27 but had not labeled all of them. We now provide information on all variants mapped onto the structure of each protein modeled in our RaptorX analysis (Fig. S22) as well as a figure showing pairs of protein models that include the locations of substitutions (Fig. S26). In addition to these visualizations in the context of protein structure, we have added annotations highlighting substitutions between species in the sequence alignments discussed above (Fig. S43-S44).

iii) My understanding is that there should be partner mutations in the mito and nuclear genomes: AA works (in one species), BB works (in a second species), but AB and BA do not work (A and B are incompatible). Where is this analysis? I am left hanging with general comments about mutations at interfaces, and mutations predicted to affect protein function – but with a lack of any specific information. We are informed “one *ndufs5* substitution directly contacts a substitution in *nd2*” on page 13 – Fig S16?? and Fig 3B (but what should I look at in 3B?) We are referred to an ‘unstructured loop in ND6’ – but 99.4% of ND6 was modeled in mouse complex I – and is this loop on the matrix side or the intermembrane space side (together with the proteins highlighted in 3A)? I find the analyses of the mutations in the protein structure too generic and lacking specific information to substantiate the proposal of interacting substitutions.

We very much appreciate this feedback from reviewer 2, and have substantially changed the presentation in the manuscript and supplement in response (see lines 314-328; Supplementary Information 1.4.6). We address each of the concerns in turn here.

iii) My understanding is that there should be partner mutations in the mito and nuclear genomes: AA works (in one species), BB works (in a second species), but AB and BA do not work (A and B are incompatible). Where is this analysis? I am left hanging with general comments about mutations at interfaces, and mutations predicted to affect protein function – but with a lack of any specific information.

The reviewer correctly points out that in the Dobzhansky-Muller model (illustrated in Fig. S41), single interacting substitutions are envisioned. While the general principles of the model have been upheld, to our knowledge no studies to date have directly identified interacting substitutions (e.g. as in protein-protein interactions), due to the challenges we outline here:

- 1) Very few studies have successfully mapped any of the individual genes involved in hybrid incompatibilities. There are only about a dozen incompatibility genes known in total (see Presgraves et al. 2010 for a review). From these studies, we know of only 3 cases where the partner gene (or in our case genes) have been identified (Meiklejohn et al. 2013 *PLoS Genetics*; Powell et al. 2020 *Science* & Lu et al. 2020 *PNAS*; Boocock et al. 2021 *Science*), and only one case where researchers have successfully identified the substitutions that may interact. In that case (Meiklejohn et al 2013 *PLoS Genetics*), it was possible to identify the substitutions in question since each gene only had one substitution, but the mechanism of their interaction is unknown.
- 2) The technical challenges of narrowing hybrid incompatibilities to individual interactions is compounded if the genes involved contain multiple substitutions. This is because even when mapping to individual genes is successful, it is extremely difficult to distinguish between possible interacting mutations within the gene because recombination will so infrequently decouple these mutations. For example, a recombination event between any of the substitutions in *ndufs5* that distinguish *X. birchmanni* and *X. malinche* is predicted to occur only once in 25,000 meioses.
- 3) Finally, in the case of the incompatibility we map, in mitochondrial genes no recombination occurs at all and due to the high substitution rate, there are, as the reviewer rightly points out, multiple possible causal substitutions. Thus, in order to investigate this point, we relied on the protein structure modeling, identifying substitutions that were in proximity in the structure and/or were predicted to have an impact on protein function given their rarity over evolutionary timescales. We believe that the protein modeling (Fig. 4, Fig. S26, S27) underscores likely points of interaction, but have revised our language throughout the manuscript to highlight our uncertainty. We have revised figures (Fig. S22, S26, S43-S44) to show all substitutions, allowing readers to have a clearer view of the complexity of possible interactions.

In summary, we have now substantially revised our presentation in the manuscript to highlight the issues that reviewer 2 raises, be explicit about what the analyses we are able to conduct can and cannot tell us, and make it clear how many pairs of substitutions are predicted to be in physical contact (2-5 depending on the structural model and distance threshold). Regarding the

reviewer's request for more specific information about interacting substitutions, we highlight one pair of substitutions where large side chains could generate a steric clash (Supplementary Information 1.4.6), but confine this discussion to the supplement due to its speculative nature.

We are informed “one ndufs5 substitution directly contacts a substitution in nd2” on page 13 – Fig S16?? and Fig 3B (but what should I look at in 3B?) We are referred to an ‘unstructured loop in ND6’ – but 99.4% of ND6 was modeled in mouse complex I – and is this loop on the matrix side or the intermembrane space side (together with the proteins highlighted in 3A)?

The *nd6* loop in question was referred to as “unstructured” due to its lack of secondary structure, rather than lack of cryoEM structural resolution; we apologize for the confusion and have corrected our wording throughout the text. The *nd6* loop is on the intermembrane side of the complex, as are all the contact points between nuclear and mitochondrial genes highlighted in the manuscript.

In addition, we have modified Fig. 4B to highlight the predicted contact point between *ndufs5* and *nd2*, and other possible contact points between *ndufs5* and *nd6*. We have added detailed information about all possible interactions between mitochondrial and nuclear substitutions (lines 314-328; Supplementary Information 1.4.6). **Briefly, there is clear evidence of at least two contacts between *ndufs5* and mitochondrial substitutions, and the possibility of as many as five contacts between substitutions in the proteins of interest.**

I find the analyses of the mutations in the protein structure too generic and lacking specific information to substantiate the proposal of interacting substitutions. Have the authors actually identified physically interacting positions or not? Perhaps one? I would like to see a comprehensive analysis of the structural interdependence of these mutations in order to properly substantiate the statement that “the lethal form of the hybrid incompatibility is driven by dysfunctional protein-protein interactions in Complex I of the mitochondrial electron transport chain”.

We have significantly revised our language and presentation based on reviewer 2's feedback. Although we cannot identify physically interacting residues with absolute certainty with the modeling approaches we use, we now make clear that there are **two positions** with amino acid differences between species where we have high confidence of physical interaction based on RaptorX modeling (both between mitochondrial proteins and *ndufs5*; Fig. 4, Fig. S27) and **an additional three positions** where we consider physical interaction between substitutions possible given proximity between mutations (between *nd6*, *ndufs5*, and *ndufa13*; Fig. 4, Fig. S27) or uncertainties in the structural modeling (Fig. S25). Specifically, uncertainties in the structural modeling in our analyses arise from the choice of cryoEM template sequence (Fig. S25; Table S7) and a region of low local conservation between mammalian templates and the *Xiphophorus nd6* sequence between transmembrane helices 4 and 5 (see new discussion in Supplementary Information 1.4.6).

Referee #3 (Remarks to the Author):

Xiphophorus is an outstanding system to understand the genetic basis of hybrid defects that maintain species apart. In this piece Moran et al. try to identify alleles that might lead to hybrid mortality in early stages of divergence using genome patterns of segregation in naturally occurring hybrid zones. The manuscript has several sections that aim to identify the multiple genetic components of what the authors suggest is a complex hybrid incompatibility. However, I have several major concerns that leave me unconvinced.

We thank reviewer 3 for their recognition of the importance of this topic, close reading of the manuscript, and detailed comments for its improvement. We have addressed each of the concerns raised by reviewer 3 in detail with new analyses, data collection, and simulations, and hope that they will find the manuscript much strengthened.

First, the authors use association mapping in hybrids to identify the nuclear partners of the incompatibility between *X. birchmanni* × *X. malinche*. The approach is the same as the used in a previous paper (Powell et al. 2020) and has the potential to reveal hybrid incompatibilities in species that hybridize in nature. A difference in this paper is that the authors do not regress on a particular deleterious phenotype (such as the size of melanoma) but instead do so on the mitochondrial genotype of the hybrids. The details on this approach could be further elaborated in the methods as they were not clear to me. A question that came to my mind was whether the ancestry of *xmrk* and *myrip* (identified from Powell et al. 2020) could affect the results of the mapping here presented because of hybrid lethality.

We have added additional description in the main text (lines 158-159, 558-561, 565-566) and supplement (Supplementary Information 1.1.5) on the admixture mapping approach used here, which is similar to that used by Powell et al. 2021 but is designed to detect genotype x genotype interactions rather than genotype x phenotype interactions. Briefly, our approach tests for correlations in ancestry between genotypes at the mitochondria and focal nuclear loci, after regressing out the expected covariance between the mitochondrial and nuclear genotypes based on the individuals' genome-wide ancestry.

We recognize reviewer 3's concern with how previous incompatibilities identified in the *Xiphophorus* system could affect the mitonuclear results described here, and address such concerns in the revised manuscript. We did not include an analysis of *xmrk* and its interactor in the original manuscript because our past work has suggested that *xmrk* causes melanoma later in life, and thus would be inconsistent with the phenotype of embryonic lethality we observe here. Moreover, because individuals with the mitonuclear incompatibility we focus on here do not survive to adulthood, all individuals that develop melanoma in the natural populations studied by Powell et al. 2020 are not affected by the mitonuclear incompatibility.

However, to address the reviewer's question we have now included an analysis of melanoma-associated loci in embryos with and without the incompatible mitonuclear genotype combination (Supplementary Materials 1.1.9). Specifically, we identified embryos that are predicted to have melanoma risk based on their genotypes at *xmrk/myrip* and *cd97* and asked if the frequency of these individuals differed between embryos with and without the incompatible mitonuclear genotype combination using a Fisher's exact test. We see that the frequency of individuals that

would be predicted to develop melanoma based on their genotypes, should they survive to adulthood, did not differ between groups (Fisher's exact p-value =1).

The authors look at admixture deviation in a controlled F2 cross in the lab. They find an X-linked haplotype that is rare in one of the directions of the cross and conclude that this paucity of admixture is caused by a mitonuclear interaction. In the results from this first part of the manuscript the authors identify an X-linked haplotype of 380 kb that contains over 30 genes. Since this haplotype is so rare in F2s, the authors conclude that this dearth of genetic ancestry must be caused by a mitonuclear hybrid incompatibility.

We appreciate these comments from reviewer 3 as they highlight several issues that were not clear in the original manuscript. We hope these issues are now clear in the revised version and that the claims have been strengthened by our collection of additional data. In addition, we have added simulations to demonstrate some of the most important points. We also note here for clarity that chromosome 13 is not the X chromosome or X-linked (the X chromosome is chromosome 21 in *Xiphophorus*; Scharl et al. *Nature Genetics* 2013).

First, the reviewer raises questions about the confidence at which we can conclude that the chromosome 13 region is involved in a mitonuclear incompatibility. This is a prediction we made based on the observed genotypes at the chromosome 13 region (that we later test with admixture mapping, see below). Specifically, we originally stated that one would not expect to see an absence of homozygous *X. birchmanni* genotypes even with a dominant nuclear-nuclear incompatibility (Supplementary Information 1.1.2). We now show this explicitly in simulations in Supporting Information 1.1.1. Because homozygous parental genotype combinations would escape lethality, given our sample size we would expect to see on average ~100 individuals that are homozygous *X. birchmanni* at chromosome 13 even in the case of a dominant nuclear-nuclear incompatibility (Supplementary Information 1.1.1, Fig. S37). However, we do not take these patterns as conclusive evidence that there is a mitonuclear incompatibility involving chromosome 13, and go on to map it directly using hybrid populations (see below).

We very much appreciated this comment from reviewer 3 because looking quantitatively at the expectations for genotypes under different forms of selection impacted our interpretation of our other existing data. **Specifically, the results of these simulations underscored that the observed genotypes on chromosome 6 in F2s are also suggestive of a mitonuclear incompatibility** (Supplementary Information 1.1.1). This is another aspect of the paper that reviewer 3 commented on, which we believe is now strengthened (see below).

Given the relatively low power of the mapping approach (as stated by the authors in line 175), I am not certain they can rule out that a nuclear-nuclear interaction might be involved in the trait, as readily as they do in the manuscript.

We discussed power in several contexts and we realize that this was unclear in the original manuscript. The comment that the reviewer refers to was intended to refer to the moderate sample size of the admixture mapping experiment, not to our power to detect nuclear-nuclear versus mitonuclear interactions.

To address the reviewer's concerns about power and precision of mapping results, we have added additional data, simulations and analyses. First, we have increased our sample size by 50% to improve power. Second, in response to this comment, we have also added simulations to evaluate our power to detect nuclear-nuclear and different types of mitonuclear interactions in the natural hybrid datasets (Supplementary Information 1.2.3).

This result and further assumption are fundamental for the paper because the admixture mapping results are contingent on the existence of a mitochondrial partner in the interaction.

We respectfully disagree with the reviewer's statement here. Specifically, the admixture mapping analysis is designed to test the hypothesis that there is a mitochondrial association anywhere in the nuclear genome. Our simulations in Supplementary Information 1.1.5 demonstrate that in the absence of a true mitonuclear incompatibility, we do not expect to detect an association that exceeds the genome-wide significance threshold. However, we identify exceptionally strong evidence for mitonuclear incompatibility in the focal region of chromosome 13 in two different hybrid populations using an admixture mapping approach (Supplementary Information 1.1.5). Combined with the added simulation results (Supporting Information 1.1.1 & 1.2.3), this gives us confidence that a mitonuclear interaction is involved in generating the segregation distortion observed on chromosome 13 in F₂ hybrids.

This makes sense but seems rather arbitrary as a conclusion. For example, the dearth of ancestry could be caused by other maternal effects (such as RNA deposition), especially given that the downstream phenotype is a delay in developmental time.

We appreciate this important point about maternal effects from reviewer 3. First, additional mapping approaches (see below) now allow us to identify the genes involved in the hybrid incompatibility with more precision, pointing more conclusively to an interaction involving *ndufs5*.

We have also added analysis on maternal effects to the revised manuscript (Supplementary Information 1.1.7). Our unique dataset of siblings, who have the same maternal effects such as RNA deposition and the same mitochondrial genotype, but differ in their nuclear genotypes due to recombination and independent assortment, allows us to assess this directly in the revised manuscript. Specifically, we developmentally staged entire broods if any individual in the brood exhibited developmental lag. Under a maternal effects model, we would expect most if not all individuals in the brood to exhibit development lag, as they inherit the same maternally deposited RNAs and proteins. Instead, we observe ~50% of individuals lagging in focal families, and find a strong effect of chromosome 13 genotype on the probability that an individual will exhibit developmental lag (Fig. 2; see Supporting Information 1.1.7 for analysis details). We also note that inhibition of Complex I activity in zebrafish causes a similar developmental delay phenotype (see Melo *et al.* 2015, *Environ. Sci. Pollut. Res.*). We have added references to this work in the main text.

More generally, we were unsure if the reviewer was implying the possibility of a hybrid incompatibility between a maternally deposited RNA or protein and the chromosome 13 region. Such a mechanism has never been described (nor to our knowledge have epistasis-mediated maternal effects) but of course would be very exciting. However, we believe that given the fine-

mapping results in the revised manuscript implicating *ndufs5* along with other evidence for reduced Complex I function, there is strong evidence for the mechanism described in the paper.

Since the authors do not elaborate about the other genes in this genome segment I can't assess whether this hypothesis is possible or likely. I am not implying that the authors' conclusion is not possible, but it seems like a more systematic exploration of the alternatives is necessary.

Both our new admixture mapping results and analysis of the sibling data allowed us identify a much narrower region associated with the incompatible locus on chromosome 13, containing only three genes (*ndufs5*, *rnf19b*, and *macf1*). Notably, these are the same three genes we had originally identified using the ancestry deficit analysis in natural hybrid populations (Fig. S6). The precise concordance between these three lines of evidence (admixture mapping, sibling analysis, and ancestry deficit analysis) give us very high confidence that one of these three genes is the focal point of a mitonuclear incompatibility on chromosome 13.

We agree with reviewer 3's point regarding exclusion of other genes in this interval and address this in more detail below.

The authors then explore their conclusion of the putative mitonuclear interaction in more depth. They conclude that of the ~30 genes in the focal 380 kb segment, two have mitochondrial functions, but only one interacts directly with mitochondrially encoded proteins, implicating this gene (*ndufs5*) as the cause of the incompatibility. This rationale to narrow down the allele is somewhat odd because the difference between 'having a mitochondrial function' and 'a direct mitochondrial interaction' is never really elaborated in the manuscript, and several re-reads, I don't understand why the incompatibility must occur through proteins that physically interact with each other and not through alleles that are epistatic to each other.

We appreciate these points raised by reviewer 3. New data and analyses now allow us to narrow in on three genes (from the 32 identified in the original manuscript) that could underlie the chromosome 13 mitonuclear interaction. One of these genes is *ndufs5*, which directly interfaces with mitochondrial proteins in Complex I (i.e. directly interacts with mitochondrial proteins). We have clarified our language describing this throughout the manuscript.

Motivated by reviewer 3's concerns, we took two specific approaches to interrogate possible mitochondrial interactions of *rnf19b* and *macf1*, the other two genes in our new associated interval on chromosome 13. First, we used the *X. maculatus* STRING database to search for any known interaction between these two genes and any mitochondrial or mitonuclear gene (GO:0005739). Of 1860 annotated interactions for *macf1* and 748 annotated interactions for *rnf19b*, there were no annotated interactions with mitochondrial or mitonuclear genes for *macf1* and one annotated interaction for *rnf19b*, with the gene *park2*.

rnf19b and *park2* interact via *ube213* (Fig. S38), and play a role in removing damaged mitochondria via mitophagy (reviewed in Matsuda & Tanaka 2015). We identify several nonsynonymous changes between *X. birchmanni* and *X. malinche* in *park2* but none in *ube213*. However, we do not see evidence that ancestry at either *park2* or *ube213* is unusual in F₂ hybrids (as would be expected if this interaction were under selection). We also do not see evidence for

non-random assortment between *rnf19b* and *park2* or *rnf19b* and *ube2l3* in F₂ hybrids, again suggesting that these genotype combinations are not under selection in hybrids. These analyses are described in Supplementary Information 1.1.8.

We also performed a partial correlation analysis between mitochondrial genotypes and genotypes at *park2* and *ube2l3* in the admixture mapping population. We found no evidence for a correlation in ancestry between either of these genes and the mitochondria, nor evidence for a correlation between their ancestry and *rnf19b* (all $P > 0.3$).

We believe these analyses, described in detail in Supplementary Information 1.1.8, significantly strengthen our conclusion that the associated gene in the chromosome 13 interval is *ndufs5*.

In the third part of the manuscript, the authors state that there is evidence for a third partner to this epistatic interaction. The evidence for the three-way interaction is rather weak. The rationale is that if they are able to detect other alleles that have undergone selection in hybrids at the Mitochondrial protein respiratory Complex I, then those alleles must be part of the hybrid incompatibility. I might be missing something (the methods are brief in this aspect) but my understanding is that the whole premise of this is a partial correlation analysis in which they find that *ndufa13* has no mixed ancestry in hybrid zones.

We appreciate this point from reviewer 3 and we have now added additional data, analyses, and clarified the analyses that were present in the original manuscript. We no longer rely on the signal of selection at *ndufa13* as evidence that it is involved in the mitonuclear incompatibility, though we view it as complementary evidence.

First, we strengthened evidence for a role of *ndufa13* by collecting additional data from hybrid individuals from the admixture mapping population, increasing our sample size by ~50%. This improves our power, resulting in a signal of interaction between the chromosome 6 region containing *ndufa13* and the mitochondrial genotype that approaches significance at a 10% FPR threshold (Fig. 1).

We also include two analyses of the data from F₂ hybrids that support our inference that chromosome 6 region is incompatible with the *X. malinche* mitochondria. We note that some related data to that added here was presented in a different context (in the study of ancestry repeatability between *X. birchmanni* x *X. cortezi* and *X. birchmanni* x *X. malinche* hybrid zones) in recent paper from our group (Langdon et al. *PLoS Genetics* 2022).

- The chromosome 6 region where *ndufa13* is found is also a segregation distorter in F₂ hybrids with an *X. malinche* mitochondria (now shown in Fig. 1). Although it does not generate as extreme segregation distortion as the region containing *ndufs5*, it is a strong segregation distorter in F₂ hybrids (63% *X. malinche* ancestry at *ndufa13* among surviving individuals versus 67% *X. malinche* surrounding *ndufs5* and 50% genome-wide).
 - We now show with simulations that the paucity of *X. birchmanni* homozygotes at *ndufa13* is unexpected for a nuclear-nuclear incompatibility and instead points to a mitonuclear incompatibility (Supplementary Information 1.1.1)

- When we perform a χ^2 analysis of two-locus genotypes at *ndufs5* and *ndufa13* in F₂ hybrids, we see massive departures from expectations under independent segregation (Supporting Information 1.1.8). We note that we cannot evaluate from the F₂ data alone the specific architecture of the interaction between the mitochondria, *ndufs5*, and *ndufa13*, as all F₂ individuals derive their mitochondrial genome from *X. malinche* (as discussed in Supplementary Information 1.1.11).

Regarding the exact architecture of the mitonuclear incompatibility, Supplementary Information 1.1.11 describes in detail how the extreme lethality of homozygous *X. birchmanni* ancestry at either *ndufs5* or *ndufa13* makes it impossible to test whether they represent separate pairwise incompatibilities with the *X. malinche* mitochondria, or a single three-way epistatic interaction. Regardless, we believe the added data and analyses convincingly establish strong selection on *X. birchmanni* ancestry in the region containing *ndufa13* when it is combined with the *X. malinche* mitochondria, and provide further evidence for their interaction.

This makes me revisit a point that I made earlier. The authors state that there is no nuclear-nuclear interactions involved in the trait, and yet, the potential third partner is autosomal. This result needs to be better reconciled with the results from the genome association mapping.

We apologize for the lack of clarity in the original manuscript here and hope the revised manuscript is now clear. Our argument was not that there are no nuclear-nuclear interactions but that nuclear-nuclear interactions alone could not explain the complete absence of homozygous *X. birchmanni* ancestry at chromosome 13 (which we now show in the simulations discussed above; Supplementary Materials 1.1.1). We hope that the new data, simulations, and analyses we have added reconcile and strengthen these results for the reviewer. We also refer the reviewer to a more in-depth discussion of these issues in Supplementary Information 1.1.11 and 1.2.3.

The paper makes no effort to rule out potential causes for this long range admixture LD which makes me think that the results are less conclusive the authors make it sound.

We were not entirely clear on what the reviewer was referring to here as potential causes of long-range admixture LD. We have included additional analyses in the revised manuscript but are happy to address this more directly if we did not evaluate the issue the reviewer was referring to.

Long range admixture LD can be caused by several processes. One is selection maintaining an association between interacting alleles, the process that we conclude is important for the covariance in ancestry between the mitochondria and nuclear loci on chromosome 6 and chromosome 13. Other processes that can generate admixture LD are unaccounted for population structure and ancestry-assortative mating (Zaitlen et al. *Genetics* 2017). In the original version of the manuscript we used the $r|a$ measure that accounts for background LD and population structure (Schumer & Brandvain *Molecular Ecology* 2016). We apologize if this was unclear in the original text and have added more details to Supplementary Information 1.1.9 and to the main text Methods (lines 565-566).

We now formally evaluate evidence for assortative mating in the hybrid population used for admixture mapping (following Schumer et al. *PNAS* 2017 and Powell et al. *Evolution* 2021),

where we have access to paired mother and embryo genotypes. We find no evidence for ancestry assortative mating in this population, which can be an important contributor to long-range admixture LD (Supplementary Information 1.1.9). Specifically, mating patterns observed in the Calnali Low admixture mapping population are precisely in line with those expected under random mating by ancestry (Fig. S40).

Finally, we evaluated evidence for a misassembly impacting the chromosome 6 and chromosome 13 regions, which could artifactually cause long-range admixture LD. Using the $r|a$ metric, we show that admixture LD shows a typical pattern of decay over genetic distance in both regions, rather than a sudden drop that may be expected in the case of a misassembly (Fig. S40; Supplementary Information 1.1.9).

In short, the authors pose an audacious hypothesis that will require herculean genetic work to verify. In spite of my enthusiasm for the topic, the paper has too many assumptions which dampens my enthusiasm for the current piece.

We appreciate the reviewer's comments and suggestions for improvement of the manuscript. We believe the concerns reviewer 3 raised have been thoroughly addressed in the revised manuscript, and we hope that the assumptions they objected to are well-founded in the new version.

Referee #4 (Remarks to the Author):

The molecular mechanisms underlying hybrid incompatibility are fundamental questions in evolutionary biology. When discovered, they can explain why and how closely related species maintain reproductive barriers that control speciation. Nuclear-mitochondrial mismatch is known to be a major mechanism of hybrid incompatibility, particularly in yeast and plants. This is evidenced by mitonuclear coevolution during speciation and is a result of the strict mitochondrial-nuclear coordination required for successful biogenesis of the electron transport chain (ETC). However, the specific mismatches between the 13 mt-encoded proteins and the >120 nuclear-encoded ETC subunits/assembly factors that drive hybrid incompatibility remain undefined. In this work, Moran and colleagues, through a heroic effort using genetic mapping of naturally hybridizing wild admixture and lab-bred swordtail populations, narrow down the players responsible to hybrid incompatibility to just 4 genes: *nd2*, *nd6*, *ndufs5* and potentially *ndufa13*. They propose that incompatibility of mitochondrial *nd2/6* with nuclear *ndufs5* leads to embryonic lethality of hybrids as a result of impaired Complex I enzymatic function. Heterozygous hybrids attempt to mitigate the incompatibility by preferentially incorporating the compatible *ndufs5* homologue into CI. They further pinpoint a handful of residues that might be responsible for the incompatibility between *x.malinche* and *x.birchmanni* *nd2/6* and *ndufs5*. Lastly, they show that *ndufs6/a13* undergo rapid evolution and co-introgression with compatible mitochondrial genotypes. The authors argue that these candidates fulfill requirements of classic models as molecular determinants of hybrid incompatibility. I will judge the paper according to the following criteria, wherein I will also incorporate my critique.

We thank Reviewer 4 for describing the value of this work and the immense mapping efforts required to generate the data we present. We also very much appreciate their constructive criticisms of the manuscript, which we respond to point-by-point below.

1) Is it novel and significant?

Since the concept of mito-nuclear incompatibility is well known, conceptually this study offers a moderate advancement, in that it solidifies a known theory by advancing it to a genic level. However, it is the first study to implicate specific genes in hybrid incompatibility in wild naturally hybridizing populations, which makes it a pioneering study. Overall novelty is 8/10. The study's significance largely depends on how widely applicable these genes are to general hybrid incompatibility. Can the authors address the question of how general the genetic interactions between *nd2/6* and *ndufs5/a13* are in other hybridizing species that experience incompatibility? In simple words, how generally applicable are their findings to other hybridizing species? If so, I would say their findings are very significant and offers as broad conceptual advance.

We appreciate this important point from Reviewer 4, as well as their acknowledgement of the novelty and significance of our work on naturally hybridizing species. We think this is a very intriguing question, and an exciting direction for future work. Historically, the literature in evolutionary biology has assumed that each interacting gene pair has an equal probability of becoming involved in a hybrid incompatibility (Orr *Evolution* 1995; Turelli & Orr *Genetics* 2000; Orr & Turelli *Evolution* 2001; Presgraves *Current Biology* 2010). However, both the prevalence of mitonuclear incompatibilities (e.g. Lee et al. *Cell* 2008; Meiklejohn et al. *PLoS Genetics* 2013; Luo et al. *Nature Genetics* 2013) and emerging work on nuclear-nuclear

incompatibilities from our group and other groups (Powell et al. *Science* 2020, Lu et al. *PNAS* 2020, Alcázar et al. 2014 *PLoS Genetics*; Chae et al. 2014 *Cell*) suggest that this assumption is not well-founded in the empirical literature. Based on the results of our study, we consider it very likely that other hybrid incompatibilities will localize to the interface of mitochondrial and nuclear proteins in Complex I, where coevolution drives correlated amino acid changes (see added figures - Fig. S31, S45, S46). However, whether identical genes are likely to be involved is an exciting question that awaits results from future work.

2) Are the data of high quality?

Yes, reading through the methods, this study was very well executed. Data quality 9/10.

We thank the reviewer for their recognition of the quality of our data, which we believe is further strengthened in the revised manuscript.

3) Did the authors make the right interpretation? Is it likely to be true?

This is perhaps what I found to be the most problematic with the study. The segregation patterns demonstrated in Figure 2A, B in viable versus non-viable hybrids convinces me that the authors have indeed identified the correct locus. However, given that there are 32 genes in this interval, I am not entirely convinced that they have indeed identified the right gene.

We appreciate this comment from reviewer 4, which was also noted by one of the other reviewers. In response to this comment, we have increased our sample size by ~50%, giving us very high resolution of the region containing *ndufs5*. Based on this mapping, we identify only three genes, including *ndufs5*, each of which we explore in detail for evidence of mitonuclear interaction in Supplementary Information 1.1.8. Moreover, this fine-mapping result is bolstered by a unique analysis we have added comparing local ancestry in developmentally delayed versus normally developing siblings (Fig. 2, Supplementary Information 1.1.7). Together, these analyses provide compelling evidence that *ndufs5* is indeed the causal gene on chromosome 13.

I raise this objection because the small reduction seen in Complex I efficiency (Fig 2D), which does not affect maximal respiration (Fig 2E) is unlikely to have such profound impacts on embryonic development. Similarly, Fig 2G demonstrates a 10% preference bias in incorporation of the compatible allele. If the incompatible *x.birchmanni* proteins really destabilize CI leading to lethal effects, these should never be incorporated/stable enough to be detected.

We thank reviewer 4 for this concern, which was also raised by reviewer 2. With increased confidence that the genes in question are involved in the hybrid incompatibility, we can better evaluate alternative hypotheses for moderate impacts on Complex I function in individuals heterozygous for the incompatibility.

We agree with reviewer 4 that modest incorporation bias is perhaps puzzling. If heterozygotes suffer no viability consequences, and this is associated with a ~10% preference bias in incorporation of the compatible allele, one might assume that homozygotes should have only moderately reduced function, as the reviewer outlines. However, we note that **destabilization that prevents incorporation may not be the only relevant mechanism**. If mitochondria containing assembled Complex I with the *X. birchmanni* nuclear genes and *X. malinche* mitochondrial genes are subfunctional but still tend to assemble, we might expect exactly what we observe here: heterozygous individuals displaying a modest incorporation bias and reduced

Complex I function. By contrast, homozygotes would have limited buffering capacity if all their mitochondria were subfunctional, providing a natural explanation for lethality differences between heterozygotes and homozygotes.

In addition to exploring some of the possibilities suggested by reviewer 4 below, in the revised manuscript we also highlight an exciting piece of data that was detected in our initial experiments but that we did not previously present. Specifically, in addition to reduced Complex I efficiency, we see quite a dramatic difference between individuals heterozygous for the hybrid incompatibility and their parent species in the time required to reach peak Complex I-driven respiration (Fig. 3; S20). This difference is not detected in measurements of time required to reach peak Complex II-driven respiration (Fig. 3). This intriguing observation is another indication of reduced Complex I function in heterozygotes.

Generally, a 10% drop in CI activity will not lead to lethality due to the inherent buffering capacity of mitochondria and of high glycolytic rates during larval development. Zebrafish larvae can withstand rotenone treatment to quite high levels (>100nM, which significantly affects CI activity) and still remain viable (PMID: 28732770).

In regards to general questions about the potential for Complex I dysfunction to cause lethality, we explored buffering capacity of swordtail fry treated with rotenone directly (as suggested by reviewer 4 here and below). We find that newborn swordtail fry have dramatically reduced survival after rotenone exposure, with 93% mortality after 24 hours of exposure to 100 nM of rotenone and 100% mortality after 48 hours of exposure. This experiment is described in Supplementary Information 1.3.2.

Reviewer 4's comments here are also complicated by the fact that viability in swordtail embryos is strongly dependent on completing development in the maternal environment. We discuss this in more detail and describe new data collected in the next section.

Model organisms like mice, and humans, can withstand NDUFS4 mutations and be born live, only to succumb to Leigh syndrome in childhood. Therefore, the authors conclusion that a small reduction in CI efficiency can lead to such drastic negative selection needs further examination.

We found this comment from reviewer 4 very interesting as it compelled us to think about the unique biology of swordtails and how it may relate to mammalian models, which we now discuss briefly in the manuscript (lines 232-234). We note again that the modest reduction in Complex I function that we report in heterozygotes (Fig. 3) is expected to be more extreme in homozygotes.

We showed in the original manuscript that swordtail embryos with *X. malinche* mitochondria and homozygous *X. birchmanni* ancestry at *ndufs5* lag their siblings in development. **We now show that developmental lag is specifically associated with an excess of *X. birchmanni* ancestry at *ndufs5* (Fig. 2).** As the reviewer points out, this is reminiscent of developmental delays in zebrafish with pharmacological Complex I inhibition (Melo *et al.* 2015, *Environ. Sci. Pollut. Res.*). While in zebrafish this developmental delay is not lethal, in swordtails embryos born even a few days prematurely do not survive (see Supplementary Information 1.3.1). We have added additional data on the phenomenon to the revised manuscript (Table S4).

Thus, the interaction between Complex I-mediated developmental delay and the developmental requirements of swordtails may be precisely the explanation for the lethality of this hybrid incompatibility. This result highlights exciting similarities and differences to Complex I dysfunction in other species.

Some suggestions to enhance the credibility of the authors' conclusions:

- i. Demonstrate that mild inhibition of CI pharmacologically can lead to lethality of *Xiphophorus* larvae.
- ii. Demonstrate that *X. birchmanni* *ndufs5* is sufficient to destabilize CI in cells with *X. malinche* mt-genome. Some suggestions:
 - Blue Native PAGE of liver/muscle mitochondria from heterozygous hybrids compared to parental mitochondria should reveal significant CI disassembly, particularly when probed with proteins from the ND1/2 modules.
 - Recapitulate bioenergetic and biochemical defects in fibroblasts derived from *X. malinche* with stable overexpression of *X. birchmanni* *ndufs5*.
 - Recapitulate the proposed incompatible residues in a cell culture system i.e. mutate the substituted NDUFS5 residues in mammalian cell culture system and demonstrate that it leads to mito-nuclear incompatibility. This would also help to prove that this system works in even higher organisms.
 - In experimental genetics, the classic way to prove that something is responsible for a phenotype is to rescue it. I am not sure if transgenic techniques work for swordfish. Is there was a way to introduce *X. malinche* *ndufs5* mRNA into hybrids to rescue the *X. birchmanni* x *X. malinche* *ndufs5* homozygotes? That would be very convincing.

Any other methods to demonstrate that the other 31 genes are not responsible will also be welcomed.

We believe that the new mapping results in the revised manuscript that precisely localize the incompatible region on chromosome 13 to three genes address reviewer 4's main concerns here. However, we appreciated the suggestions for other data that would improve the strength of our study and have collected new data where possible (see below).

Regarding some of the specific suggestions, we note that swordtails are not a model system (with only about a dozen labs working on any swordtail species globally), and thus many of the approaches referenced here are not currently possible. Specifically, there are no cell lines for any of the species we study and transgenic approaches have thus far failed in the entire family to which swordtails belong, including guppies, mollies, and other species. Regarding the reviewer's suggestions for using mammalian cell lines, the nuclear proteins in question have ~90% sequence similarity to mammalian orthologs and ~80% similarity to mitochondrial orthologs, making modifications of mammalian cell lines complex and any results difficult to interpret.

Inspired by reviewer 4's suggestions, we performed experiments to test buffering capacity in response to rotenone treatment which pharmacologically inhibits Complex I. We exposed newborn *Xiphophorus* fry to 100 nM of rotenone and DMSO or a DMSO control (it is not currently possible to maintain viability of *Xiphophorus* embryos outside of the maternal

environment). While fry exposed to DMSO had 100% viability over 48 hours, fry exposed to DMSO and 100 nM of rotenone experienced 93% mortality over 24 hours (n=15 of each group) and 100% mortality over 48 hours. These results are now described on lines 234-239 and Supplementary Information 1.3.2 and indicate that *Xiphophorus* fry are more sensitive to Complex I inhibition than zebrafish.

Reviewer Reports on the First Revision:

Referees' comments:

Referee #3 (Remarks to the Author):

This is a revised manuscript by Benjamin Moran and collaborators in which the authors suggest a potential hybrid incompatibility between three genes within respiratory Complex I (between the nuclear genome of *Xiphophorus birchmanni* and the mitochondrial genome of *X. malinche*). The authors revised their manuscript and have added additional individuals to their mapping population. In general, the manuscript remains an important piece because it reveals alleles potentially involved in a hybrid inviability phenomenon but my original concern, that there is no conclusive test of the three-way interaction, remains. As it stands, the current results can be alternatively be interpreted as multiple incompatibilities with an additive effect. This is an important limitation in the manuscript because the whole framing in the introduction (lines 54-68) discuss the potential importance of complex interactions (broadly defined as multiallelic epistatic interactions) in hybrid inviability. After my review that reported this concern, the authors have added a substantial section with population genetics simulations aiming to rule out potential nuclear-nuclear interactions. I appreciate the effort and the willingness to engage with the review, but in my opinion, this line of evidence is not sufficient to prove the nature of the interaction. I would argue that the disagreement here might be of standard of evidence regarding the evidence to demonstrate a genetic interaction. In my opinion, to properly demonstrate the existence of a genetic interaction, the authors will have to do forward genetic assays and the presented simulations are simply not enough. This is an impressive population genetics tour de force and in that regard has the potential to become an important piece and admittedly the authors have softened some of their claims (e.g., line 36 where they state that these are the alleles 'most likely to contribute'). Nonetheless, since this is such a big claim it needs to be substantiated with functional genetics work and speculation is less than ideal. I am afraid they have not provided such level of evidence and my enthusiasm for the piece remains moderate.

Referee #4 (Remarks to the Author):

I appreciate that the reviewers have undergone extensive additional genetic analyses and an increase in sample size to strengthen their conclusion that *ndufs5* is responsible for the nuclear-mito incompatibility leading to hybrid incompatibility.

I also note that they have demonstrated that heterozygotes have an increase in latency to maximal Complex I activity, as well as a lethal effect of rotenone at 100nM to *Xiphophorus* fry.

I have no further comments regarding the validity of *ndufs5* being the determinant of mito-nuclear incompatibility in hybrids.

However, I am still unconvinced that the mechanism proposed by the authors - that mismatches in contact sites (deduced purely by modeling) are the cause of mild Complex I defects that ultimately lead to embryonic lethality - holds up to experimental scrutiny.

100nM is very toxic to zebrafish fry, and would be predicted to shut down Complex I instead of lead

to mild CI defects (PMID: 28732770). There are also prior reports that a longer latency to maximal activity is a determinant of overall OXPHOS competence. Complex I activity is inherently rate limited by Complex III anyway, so I would not expect a slower on-rate to affect overall respiration, as evidenced by the unchanged CI+II activity.

I appreciate that *Xiphophorus* is not a model organism where genetic manipulations are possible, nor cell culture, so the data required to convince me are impossible to obtain. However, these inherent limitations also make it impossible for the authors to make firm conclusions regarding the exact mechanism of hybrid incompatibility. As such, while the genetics are solid, the proposed mechanism is speculative at best.

Referee #5 (Remarks to the Author):

This is super interesting finding but still lacks a mechanistic explanation. Difficult arises from lack of measuring mitochondrial complex I function (NAD⁺ regeneration tied to metabolite production, ATP tied to proton pumping, ROS production).

I would like to focus on Reviewer 4 comments: "Model organisms like mice, and humans, can withstand NDUF54 mutations and be born live, only to succumb to Leigh syndrome in childhood. Therefore, the authors conclusion that a small reduction in CI efficiency can lead to such drastic negative selection needs further examination."

The authors response was: "We found this comment from reviewer 4 very interesting as it compelled us to think about the unique biology of swordtails and how it may relate to mammalian models, which we now discuss briefly in the manuscript (lines 232-234). We note again that the modest reduction in Complex I function that we report in heterozygotes (Fig. 3) is expected to be more extreme in homozygotes."

Two recent reports in mice can shed light on this critical aspect.

1. NDUF52 loss (a key catalytic component that abolishes complex I activity) is embryonic lethal but the heterozygous have mild decrease in complex I activity but no phenotype even as they age (PMID: 35338200).

2. A more relevant study is the observation that yeast single protein NDI1 (NADH dehydrogenase) can rescue NDUF54 leigh syndrome mouse model (PMID: 35338200).

Key experiment that needs to be conducted.

I think an important experiment is to test whether ectopic expression of NDI1 rescues in the current study. If it does then it is complex I function is driving the phenotype. If it does not then it is still interesting as it suggests something else about mismatch hybrid effects survival. As a control, rotenone toxicity will be prevented by NDI1 expression.

Response to editor

Our manuscript addresses one of the oldest problems in evolutionary biology: how genetic incompatibilities evolve and what their consequences are after hybridization. While past work has almost exclusively focused on tractable lab models that do not naturally hybridize, we uniquely tackle this question in a naturally hybridizing vertebrate system. Because the species we study are wild organisms where genetic manipulations are simply not possible, certain experiments are currently out of reach, but in the enclosed manuscript we have taken creative approaches to design new experiments that address outstanding reviewer concerns. We also attempted other experiments suggested by the editorial team that are not included in the resubmitted manuscript, which we describe briefly below.

As we discussed when we spoke last fall, studying the protein-protein interactions of Complex I in a wild organism brings a suite of technical challenges that may take many years to overcome. While Co-IP approaches would be an ideal way to examine whether there is less binding between matched versus mismatched versions of the proteins, we found that commercially available antibodies for *ndufa13* and *ndufs5* were not suitable for use in *Xiphophorus*, either because they did not bind or they only bound when Complex I was denatured.

Given that these approaches were unsuccessful, the editorial team suggested alternative approaches including SPR and FRET. One of the conceptual challenges with these suggested approaches is the sheer complexity of the Complex I structure and assembly, which has made biochemical work notoriously challenging (reviewed in Formosa et al. *Seminars in Cell & Developmental Biology* 2018). Complex I is a 45-protein complex with 15 assembly factors and is partially embedded in the mitochondrial membrane. Indeed, two of the proteins of interest are membrane embedded (see Figure 4A). Moreover, key features of the assembly of the complex are still poorly understood; current data supports the idea that Complex I is initially assembled into sub-complexes which are later stitched together and anchored to the mitochondrial membrane (Vercellino & Sazanov *Nat Rev Mol Cell Biol* 2022). Thus, as we discussed last fall, interpreting the results of the biochemical assays proposed by the editorial team on Complex I proteins in isolation is extraordinarily challenging and may not reflect biologically meaningful results.

Despite these considerations we did attempt to produce large quantities of protein for SPR experiments in collaboration with Professor Judith Frydman's lab (which ultimately were unsuccessful due to inadequate concentrations of most proteins, discussed in more detail below). After this, we also worked with a core facility at Stanford to synthesize portions of these peptides, which was successful for *X. birchmanni* and *X. malinche* ND6 but not for the key residues of NDUFS5 or NDUFA13 because of issues generated by the highly hydrophobic parts of the proteins that are typically membrane embedded. Given that these approaches were unsuccessful despite substantial effort, **we followed your suggestion to modify our language throughout the manuscript regarding how substitutions that differ between species impact protein-protein interactions in Complex I.** Specifically, we have moved much of our discussion about substitutions that are predicted to be in physical contact to the supplement and interpreted modeling results cautiously, making it clear that such inferences await experimental validation (lines 197-198; 335; 352-359; 429; 544-545; 547-548). We have also added a note to

the manuscript clarifying that genetic manipulations are not possible in this system (lines 356-359), as this appears to have been unclear to several of the reviewers.

To address the remaining reviewer concern, we undertook an ambitious series of experiments to rigorously evaluate whether the interactions between the *X. malinche* mitochondria, *ndufs5*, and *ndufa13* are best characterized as two simple hybrid incompatibilities that coincidentally involve neighboring proteins in Complex I (Figure 4A), or a ‘complex’ three-way hybrid incompatibility. The ubiquity of three-way hybrid incompatibilities was proposed based on theoretical results in evolutionary biology nearly 30 years ago but there are only two cases to our knowledge that have directly mapped the genes involved in such an incompatibility and investigated the nature of their interactions (in yeast: Boocock et al. *Science* 2021; in *Drosophila*: Phadnis et al. *Science* 2015 & Cooper et al. 2018, 2019).

In our revised manuscript, we have added a developmental study of 235 F₂ embryos with all possible genotype combinations to investigate the consequences of ancestry mismatch at *ndufs5* and *ndufa13* separately and in combination. We also pinpoint the developmental stage at which individuals with *ndufa13* mismatch suffer lethality. Using pre and post-birth developmental tracking, we found clear evidence that the developmental timing of lethality differed between *ndufs5* and *ndufa13* (Figure 2C), which is expected from two independently acting hybrid incompatibilities. We also found intriguing evidence that the proximate causes of lethality are likely different. While *ndufs5* incompatible individuals had depressed respiration and gross developmental abnormalities, *ndufa13* incompatible individuals had heart defects (among other abnormalities observed in both genotypes; Supplementary Information 1.3.4-1.3.5). This is an intriguing observation given the association between mutations in Complex I genes and cardiovascular diseases in mammals.

The separable phenotypic impacts of ancestry mismatch at *ndufs5* and *ndufa13* could indicate that although the proteins are in physical contact in Complex I, they represent distinct hybrid incompatibilities. However, despite their distinct phenotypic effects, we also found evidence that **their impacts on survival were not independent**. Using an expanded dataset of 1010 F₂ hybrids, we found that rare survivors of the *ndufa13* incompatibility were unlikely to harbor even one *X. birchmanni* allele at *ndufs5*. In other words, we found a much lower frequency of *ndufs5* heterozygotes in individuals that survived the *ndufa13* incompatibility than expected from selection on *ndufs5* in isolation (lines 272-285; Supplementary Information 1.2.5; Figure 2I).

Crucially, our new results highlight a major shortcoming in current work on hybrid incompatibilities in the evolutionary biology community. While the idea that hybrid incompatibilities are common and evolve rapidly between species is a cornerstone of modern evolutionary biology, theoretical work has continued to outpace empirical research in this area. This has led to widely accepted principles – such as the idea that three-way incompatibilities are common – that have not been deeply investigated empirically. Our results highlight the ways in which the **biology of hybrid incompatibilities can be profoundly more complex than the simple models historically used**. Instead of a two-way or three-way incompatibility, we uncover previously unappreciated complexity in genetic and phenotypic interactions in hybrids. Given that so little is known empirically about the genetic basis of hybrid incompatibilities and what

factors drive their evolution, especially in naturally hybridizing species, we believe that the revised manuscript will have a major impact on the field of evolutionary genetics.

Our revised manuscript now lends further insight into the complex mechanisms through which hybrid incompatibilities evolve. We have added additional data on what we believe is one of the most impactful results of our manuscript – but has not been the focus of reviewer feedback thus far – the evidence that historical hybridization between *X. malinche* and a third species, *X. cortezi*, **led to the introduction of the mitonuclear incompatibility into *X. cortezi***. This is an entirely novel and previously unknown mechanism through which hybrid incompatibilities can arise. **We provide further data in the revised manuscript that shows that this introgressed incompatibility results in selection on *ndufs5* and *ndufa13* when *X. cortezi* and *X. birchmanni* hybridize in nature** (Figure 5). This surprising result shows that genetic incompatibilities can be passed between species via a complex history of hybridization. Reviewer 1, who was unavailable to review the previous revision described this particular result as “tantalizing and exciting,” and we believe that the result is further strengthened with our new data. This finding uncovers, for the first time, the varied evolutionary paths through which hybrid incompatibilities can evolve.

Author Rebuttals to First Revision:

Response to reviewer 3

This is a revised manuscript by Benjamin Moran and collaborators in which the authors suggest a potential hybrid incompatibility between three genes within respiratory Complex I (between the nuclear genome of *Xiphophorus birchmanni* and the mitochondrial genome of *X. malinche*). The authors revised their manuscript and have added additional individuals to their mapping population. In general, the manuscript remains an important piece because it reveals alleles potentially involved in a hybrid inviability phenomenon but my original concern, that there is no conclusive test of the three-way interaction, remains.

We thank reviewer 3 for their recognition of the importance of our manuscript in contributing to the understanding of hybrid incompatibilities, which are poorly understood in general, particularly in species that naturally hybridize.

As it stands, the current results can be alternatively be interpreted as multiple incompatibilities with an additive effect. This is an important limitation in the manuscript because the whole framing in the introduction (lines 54-68) discuss the potential importance of complex interactions (broadly defined as multiallelic epistatic interactions) in hybrid inviability. After my review that reported this concern, the authors have added a substantial section with population genetics simulations aiming to rule out potential nuclear-nuclear interactions. I appreciate the effort and the willingness to engage with the review, but in my opinion, this line of evidence is not sufficient to prove the nature of the interaction. I would argue that the disagreement here might be of standard of evidence regarding the evidence to demonstrate a genetic interaction. In my opinion, to properly demonstrate the existence of a genetic interaction, the authors will have to do forward genetic assays and the presented simulations are simply not enough. This is an impressive population genetics tour de force and in that regard has the potential to become an important piece and admittedly the authors have softened some of their claims (e.g., line 36 where they state that these are the alleles ‘most likely to contribute’). Nonetheless, since this is such a big claim it needs to be substantiated with functional genetics work and speculation is less than ideal. I am afraid they have not provided such level of evidence and my enthusiasm for the piece remains moderate.

We took seriously reviewer 3’s concern in our revised manuscript and have focused substantial effort and data collection on answering this question. We believe the findings of the creative experiments we have included in our revised manuscript directly address the reviewer’s concerns, while acknowledging that because the species we study are wild organisms – not lab models – we are unable to use the precise approaches discussed by the reviewer. Indeed, we think that this is exactly what makes our study exciting for the field, since it studies hybrid incompatibilities and their evolutionary causes in actively hybridizing species (in contrast to most previous studies that have successfully mapped incompatibilities, e.g. Boocock *et al. Science* 2021, Lu *et al. PNAS* 2020, Phadnis *et al. Science* 2015).

In the revised manuscript, we have performed detailed physiological and developmental studies of F₂ embryos with all possible genotype combinations (a total of 235 embryos). This allowed us to document the consequences of mismatched ancestry at *ndufs5*, *ndufa13*, or both in individuals with the *X. malinche* mitochondria, before the onset of embryonic lethality (Figure 2). We also

tracked individuals throughout their lives to understand the developmental stage at which lethality due to *ndufa13* mismatch occurs. **This rich dataset gives us unprecedented insights into the mechanisms through which lethality due to hybrid incompatibility is occurring in a naturally hybridizing vertebrate system.** We summarize our results briefly here and in more detail in the revised manuscript (lines 241-285).

As previously reported, we confirm that embryos with mismatched ancestry at *ndufs5* do not complete embryonic development. We find that these embryos have reduced rates of respiration throughout embryonic development compared to heterozygous individuals (Figure 2). By contrast, embryos with *ndufa13* mismatch complete embryonic development. However, these individuals have cardiac defects – specifically enlarged sinu-atrium and reduced heart rate – that may explain challenges to survival later in life. Post-birth, we find that these individuals most frequently die within the first day of life but a small fraction survive to adulthood (these surviving individuals also tend to have cardiac defects; Supplementary Information 1.3.3).

Importantly, we found that **rare surviving individuals with the *ndufa13* incompatibility were less likely to have any *X. birchmanni* alleles at *ndufs5*.** This suggests that ancestry mismatch at *ndufs5*, even in the heterozygous state, may sensitize individuals to the *ndufa13* incompatibility. (We note that a comparable analysis cannot be performed for individuals homozygous for the *ndufs5* incompatibility as we have demonstrated 100% lethality of this genotype; see lines 141-144; 228-232).

Thus, we believe that our results are most consistent with two strong two-way interactions and a weak three-way interaction between *ndufs5*, *ndufa13*, and the mitochondria. Viewing these as simply separate two-way incompatibilities would be overly reductive, as there are non-additive impacts on fitness of distinct genotype combinations at *ndufs5* and *ndufa13* that cannot be predicted from interactions with the mitochondria alone (Figure 2I). We hope the reviewer will find this important question satisfactorily addressed in the revised manuscript.

This rich picture of the phenotypic and fitness effects of hybrid incompatibility loci opens up a number of new questions for the field. **Importantly, we find that our data does not cleanly fit into existing models, highlighting the fundamental importance of evaluating long-standing theoretical hypotheses in evolutionary genetics.** This finding adds to results reported in the previous version of the manuscript that showed that the fitness effects of different mitonuclear genotype combinations are unexpected under current theoretical models of how hybrid incompatibilities evolve (Supplementary Information 1.2.4; Fig. S56).

We note that we have also substantially revised the framing of the introduction and discussion in response to the reviewer's comments. In particular, we have revised our discussion of the architecture of hybrid incompatibilities. We have also emphasized our finding that the mitochondrial genome and interacting nuclear genes play a large role in the evolution of hybrid incompatibilities in this system, and that rapid evolution at the protein level may be important in the emergence of hybrid incompatibilities.

In the reviewer's comments they also alluded to simulations regarding nuclear-nuclear interactions added in the last revisions. We want to clarify that we no longer rely solely on

simulations for this point but directly on admixture mapping scanning for statistical associations between each of the regions identified as mitochondrially interacting. While we do not present this data in the paper, we summarize it here. There were no nuclear regions that passed the genome-wide significance threshold for interactions with *ndufs5* or *ndufa13* in our admixture mapping dataset. Consistent with the results from our analyses in F₂ hybrids described above, we do detect a modest statistical interaction between genotypes at *ndufa13* and *ndufs5* (after accounting for mitochondrial ancestry; $p=0.014$). Given that these associations are substantially weaker than those detected with admixture mapping as a function of mitochondrial ancestry, this highlights that the data cannot be interpreted as a nuclear-nuclear incompatibility that does not involve the mitochondria. If we have misinterpreted the reviewer's comments we are happy to revisit this point.

Response to reviewer 4

I appreciate that the [authors] have undergone extensive additional genetic analyses and an increase in sample size to strengthen their conclusion that *ndufs5* is responsible for the nuclear-mito incompatibility leading to hybrid incompatibility.

I also note that they have demonstrated that heterozygotes have an increase latency to maximal Complex I activity, as well as a lethal effect of rotenone at 100nM to *Xiphophorus fry*. I have no further comments regarding the validity of *ndufs5* being the determinant of mito-nuclear incompatibility in hybrids.

We are glad that the experiments and analyses included in the previous revisions clarified these questions for reviewer 4.

However, I am still unconvinced that the mechanism proposed by the authors - that mismatches in contact sites (deduced purely by modeling) are the cause of mild Complex I defects that ultimately lead to embryonic lethality - holds up to experimental scrutiny. 100nM is very toxic to zebrafish fry, and would be predicted to shut down Complex I instead of lead to mild CI defects (PMID: 28732770). There are also prior reports that a longer latency to maximal activity is a determinant of overall OXPHOS competence. Complex I activity is inherently rate limited by Complex III anyway, so I would not expect a slower on-rate to affect overall respiration, as evidenced by the unchanged CI+II activity.

We thank reviewer 4 for these comments, and have added additional experiments to the revised manuscript that we believe address these concerns. First, we have repeated rotenone experiments in newborn *X. birchmanni* and *X. malinche* fry at lower concentrations (those that have a less severe impact on zebrafish fry). Intriguingly, we found that while *X. birchmanni* fry (with *X. birchmanni* mitochondria) are largely resilient to concentrations of 75 nM of rotenone, *X. malinche* fry suffer substantial lethality at these concentrations (Supplementary Information 1.3.2). This is an interesting observation as it suggests that individuals with *X. malinche* mitochondria may be more susceptible to Complex I dysfunction in general, which could then be exacerbated by ancestry mismatch at Complex I proteins. Consistent with this, we previously found that individuals with *X. birchmanni* mitochondria and ancestry mismatch at *ndufs5* still experience higher than expected mortality (Figure 1; Supplementary Information 1.2.1), but have substantially higher rates of survival than their counterparts with *X. malinche* mitochondria.

More importantly, in the revised manuscript, we directly determine overall rates of respiration in embryos with *X. malinche* mitochondria and ancestry mismatch at *ndufs5*, *ndufa13*, or both. We find that in the homozygous state, *ndufs5* mismatch dramatically depressed overall respiration (Figure 2), while *ndufa13* mismatch has more mild effects on respiration (lines 269-271). By contrast, *ndufa13* mismatch causes cardiovascular abnormalities not observed in other genotypes. Both incompatible genotypes impact heart rate, head width, and length (Figure 2; Supplementary Information 1.3.5).

We appreciate reviewer 4's point that we have not satisfactorily addressed their concern that mismatch at specific contact sites are the cause of hybrid incompatibility. We address this concern in concert with other comments from the reviewer in the next paragraph.

I appreciate that *Xiphophorus* is not a model organism where genetic manipulations are possible, nor cell culture, so the data required to convince me are impossible to obtain. However, these inherent limitations also make it impossible for the authors to make firm conclusions regarding the exact mechanism of hybrid incompatibility. As such, while the genetics are solid, the proposed mechanism is speculative at best.

As reviewer 4 acknowledges, *Xiphophorus* is not a model organism, making it difficult to directly address the hypothesis we posed in the original version of the manuscript: that mismatch in amino acid sequences in physical contact is important in the breakdown of Complex I function in hybrid individuals. However, we took seriously the reviewer's concerns and tried to address them with other approaches. Specifically, we initially sought to use SPR approaches recommended by the editorial team to evaluate potential binding affinity between the *X. birchmanni* and *X. malinche* alleles of *nd6*, *ndufs5*, and *ndufa13* compared to within-species combinations. We note that interpretation of such experiments is limited by the fact that Complex I is a 45-protein complex and two of our proteins of interest are embedded in the mitochondrial membrane, but that pursuing them seemed reasonable in the absence of tractable *in vivo* or *in vitro* methods for *Xiphophorus*. In practice, we found that the proteins of interest were extremely challenging to purify despite nearly a year of effort from our co-author Dr. Judith Frydman's lab, and we were also not able to successfully synthesize all of the peptides of interest (three out of six peptides could not be successfully synthesized).

Since we were unable to address this question directly, we have substantially changed our language throughout the manuscript and instead present this as a hypothesis we hope to address in future work (lines 197-198; 335; 352-359; 429; 544-545; 547-548). We have also re-written the manuscript to better emphasize the impact of our work from the perspective of evolutionary biologists. While *Xiphophorus* is not amenable to approaches used in other systems, we argue that **this is exactly what makes this study especially interesting: *Xiphophorus* is a wild organism that naturally hybridizes, and thus, unlike many previous studies, the effects we identify here impact the evolution of these species in nature.**

Response to reviewer 5

This is super interesting finding but still lacks a mechanistic explanation. Difficult arises from lack of measuring mitochondrial complex I function (NAD⁺ regeneration tied to metabolite production, ATP tied to proton pumping, ROS production).

I would like to focus on Reviewer 4 comments: “Model organisms like mice, and humans, can withstand NDUFS4 mutations and be born live, only to succumb to Leigh syndrome in childhood. Therefore, the authors conclusion that a small reduction in CI efficiency can lead to such drastic negative selection needs further examination.” [..]

Two recent reports in mice can shed light on this critical aspect.

1. NDUFS2 loss (a key catalytic component that abolishes complex I activity) is embryonic lethal but the heterozygous have mild decrease in complex I activity but no phenotype even as they age (PMID: 35338200).

2. A more relevant study is the observation that yeast single protein NDI1 (NADH dehydrogenase) can rescue NDUFS4 leigh syndrome mouse model (PMID: 35338200).

Key experiment that needs to be conducted.

I think an important experiment is to test whether ectopic expression of NDI1 rescues in the current study. If it does then it is complex I function is driving the phenotype. If it does not then it is still interesting as it suggests something else about mismatch hybrid effects survival. As a control, rotenone toxicity will be prevented by NDI1 expression.

We thank reviewer 5 for highlighting the importance of our findings. We agree that these results will be exceptionally interesting to Nature’s broad readership, and our results are particularly groundbreaking in the field of evolutionary biology, where this is only the third hybrid incompatibility identified in a vertebrate system.

We appreciate reviewer 5 pointing us to this relevant literature, which we have now cited in the manuscript (lines 311-313). Unfortunately, as now discussed in lines 356-359 of the manuscript, *Xiphophorus* is not a model system in which tools to pursue mechanistic studies are available. As noted in our previous response to reviewers, only about a dozen labs are working on any swordtail species globally, and thus many of the approaches referenced here are not currently possible. Specifically, there are no cell lines for any of the species we study and transgenic approaches have thus far failed in the entire family to which swordtails belong, including guppies, mollies, and other species. Indeed, to our knowledge, **we are the only lab in the world successfully maintaining *X. malinche* in captivity.**

We realize that these limitations mean that we are less able to delve into mechanistic explanations proposed by the reviewer than might be possible in species such as zebrafish, mice or *Drosophila*. However, **we believe that the true strength of our manuscript is in the insights into fundamental evolutionary questions that have not been previously studied empirically.** Specifically, our results uncover unappreciated complexity regarding the genetic architecture of hybrid incompatibilities and the mechanisms through which they evolve, as well as their importance in locally restricting gene flow in naturally hybridizing populations (lines 403-413).

Nevertheless, we believe that physiological experiments added in our revised manuscript are helpful in interpreting these questions. We now show that embryos homozygous for mismatched *ndufs5* have extremely low overall rates of respiration (individuals with mismatched ancestry at *ndufa13* have less pronounced respiratory defects, lines 269-271). Combined with previous data showing reduced Complex I driven respiration in *ndufs5* heterozygotes, we believe this data points to an inability to compensate for reduced Complex I activity in individuals homozygous for *ndufs5* mismatch, ultimately contributing to the lethality of this genotype. We find evidence that individuals incompatible at *ndufa13* suffer lethality at a distinct developmental stage shortly after birth, potentially through cardiovascular defects, as well as evidence for more subtle interactions between the two incompatible genotype combinations (Figure 2; Supplementary Information 1.2.5). **Together, this work gives us unprecedented insight into the genetic and evolutionary mechanisms underlying a lethal hybrid incompatibility.**

Reviewer Reports on the Second Revision:

Referees' comments:

Referee #4 (Remarks to the Author):

Given the constraints of the experimental system, I believe that the authors have done as much as possible to address my previous concerns that Complex I deficiency could be the cause of the hybrid lethality through incompatibility.

I was impressed by the expanded examination of the phenotypes of homozygous hybrid embryos, which has convinced me that the incompatibility at S5 and A13 is indeed sufficiently severe to cause lethality and negative selection of hybrids.

As such, from my point of view of Complex I biology, I believe authors have done all that is possible, and I have no further concerns.

Referee #6 (Remarks to the Author):

The authors provide compelling evidence for the genetic basis of hybrid incompatibilities in swordtails. Although they are limited by the non-model nature of the system (i.e., cannot genetically modify individuals to confirm exactly what is going on), I think that is what makes this paper the most exciting and I appreciate what the authors have suggested as future avenues of research. In my opinion, the authors have done an excellent job in addressing previous reviewer concerns and the addition of the F2 developmental study is outstanding and has no comparison in the current evolutionary biology literature. I think the language has been toned down enough regarding exactly how the mutations impact interactions in mitochondrial complex 1. Overall I commend the authors on an excellent contribution. It is well written, thorough, and thoughtful.